# Estimating Rank-One Spikes from Heavy-Tailed Noise via Self-Avoiding Walks

**Jingqiu Ding**[*]
ETH Zurich
jding@ethz.ch

**Samuel B. Hopkins**[*]
UC Berkeley
hopkins@berkeley.edu

**David Steurer**[*]
ETH Zurich
dsteurer@inf.ethz.ch

## Abstract

We study symmetric spiked matrix models with respect to a general class of noise distributions. Given a rank-1 deformation of a random noise matrix, whose entries are independently distributed with zero mean and unit variance, the goal is to estimate the rank-1 part. For the case of Gaussian noise, the top eigenvector of the given matrix is a widely-studied estimator known to achieve optimal statistical guarantees, e.g., in the sense of the celebrated BBP phase transition. However, this estimator can fail completely for heavy-tailed noise.

In this work, we exhibit an estimator that works for heavy-tailed noise up to the BBP threshold that is optimal even for Gaussian noise. We give a non-asymptotic analysis of our estimator which relies only on the variance of each entry remaining constant as the size of the matrix grows: higher moments may grow arbitrarily fast or even fail to exist. Previously, it was only known how to achieve these guarantees if higher-order moments of the noises are bounded by a constant independent of the size of the matrix.

Our estimator can be evaluated in polynomial time by counting self-avoiding walks via a color coding technique. Moreover, we extend our estimator to spiked tensor models and establish analogous results.

## 1 Introduction

Principal component analysis (PCA) and other spectral methods are ubiquitous in machine learning. They are useful for dimensionality reduction, denoising, matrix completion, clustering, data visualization, and much more. However, spectral methods can break down in the face of egregiously-noisy data: a few unusually large entries of an otherwise well-behaved matrix can have an outsized effect on its eigenvectors and eigenvalues.

In this paper, we revisit the *single-spike recovery problem*, a simple and extensively-studied statistical model for the core task addressed by spectral methods, in the setting of *heavy-tailed noise*, where the above shortcomings of PCA and eigenvector-based methods are readily apparent [Joh01]. We develop and analyze algorithms for this problem whose provable guarantees *improve over traditional eigenvector-based methods*. Our main problem is:

**Problem 1.1** (Generalized spiked Wigner model, recovery)**.** *Given a realization of a symmetric random matrix of the form $Y = \lambda x x^\top + W$, where $x \in \mathbb{R}^n$ is an unknown fixed vector with $\|x\| = \sqrt{n}$, $\lambda > 0$, and the upper triangular off-diagonal entries of $W \in \mathbb{R}^{n \times n}$ are independently (but not necessarily identically) distributed with zero-mean and unit variance $\mathbb{E}\, W_{ij}^2 = 1$, estimate $x$.*

The main question about the spiked Wigner model is: how large should the signal-to-noise ratio $\lambda > 0$ be in order to achieve constant correlation with $x$? The standard algorithmic approach to

---

[*]equal contribution

solve the spiked Wigner recovery problem is PCA, using the top eigenvector of the matrix $Y$ as an estimator for $x$. This approach has been extensively studied (e.g. in [BBAP05, PRS13]), usually under stronger assumptions on the distribution of the entries of $W$.

Assuming boundedness of 5 moments, i.e. $\mathbb{E}|W_{ij}|^5 \leq O(1)$, a clear picture has emerged: the problem is information-theoretically impossible for $\lambda\sqrt{n} < 1$, and for $\lambda\sqrt{n} > 1$ the top eigenvector of $Y$ is an optimal estimator for $x$ – this is the celebrated BBP phase transition [BBAP05, PRS13]. If we weaken the assumption to $\mathbb{E}|W_{ij}|^4 \leq O(1)$, it is well known that $\mathbb{E}\|W\| \leq 2\sqrt{n}$, so PCA will estimate $x$ nontrivially when $\lambda\sqrt{n} > 2$. However, many natural random matrices do not satisfy these conditions – consider for instance random sparse matrices or matrices with heavy-tailed entries.

Our setting allows for much nastier noise distributions: we assume only that the entries of $W$ have unit variance – $\mathbb{E}|W_{ij}|^{2.01}$ may grow arbitrarily fast with $n$, or even fail to exist. Under such weak assumptions, the top eigenvector of $Y$ may be completely uncorrelated with the planted spike $x$, for $\lambda\sqrt{n} = O(1)$. In this paper, we ask:

> *Main Question: For which $\lambda > 0$ is recovery possible in the spiked Wigner model via an efficient algorithm under heavy-tailed noise distributions?*

A natural strategy to deal with heavy-tailed noise is to truncate unusually large entries before performing vanilla PCA. However, truncation-based algorithms can fail dramatically if the distributions of the noise entries are adversarially chosen, as our random matrix model allows. We provide counterexamples to truncation-based algorithms in Section A.3.

## 1.1 Our Contributions

In this work, we develop and analyze computationally-efficient algorithms based on *self-avoiding walks*. PCA or eigenvector methods can be thought of as computing a power $Y^\ell$ of the input matrix, for $\ell \to \infty$. The polynomial $Y^\ell$ in the entries of $Y$ can be expanded in terms of length-$\ell$ walks in the complete graph on $n$ vertices. Our algorithms, by contrast, are based on a different degree-$\ell$ polynomial in the entries of $Y$, which can be expanded in terms of length-$\ell$ self-avoiding walks. We describe the main ideas more thoroughly below, turning for now to our results.

**Spiked Matrices with Heavy-Tailed Noise:** The first result addresses the main question above, demonstrating that our self-avoiding walk algorithm addresses some of the shortcomings of PCA and eigenvector-based methods for the spiked Wigner recovery problem in the heavy-tailed setting.

**Theorem 1.2.** *For every $\delta > 0$, there is a polynomial-time algorithm such that for every $x \in \mathbb{R}^n$ with $\|x\|_2 = \sqrt{n}$ and $\|x\|_\infty \leq n^{1/2-\delta}$ and every $n^{1/2}\lambda \geq 1 + \delta$, given $Y = \lambda xx^\top + W$ distributed as in the spiked Wigner model, the algorithm returns $\hat{x}$ such that $\mathbb{E}\langle \hat{x}, x\rangle^2 \geq \delta^{O(1)} \cdot \|x\|_2^2 \cdot \mathbb{E}\|\hat{x}\|_2^2$.*

To interpret the result, we note that even if the entries of $W$ are Gaussian, when $\lambda\sqrt{n} < 1$ no estimator $\hat{x}$ achieves nontrivial correlation with $x$ [PWBM18], so the assumption $\lambda\sqrt{n} \geq 1 + \delta$ is the weakest one can hope for. Furthermore, under this assumption, when $\delta$ is close to 0, it is information-theoretically impossible to find $\hat{x}$ such that $\langle x, \hat{x}\rangle^2/(\|x\|^2\|\hat{x}\|^2) \to 1$. The guarantee we achieve, that $\hat{x}$ is nontrivially correlated to $x$, is the best one can hope for. (For the regime $\lambda\sqrt{n} \to \infty$, our algorithm does achieve correlation going to 1. Improving the $\delta^{O(1)}$ term to be quantitatively optimal is an interesting open question.)

**Spiked Tensors with Heavy-Tailed Noise:** The self-avoiding walk approach to algorithm design is quite flexible, and in particular is not limited to spiked matrices. We also study an analogous problem for *spiked tensors*. The single-spike tensor model is the analogue of the spiked Wigner model above, but for the task of recovering information from noisy multi-modal data, which has many applications across machine learning [AGH+14, RM14].

**Theorem 1.3.** [2] *For every $c > 0, \delta < 1$ there is a polynomial-time algorithm with the following guarantees. Let $x \in \mathbb{R}^n$ be a random vector with independent, mean-zero entries having $\mathbb{E}\,x_i^2 = 1$ and $\Gamma = \mathbb{E}\,x_i^4 \leq n^{o(1)}$. Let $\lambda > 0$. Let $Y = \lambda \cdot x^{\otimes 3} + W$, where $W \in \mathbb{R}^{n \times n \times n}$ has independent,*

*mean-zero entries with* $\mathbb{E}\, W_{ijk}^2 = 1$. *Then if* $\lambda \geq cn^{-3/4}$, *the algorithm finds* $\hat{x} \in \mathbb{R}^n$ *such that* $\mathbb{E}\langle x, \hat{x}\rangle \geq \delta \cdot (\mathbb{E}\, \|x\|_2^2)^{1/2} \cdot (\mathbb{E}\, \|\hat{x}\|_2^2)^{1/2}$.

Under the additional assumption that for an arbitrarily small $\varepsilon = \Omega(1)$ all entries in $W$ have bounded $12 + \varepsilon$-th moments, a slightly modified algorithm finds $\hat{x}$ such that $\langle x, \hat{x}\rangle \geq (1 - n^{-\Omega(1)})(\mathbb{E}\, \|x\|_2^2)^{1/2} \cdot (\mathbb{E}\, \|\hat{x}\|_2^2)^{1/2}$, as shown in appendix B.6. (We have not made an effort to optimize the constant 12; some improvement may be possible.) The results are stated for order-3 tensors for simplicity; there is no difficulty in extending them to the higher order case. (See appendix C.)

Prior work considers the spiked tensor model only in the case that $W$ has either Gaussian or discrete entries [HSS15, WAM19, BCRT19, Has19, BGL$^+$16a, BGL16b, RRS17], whereas our results make much weaker assumptions, in particular allowing the entries of $W$ to be heavy-tailed. The requirement that $\lambda \geq \Omega(n^{-3/4})$ is widely believed to be necessary for polynomial-time algorithms [HKP$^+$17]. Sub-exponential time algorithms are known recover $x$ successfully for $\lambda \leq n^{-3/4 - \Omega(1)}$ in Gaussian and discrete settings [BGL16b, WAM19, Has19, RRS17] – we show that a sub-exponential time version of our algorithm achieves many of the same guarantees while still allowing for heavy-tailed noise. Concretely, we extend Theorem 1.3 as follows:

**Theorem 1.4.** *In the same setting as theorem 1.3, for any* $c \geq n^{-1/8 + \Omega(1)}$ *and* $\delta < 1$, *when* $\lambda \geq cn^{-3/4}$ *there is an* $n^{O(1/c^4)}$-*time algorithm such that* $\mathbb{E}\langle x, \hat{x}\rangle \geq \delta \cdot (\mathbb{E}\, \|x\|_2^2)^{1/2} \cdot (\mathbb{E}\, \|\hat{x}\|_2^2)^{1/2}$.

In particular, the tradeoff we obtain between running time and signal-to-noise ratio $\lambda$ matches lower bounds in the *low-degree model* for the (easier) case of Gaussian noise [KWB19], for $c \geq n^{-1/8 + o(1)}$.

**Numerical Experiments:** We test our algorithms on synthetic data – random matrices (and tensors) with hundreds of rows and columns – empirically demonstrating the improvement over vanilla PCA.

## 1.2 Our Techniques

We now offer an overview of the self-avoiding walk technique we use to prove Theorems 1.2 and 1.3. For this exposition, we focus on the case of spiked matrices (Theorem 1.2).

Our techniques are inspired by recent literature on *sparse stochastic block models,* in particular the study of nonbacktracking random walks in sparse random graphs [Abb17]. We remark further below on the relationship with this literature, but note for now that a self-avoiding walk algorithm closely related to the one we present here appeared in [HS17] in the context of the sparse stochastic block model with overlapping communities. In the present work we give a refined analysis of this algorithm to obtain Theorem 1.2, and extend the algorithm to spiked tensors to obtain Theorem 1.3.

Recall that given a spiked random matrix $Y = \lambda x x^\top + W$, our goal is to estimate the vector $x$. For simplicity of exposition, we suppose $x \in \{\pm 1\}^n$. To estimate $x$ up to sign, we will in fact aim to estimate each entry of the matrix $xx^\top$. Our starting point is the observation that any sequence $i_0, i_1, \ldots, i_\ell \in [n]$ without repeated indices (i.e. a length-$\ell$ self-avoiding walk in the complete graph on $[n]$) gives an estimator of $x_{i_0} x_{i_\ell}$ as follows:

$$\mathbb{E}_W \prod_{j=0}^{\ell-1} Y_{i_j, i_{j+1}} = \lambda^\ell x_{i_0} x_{i_1}^2 \ldots x_{i_{\ell-1}}^2 x_{i_\ell} = \lambda^\ell x_{i_0} x_{i_\ell}. \tag{1}$$

To aggregate these estimators into a single estimator for $xx^\top$, we relate them to self-avoiding walks in the complete graph on $[n]$. We denote by $\text{SAW}_\ell(i, j)$ the set of length-$\ell$ self-avoiding walks between $i, j$ on the vertex set $[n]$. Then we associate the polynomial $\prod_{j < \ell} Y_{i_j, i_{j+1}}$ to $\alpha = (i_0, i_2, \ldots, i_\ell) \in \text{SAW}_\ell(i, j)$, where $i_0 = i, i_\ell = j$, and we denote this polynomial as $\chi_\alpha(Y)$.

We define the *self-avoiding walk matrix*:

**Definition 1.5** (Self-avoiding walk matrix). Let $P(Y) \in \mathbb{R}^{n \times n}$ be given by

$$P_{ij}(Y) = \sum_{\alpha \in \text{SAW}_\ell(i,j)} \chi_\alpha(Y)$$

Our estimator for $x_i x_j$ will simply be $\frac{P_{ij}(Y)}{\lambda^\ell |\text{SAW}_\ell(i,j)|}$. By (1), $\frac{P_{ij}(Y)}{\lambda^\ell |\text{SAW}_\ell(i,j)|}$ is an unbiased estimator for $x_i x_j$. The crucial step is to bound the variance of $P_{ij}(Y)$. Our key insight is: because we

average only over self-avoiding walks, $P_{ij}(Y)$ is multilinear in the entries of $W$, so $\mathbb{E} P_{ij}^2(Y)$ can be controlled under only the assumption of unit variance for each entry of $W$. Our technical analysis shows that $\mathbb{E} P_{ij}^2(Y)$ is small enough to provide a nontrivial estimator of $x_i x_j$ when (a) $\lambda\sqrt{n} \geq 1 + \delta$ and (b) $\ell \geq O_\delta(\log n)$, for any $\delta > 0$.

**Rounding algorithm:** Once we have $P(Y)$ achieving constant correlation with $xx^\top$, the following theorem, proved in [HS17], gives a polynomial time algorithm for extracting an estimator $\hat{x}$ for $x$.

**Theorem 1.6.** *Let $Y$ be a symmetric random matrix and $x$ a vector. Suppose we have a matrix-valued function $P(Y)$ such that*

$$\frac{\mathbb{E}\langle P(Y), xx^\top\rangle}{\left(\mathbb{E}\|P(Y)\|_F^2 \cdot \|x\|^4\right)^{1/2}} = \delta\,.$$

*then with probability $\delta^{O(1)}$, a random unit vector $\hat{x}$ in the span of top-$\delta^{-O(1)}$ eigenvectors of $P(Y)$ achieves $\langle x, \hat{x}\rangle^2 \geq \delta^{O(1)}\|x\|^2$.*

**Prior-free estimation for general $x$:** A significant innovation of our work over prior work such as [HS17] investigating estimators based on self-avoiding walks is that we avoid the assumption of a prior distribution on the planted vector $x$; instead we assume only a mild bound on the $\ell_\infty$ norm of $x$. While in the setting of Gaussian $W$ one can always assume that $x$ is random by applying a random rotation to the input matrix $Y$ (which preserves $W$ if it is Gaussian), in our setting working with fixed $x$ presents technical challenges.

In the foregoing discussion we assumed $x$ to be $\pm 1$-valued – to drop this assumption, we must forego (1) and give up on the hope that each self-avoiding walk from $i$ to $j$ is an unbiased estimator of $x_i x_j$. Instead, we are able to use the weak $\ell_\infty$ bound to control the bias of an *average* self-avoiding walk as an estimator for $x_i x_j$, and hence control the bias of the estimator $P_{ij}(Y)$. Compared to [HS17], which studies the cases of random or $\pm 1$-valued $x$, our calculation of the variance of $P_{ij}(Y)$ is also significantly more intricate, again because we cannot rely on either randomness or $\pm 1$-ness of $x$.

**Polynomial time via color coding:** The techniques described already yield an algorithm for the spiked Wigner model running in quasipolynomial time $n^{O_\delta(\log n)}$, simply by evaluating all of the self-avoiding walk polynomials. We use the *color coding* technique of [AYZ95] (previously used in the context of the stochastic block model by [HS17]) to improve the running time to $n^{O_\delta(1)}$. Briefly, color coding speeds up the computation of the self-avoiding walk estimators $P_{ij}(Y)$ with a clever combination of randomization and dynamic programming.

**Extension to spiked tensors:** The tensor analogue of the PCA algorithm for spiked matrices is the *tensor unfolding* method, where an $n \times n \times n$ input tensor $Y = \lambda x^{\otimes 3} + W$ is unfolded to an $n^2 \times n$ matrix, and then the top $n$-dimensional singular vector of this matrix is used to estimate $x$. This strategy is successful in the case of Gaussian noise, for $\lambda \gg n^{-3/4}$. To prove Theorem 1.3 we adapt the self-avoiding walk method above to handle this form of rectangular matrix. To prove Theorem 1.4, we combine the self-avoiding walk method with higher-order spectral methods previously used to obtain subexponential time algorithms for the spiked tensor model [RSS18, BGL16b].

**Relationship to PCA and Non-Backtracking Walks** To provide some further context for our techniques, it is helpful to observe the following relationship to PCA. Given a symmetric matrix $Y$, PCA will extract the top eigenvector of $Y$. Often, this is implemented via the power method – that is, PCA will (implicitly) compute the matrix $Y^\ell$ for $\ell \approx \log n$. Notice that the entries of $Y^\ell$ can be expanded as

$$(Y^\ell)_{ij} = \sum_{k_1,\ldots,k_{\ell-1}\in[n]} Y_{i,k_1} \cdot \prod_{a\in[\ell-2]} Y_{k_a,k_{a+1}} \cdot Y_{k_{\ell-1},j}$$

which is a sum over all length-$\ell$ walks from $i$ to $j$ in the complete graph. Our estimator $P(Y)$ can be viewed as removing some problematic (high variance) terms from this sum, leaving only the self-avoiding walks.

This approach is inspired by recent developments in the study of sparse random graphs, where vertices of unusually high degree spoil the spectrum of the adjacency matrix (indeed, this is morally a special case the heavy-tailed noise setting we consider). In particular, inspired by statistical physics, *nonbacktracking* walks were developed as a technique to learn communities in the stochastic block model [MNS18, AS18, SLKZ15, DKMZ11, KMM$^+$13, BLM15]. A $k$-nonbacktracking walk

$i_1, \ldots, i_\ell$ does not repeat any indices $i \in [n]$ among any consecutive $k$ steps; as $k$ increases from $0$ to $\ell$ this interpolates between naïve PCA and our self-avoiding walk estimator.

The $k$-nonbacktracking algorithm for $k < \ell$ is also a natural approach in the setting we study. (Our approach corresponds to $k = \ell$.) Indeed, there are some advantages to choosing $k = O(1)$: the $O(1)$-nonbacktracking-based estimator can be computed much more efficiently than the self-avoiding walk-based estimator. Furthermore, in numerical experiments we observe that even 1-step non-backtracking gives performance comparable with fully self-avoiding walks. However, rigorous analysis of the $O(1)$-non-backtracking walk estimator in our distribution-independent setting appears to be a major technical challenge – even establishing rigorous guarantees in the stochastic block model was a major breakthrough [BLM15, MNS18]. An advantage of our estimator is that it comes with a relatively simple and highly adaptable rigorous analysis.

## 1.3 Organization

In section 2, we discuss algorithms for the spiked matrix model, proving Theorem 1.2. In section 3 we describe our algorithm for the spiked tensor model, deferring the analysis to supplementary material. In section A.4, we provide counter-example to a naïve truncation algorithm. In section A.7 we discuss results of numerical experiments for spiked random matrices.

## 2 Algorithm for general spiked matrix model

Here we prove Theorem 1.2 by analyzing the self-avoiding walk estimator. (Some details are deferred to supplementary material.) We focus for now on the following main lemma, putting together the proof of Theorem 1.2 at the end of this section.

**Lemma 2.1.** *In spiked matrix model $Y = \lambda x x^\top + W$ with $\|x\| = \sqrt{n}$ and the upper triangular entries in symmetric matrix $W$ independently sampled with zero mean and unit variance, we assume $\|x\|_\infty^2 = n^{1-\Omega(1)}$. Then if $\lambda n^{1/2} = 1 + \delta = 1 + \Omega(1)$, setting $\ell = O(\log_{1+\delta} n)$, we have:*

$$\frac{\mathbb{E}\langle P(Y), x x^\top \rangle}{n \left( \mathbb{E}\|P(Y)\|_F^2 \right)^{1/2}} = \delta^{O(1)},$$

*where $P(Y)$ is the length-$\ell$ self-avoiding walk matrix (Definition 1.5).*

For Lemma 2.1, we will repeatedly need the following technical bound, which we prove in Appendix A.1.

**Lemma 2.2.** *Let $V \subseteq [n]$, $\|x\| = \sqrt{n}$ and $t_1, t_2 \in \mathbb{N}$. We define the quantity $S_{t_1,t_2,V}$ as the following:*

$$S_{t_1,t_2,V} = \mathop{\mathbb{E}}_{(v_1,\ldots,v_{t_1+t_2}) \subseteq [n] \setminus V} \left[ \prod_{i=1}^{t_1} x_{v_i}^2 \prod_{i=t_1+1}^{t_1+t_2} x_{v_i}^4 \right]$$

*where $(v_1, v_2, \ldots, v_{t_1+t_2})$ is uniformly sampled from all size-$(t_1 + t_2)$ ordered subsets of $[n] \setminus V$ (without repeating elements). Then assuming $|V|, t_1, t_2 = O(\log n)$ and $\|x\|_\infty^2 = n^{1-\Omega(1)}$, we have $S_{t_1,t_2,V} \le (1 + n^{-\Omega(1)})\|x\|_\infty^{2t_2}$. Further if $t_2 = 0$, we have $S_{t_1,t_2,V} \ge 1 - n^{-\Omega(1)}$.*

From the case $t_2 = 0$, one can easily deduce the following bound on $\mathbb{E}\langle P(Y), x x^\top \rangle$.

**Lemma 2.3.** *Under the same setting as lemma 2.1, we have $\mathbb{E}\langle P(Y), x x^\top \rangle = (1 \pm o(1))\lambda^\ell n^{\ell+1}$*

*Proof.* We have

$$\mathbb{E}P_{ij}(Y) = \sum_{\alpha \in \mathrm{SAW}_\ell(i,j)} \prod_{t=1}^{\ell-1} \lambda x_{\alpha_t} x_{\alpha_{t+1}}$$

$$= \lambda^\ell \frac{(n-2)!}{(n-(\ell-1))!} x_i x_j \mathop{\mathbb{E}}_{\alpha \in \mathrm{SAW}_\ell(i,j)} \left[ \prod_{t=1}^{\ell-1} x_{\alpha_t}^2 \right]$$

$$= (1 + n^{-\Omega(1)})\lambda^\ell x_i x_j n^{\ell-1} \mathop{\mathbb{E}}_{\alpha \in \mathrm{SAW}_\ell(i,j)} \left[ \prod_{t=1}^{\ell-1} x_{\alpha_t}^2 \right],$$

where the expectation is taken uniformly over $\alpha \in \text{SAW}_\ell(i,j)$. For simplicity of notation, we denote $\mathbb{E}_{\alpha \in \text{SAW}_\ell(i,j)} \prod_{t=1}^{\ell-1} x_{\alpha_t}^2$ as $S_{ij}$. Then according to lemma 2.2, we have $S_{ij} = 1 \pm o(1)$. Therefore we have $\langle P(Y), xx^\top \rangle = (1 \pm o(1))\lambda^\ell n^{\ell+1}$. $\qquad\square$

To prove Lemma 2.1, the remaining task is to bound the second moment $\mathbb{E}\|P(Y)\|_F^2$. We can expand the second moment in terms of pairs of self-avoiding walks. For $\alpha, \beta \in \text{SAW}_\ell(i,j)$ and corresponding polynomials $\chi_\alpha(Y), \chi_\beta(Y)$, there is a close relationship between $\mathbb{E}[\chi_\alpha(Y)\chi_\beta(Y)]$ and the number of shared vertices and edges of $\alpha, \beta$. Specifically,

$$
\begin{aligned}
\mathbb{E}[\chi_\alpha(Y)\chi_\beta(Y)] &= \mathbb{E}\left[ \prod_{(u,v)\in\alpha\cap\beta} Y_{uv}^2 \prod_{(u,v)\in\alpha\Delta\beta} Y_{uv} \right] \\
&= \prod_{(u,v)\in\alpha\Delta\beta} \lambda x_u x_v \prod_{(u,v)\in\alpha\cap\beta} \left(1 + \lambda^2 x_u^2 x_v^2\right) \\
&= \lambda^{2\ell-2k} \prod_{u\in\deg(\alpha\Delta\beta,2)} x_u^2 \prod_{u\in\deg(\alpha\Delta\beta,4)} x_u^4 \prod_{(u,v)\in\alpha\cap\beta} \left(1 + \lambda^2 x_u^2 x_v^2\right)
\end{aligned}
$$

where $k$ is number of shared edges between $\alpha, \beta$ and $\deg(\alpha\Delta\beta, j)$ is the set of vertices with degree $j$ in the graph $\alpha\Delta\beta$. The size of $\deg(\alpha\Delta\beta, 4)$ is equal to the number of shared vertices which are not incident to any shared edge. Thus for the analysis of $\mathbb{E}\,P_{ij}^2(Y) = \sum_{\alpha,\beta\in\text{SAW}_\ell(i,j)} \mathbb{E}[\chi_\alpha(Y)\chi_\beta(Y)]$, we classify pairs $\alpha, \beta \in \text{SAW}_\ell(i,j)$ according to numbers of shared edges and vertices between $\alpha, \beta$. The following graph-theoretic lemma is needed for bounding the number of such pairs in each class; we will prove it in appendix A.1.

**Lemma 2.4.** *Let $\alpha = \alpha_0, \alpha_1, \ldots, \alpha_\ell$ and $\beta = \beta_0, \beta_1, \ldots, \beta_\ell$ be two length-$\ell$ self-avoiding walks in the complete graph on $[n]$, with $\alpha_0 = \beta_0 = i$ and $\alpha_\ell = \beta_\ell = j$. Let $k$ be the number of shared edges between $\alpha, \beta$, $r$ be the number of shared vertices between $\alpha, \beta$ excluding $i, j$, and $s$ be the number of shared vertices which are not $i, j$ and not incident to shared edges. Further we denote the number of connected components in $\alpha \cap \beta$ not containing $i, j$ as $p$. Then for $\alpha \neq \beta$ we have the relation $p \leq r - s - k$, and for $\alpha = \beta$ we have $p = s = 0$ and $r = k - 1$.*

We note that for self-avoiding walks $\alpha, \beta$, the connected components of $\alpha \cap \beta$ are all self-avoiding walks. A simple corollary of lemma 2.2 turns out to be helpful, which we prove in appendix A.1

**Lemma 2.5.** *Suppose we have $x \in \mathbb{R}^n$ with norm $\sqrt{n}$. If $V \subseteq [n]$ has $|V| = O(\log n)$ and if we average over size-$h$ directed self-avoiding walks $\xi$ on vertices $[n] \setminus V$, then for $h = O(\log n)$ we have the bounds*

$$
\mathop{\mathbb{E}}_{\xi\subseteq[n]\setminus V} \left[ x_{\xi_0}^2 x_{\xi_h}^2 \prod_{(u,v)\in\xi} (1 + \lambda^2 x_u^2 x_v^2) \right] \leq (1 + n^{-\Omega(1)}) \|x\|_\infty^2 (1 + \lambda^2 \|x\|_\infty^2)^h
$$

$$
\mathop{\mathbb{E}}_{\xi\subseteq[n]\setminus V} \left[ x_{\xi_h}^2 \prod_{(u,v)\in\xi} (1 + \lambda^2 x_u^2 x_v^2) \right] \leq (1 + n^{-\Omega(1)})(1 + \lambda^2 \|x\|_\infty^2)^h
$$

$$
\mathop{\mathbb{E}}_{\xi\subseteq[n]\setminus V} \left[ x_{\xi_0}^2 \prod_{(u,v)\in\xi} (1 + \lambda^2 x_u^2 x_v^2) \right] \leq (1 + n^{-\Omega(1)})(1 + \lambda^2 \|x\|_\infty^2)^h
$$

$$
\mathop{\mathbb{E}}_{\xi\subseteq[n]\setminus V} \left[ \prod_{(u,v)\in\xi} (1 + \lambda^2 x_u^2 x_v^2) \right] \leq (1 + n^{-\Omega(1)})(1 + \lambda^2 \|x\|_\infty^2)^h
$$

*where $\xi_0$ is the label of starting vertex of $\xi$, and $\xi_h$ is the label of the end vertex of $\xi$.*

These bounds hold since we can expand the product into a sum of monomials and apply lemma 2.2 for each monomial.

Now for self-avoiding walk pairs $(\alpha, \beta)$ intersecting on a given number of edges and vertices, we bound the correlation of corresponding polynomials and hence the contribution to the variance of $P$. For simple expressions, we take $\lambda = O(n^{-1/2})$.

**Definition 2.6.** On the complete graph $K_n$, for pairs of self-avoiding walks $(\alpha, \beta)$ and $(\gamma, \xi)$, we say that $(\alpha, \beta)$ is isomorphic to $(\gamma, \xi)$ if there is a permutation $\pi : [n] \to [n]$ fixing $i, j$ such that $\pi(\alpha) = \gamma$ and $\pi(\beta) = \xi$. We partition all pairs of length-$\ell$ self-avoiding walks between vertices $i, j$ into isomorphism classes. We denote the set of all isomorphism classes containing pairs length-$\ell$ self-avoiding walks between vertices $i, j$ sharing $r$ vertices and $k$ edges as shape$(k, r, i, j)$.

We note that $r < k$ is only possible when $r + 1 = k = \ell$ (that is, the two paths are identical). We prove the following lemma in A.1.

**Lemma 2.7** (Self-avoiding walk polynomial correlation)**.** *In the spiked Wigner model $Y = \lambda x x^\top + W$, where $x$ has norm $\sqrt{n}$ and $W$ is symmetric with entries independently sampled with zero mean and unit variance, for any isomorphism class $\mathcal{S} \in$ shape$(k, r, i, j)$, we have*

$$\mathop{\mathbb{E}}_{(\alpha,\beta)\sim\mathcal{S}} \mathop{\mathbb{E}}_{W} [\chi_\alpha(Y)\chi_\beta(Y)] \leq \begin{cases} (1 + n^{-\Omega(1)})\lambda^{2\ell - 2k}\|x\|_\infty^{2(r-k)} \left(1 + \lambda^2 \|x\|_\infty^2\right)^k & \text{if } r \geq k \\ (1 + n^{-\Omega(1)}) \left(1 + \lambda^2 \|x\|_\infty^2\right)^k & \text{if } r + 1 = k = \ell \end{cases}$$

*where $(\alpha, \beta) \sim \mathcal{S}$ is taken uniformly over the isomorphism class $\mathcal{S}$ and $\chi_\alpha(Y) = \prod_{(u,v)\in\alpha} Y_{u,v}$.*

Now we finish the proof of lemma 2.1.

*Proof of Lemma 2.1.* We bound the variance of the estimator $P_{ij}(Y)$. As stated above,

$$\mathbb{E}\, P_{ij}^2(Y) = \sum_{\alpha,\beta\in\text{SAW}_\ell(i,j)} \lambda^{2\ell-2k} \prod_{u\in\deg(\alpha\Delta\beta,2)} x_u^2 \prod_{u\in\deg(\alpha\Delta\beta,4)} x_u^4 \prod_{(u,v)\in\alpha\cap\beta} (1 + \lambda^2 x_u^2 x_v^2), \quad (2)$$

where $k$ is number of shared edges between $\alpha, \beta$ and $\deg(\alpha\Delta\beta, j)$ is the set of vertices with degree $j$ in the graph $\alpha\Delta\beta$.

We note that for fixed $i, j, r, k$ there are at most $n^{2(\ell-1)-r}\ell^{O(r-k)}$ pairs of $\alpha, \beta$. For fixed $k, r$, we apply lemma 2.7. For $k < r$ the contribution to summation 2 is bounded by

$$n^{2(\ell-1)-r}\ell^{O(r-k)} \mathop{\mathbb{E}}_{\mathcal{S}\sim\text{shape}(k,r,i,j)} \left[\lambda^{2\ell-2k}\|x\|_\infty^{2(r-k)} \left(1 + \lambda^2\|x\|_\infty^2\right)^k\right]$$

$$= n^{-2} \cdot n^{2\ell} \cdot \lambda^{2\ell} \cdot n^{-r}\ell^{O(r-k)}\lambda^{-2k}\|x\|_\infty^{2(r-k)} \left(1 + \lambda^2\|x\|_\infty^2\right)^k$$

where $\mathcal{S}$ is sampled with some distribution over all shapes in shape$(k, r, i, j)$.

For $k = r + 1 = \ell$, if we take $\ell = C\log_{\lambda^2 n} n$ with constant $C$ large enough, then the contribution to summation 2 is bounded by

$$n^{\ell-1} \left(1 + \lambda^2\|x\|_\infty^2\right)^\ell \leq n^{-\Omega(1)} n^{2(\ell-1)}\lambda^{2\ell}$$

Combining all possible $k, r$, we have summation 2 bounded by

$$n^{2(\ell-1)}\lambda^{2\ell} \left[n^{-\Omega(1)} + \sum_{k=0}^{\ell-1} \left(n^{-k}\lambda^{-2k} \left(1 + \lambda^2\|x\|_\infty^2\right)^k\right) \sum_{r=k}^{\ell-1} \left(\ell^{O(r-k)}\|x\|_\infty^{2(r-k)} n^{k-r}\right)\right]$$

Since $\lambda\|x\|_\infty = n^{-\Omega(1)}$, we have $n^{-1}\lambda^{-2} \left(1 + \lambda^2\|x\|_\infty^2\right)^k \leq \frac{1}{1-\delta/2}$. Thus $\sum_{k=0}^{\ell-1} \left(n^{-k}\lambda^{-2k} \left(1 + \lambda^2\|x\|_\infty^2\right)^k\right) \leq \delta^{-O(1)}$. On the other hand, since $\ell^{O(1)}\|x\|_\infty^2 n^{-1} = n^{-\Omega(1)}$ by the assumption on $\|x\|_\infty$, we have $\sum_{r=k}^{\ell-1} \left(\ell^{O(r-k)}\|x\|_\infty^{2(r-k)} n^{k-r}\right) \leq 1 + n^{-\Omega(1)}$. Thus summation 2 is bounded by $\delta^{-O(1)}\lambda^{2\ell}n^{2\ell-2}$.

Summing over $n^2$ pairs of $i, j$ we have

$$\mathbb{E}\|P(Y)\|_F^2 \leq \delta^{-O(1)} \left(n^{2\ell}\lambda^{2\ell}\right)$$

Combining with lemma 2.3 we have $\frac{\mathbb{E}\langle P(Y), xx^\top\rangle}{n\left(\mathbb{E}\|P(Y)\|_F^2\right)^{1/2}} = \delta^{O(1)} = \Omega(1)$ and the lemma follows. $\square$

Finally, using color-coding method, the degree $O(\log n)$ polynomial $P(Y)$ can be well approximated in polynomial time, which we prove in appendix A.2

**Lemma 2.8** (Formally stated in appendix A.2). *For $\delta = \lambda n^{1/2} - 1 > 0$ and $\ell = O(\log_{1+\delta} n)$, $P(Y)$ can be accurately evaluated in $n^{\delta^{-O(1)}}$ time.*

The evaluation algorithm 1 is based on the idea of color-coding method[AYZ95]. Similar algorithm has already appeared and analyzed in the literature [HS17].

---

**Algorithm 1:** Algorithm for evaluating self-avoiding walk matrix

---

**Data:** Given $Y \in \mathbb{R}^{n \times n}$ s.t $Y = \lambda x x^\top + W$

**Result:** $P(Y) \in \mathbb{R}^{n \times n}$ where $P_{ij}(Y)$ is the sum of multilinear monomials corresponding to length $\ell$ self-avoiding walk between $i, j$(up to accuracy $1 + n^{-\Omega(1)}$)

$C \leftarrow \exp(100\ell)$;

**for** $i \leftarrow 1$ **to** $C$ **do**

    Sample coloring $c_t : [n] \mapsto [\ell + 1]$ uniformly at random;

    Construct a $\mathbb{R}^{2^{\ell+1} n \times 2^{\ell+1} n}$ matrix $M$, with rows and columns indexed by $(v, S)$, where $v \in [n]$ and $S$ is a subset of $[\ell + 1]$;

    a matrix $H \in \mathbb{R}^{n \times 2^{\ell+1} n}$ with rows indexed by $[n]$ and columns indexed by $(v, S)$ where $v \in [n]$ and $S$ is a subset of $[\ell + 1]$;

    a matrix $N \in \mathbb{R}^{2^{\ell+1} n \times n}$, with rows indexed by $(v, S)$ where $S$ is a subset of $[\ell + 1]$ and columns indexed by $[n]$;

    Record matrix $p_{c_i} = H M^{\ell-2} N$;

Return $\sum_{i=1}^{C} p_{c_i}/C$

---

We describe how to construct matrices $H, M, N$ used in the algorithm 1 given coloring $c : [n] \mapsto [\ell+1]$. For matrix $M$, the entry $M_{(v_1, S),(v_2, T)} = Y_{v_1, v_2}$ if $S \cup \{c(v_1)\} = T$ and $c(v_1) \notin S$. Otherwise $M_{(v_1, S),(v_2, T)} = 0$. For matrix $H$, the entry $H_{v_1,(v_2, S)} = Y_{v_1, v_2}$ if $S = \{c(v_1)\}$. Otherwise $H_{v_1,(v_2, S)} = 0$. For matrix $N$, the entry $N_{(v_1, S), v_2} = Y_{v_1, v_2}$ if $c(v_1) \notin S$ and $S \cup \{c(v_1), c(v_2)\} = [\ell + 1]$. Otherwise $N_{(v_1, S), v_2} = 0$.

The critical observation is that for coloring $c : [n] \mapsto [\ell + 1]$ sampled uniformly at random and $i \neq j$, we have $\frac{(\ell+1)^{\ell+1}}{(\ell+1)!} \mathbb{E}_c p_{c,i,j}(Y) = P_{i,j}(Y)$. By averaging over lots of such random colorings, we have an unbiased estimator with low variance. The proof is deferred to appendix A.2.

Combining theorem 1.6, lemma 2.1,2.8, we have theorem 1.2.

## 3 Algorithms for general spiked tensor model

For proving theorem 1.3, we use the sum of multilinear polynomials corresponding to a variant of self-avoiding walk. Here we only describe a simple special case of the algorithm, which provides estimation guarantee when $\lambda > n^{-3/4}$.

**Definition 3.1** (Polynomial time estimator for spiked tensor recovery). Given tensor $Y \in \mathbb{R}^{n \times n \times n}$, we have estimator $P(Y) \in \mathbb{R}^n$ where each entry is degree $2\ell - 1$ polynomial given by $P_i(Y) = \sum_{\alpha \in S_{\ell,i}} \chi_\alpha(Y)$, where $\chi_\alpha(Y)$ is multilinear polynomial basis $\chi_\alpha(Y) = \prod_{(i,j,k) \in \alpha} Y_{ijk}$ and $S_{\ell,i}$ is the set of directed hypergraph associated with vertex $i$ generated in the following way:

- We construct $2\ell$ levels of distinct vertices. Level $0$ is vertex $i$. For $0 < t < \ell$, level $2t$ contains 1 vertex and level $2t - 1$ contains 2 vertices. Level $2\ell - 1$ contains 1 vertex.

- We connect a hyperedge between adjacent levels $t - 1, t$ for $t \in [2\ell - 2]$. Each hyperedge directs from level $t - 1$ to level $t$.

- For vertex $u$ which lies in level $2\ell - 2$ and vertices $v, v'$ which lie in level $2\ell - 1$, we add the hyperedge $(u, v, v')$.

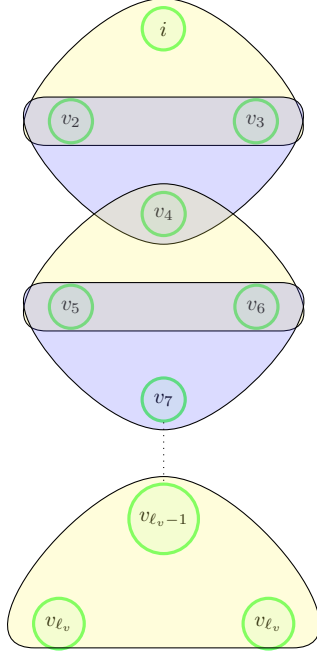

Figure 1: Illustration for a self-avoiding walk $\alpha \in S_{\ell,i}$ for tensor estimation. Each colored area corresponds to a hyperedge.

An illustration of a self-avoiding walk $\alpha \in S_{\ell,i}$ is given by figure 1.

In appendix B.4 we show that by introducing width to levels, estimation under smaller SNR $\lambda$ is possible by exploiting more computational power. In appendix B.5 we show that $P(Y)$ can be evaluated in $n^{O(v)}$ time using color coding method. These lead to the proofs of theorems 1.3, 1.4.

## 4 Conclusion

We provide an algorithm which nontrivially estimates rank-one spikes of Wigner matrices for signal-to-noise ratios $\lambda$ approaching the sharp threshold $\lambda\sqrt{n} \to 1$, even in the setting of heavy-tailed noise (having only 2 finite moments) with unknown, adversarially-chosen distribution. For future work, it would be intriguing to obtain strengthened guarantees along (at least) two directions.

First, [PWBM18, MRY18] give algorithms which recover rank-one spikes for even smaller values of $\lambda$, when a large constant number of moments of the entries of $W$ are $O(1)$. Relaxing the bounded moment assumptions while keeping $\lambda\sqrt{n} \ll 1$ would be very interesting.

In a different direction, our experiments suggest that an estimator based on non-backtracking walk performs as well as the self-avoiding walk estimator which we are able to analyze rigorously. Rigorously establishing similar guarantees for the non-backtracking walk estimator – or finding counterexamples – would be of great interest.

## 5 Acknowledgement

S.B.H. is supported by a Miller Postdoctoral Fellowship. J.D and D.S are supported by ERC consolidator grant.

## Broader impact

We study an idealized mathematical model and provide theoretical analyses with only synthetic experimental validation. Therefore a direct impact on the society is out of reach. Potentially, since the problem we consider is closely related to stochastic block model, we speculate that our algorithm

could inspire the development of more robust and universal social network analysis. Further there is a chance that color coding method could point out new directions for sum-product evaluation in some types of neural network.

Since the application area of the algorithm is still not clear, the positive and negative sides of algorithm really depend on what people are using it for.

## Footnotes

[2]The theorem is stated for planted vectors sampled from independent zero mean prior distribution; for fixed planted vector, similar guarantees can be obtained using nearly the same techniques as in the spiked matrix model

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
