[Supplementary Material]

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

# A  Spiked matrix model

## A.1  Proof of lemma 2.2, lemma 2.4, lemma 2.5 and lemma 2.7

We first prove lemma 2.2

*Proof of Lemma 2.2.* First since the bound on infinity norm of $x$, we have

$$S_{t_1,t_2,V} \leq \|x\|_\infty^{2t_2} \mathbb{E}_{(v_1,\ldots,v_{t_1+t_2})\subseteq[n]\setminus V} \left[ \prod_{i=1}^{t_1+t_2} x_{v_i}^2 \right]$$

Denote $t_1 + t_2 = t$, now we take marginal on $x_{v_t}$. Then the marginal is given by

$$\mathbb{E}_{(v_1,\ldots,v_{t_1+t_2})\subseteq[n]\setminus V} \left[ \prod_{i=1}^{t_1+t_2} x_{v_i}^2 \right]$$

$$= \mathbb{E}_{(v_1,\ldots,v_{t-1})\subseteq[n]\setminus V} \left[ \prod_{i=1}^{t-1} x_{v_i}^2 \left( \frac{n - \sum_{i=1}^{t-1} x_{v_i}^2 - \sum_{i\in V} x_i^2}{n - |V| - (t-1)} \right) \right]$$

$$= \left( 1 \pm O\left( \frac{(|V| + t)\|x\|_\infty^2}{n} \right) \right) \mathbb{E}_{(v_1,\ldots,v_{t-1})\subseteq[n]\setminus V} \left[ \prod_{i=1}^{t-1} x_{v_i}^2 \right]$$

By induction this is $(1 \pm n^{-\Omega(1)})^t$. In all we have $S_{t_1,t_2,V}$ bounded by $(1 + o(1))\|x\|_\infty^{2t_2}$.  □

*Proof of Lemma 2.4.* For self-avoiding walks $\alpha, \beta \in \mathrm{SAW}(i,j)$, the connected components of $\alpha \cap \beta$ are all self-avoiding walks. We consider quantity $g = r - k$. Then each of the $p$ connected components in $\alpha \cap \beta$ not containing $i$ or $j$ contributes 1 to $g$. Since $\alpha \neq \beta$, other connected components in $\alpha \cap \beta$ can only contain one of $i, j$. Such connected components contribute 0 to $g$. Further each shared vertex not incident to any shared edge contribute 1 to $g$ and the total number of such vertices is given by $s$. Therefore we have $r - k = p + s$  □

*Proof of Lemma 2.5.* We prove the first bound. We represent $\xi$ as an ordered set of vertices $(\xi_0, \xi_1, \ldots, \xi_h)$ then we note that the product in the expectation can be expanded to the sum of monomials:

$$\mathbb{E}_{\xi\subseteq[n]\setminus V} \left[ x_{\xi_0}^2 x_{\xi_h}^2 \prod_{i\in[h]} (1 + \lambda^2 x_{\xi_i}^2 x_{\xi_{i-1}}^2) \right] = \mathbb{E}_{\xi\subseteq[n]\ V} \left[ \sum_{S\subseteq[h]} x_{\xi_0}^2 x_{\xi_h}^2 \prod_{i\in S} \lambda^2 x_{\xi_i}^2 x_{\xi_{i-1}}^2 \right]$$

Since for fixed set $S \subseteq [h]$, the number of variables $x_u$ with degree 4 in the monomial is bounded by $|S|+1$, by lemma 2.2,we have

$$\mathbb{E}_{\xi_0,\xi_1,\ldots,\xi_h} \left[ x_{\xi_0}^2 x_{\xi_h}^2 \prod_{i\in S} (\lambda^2 x_{\xi_i}^2 x_{\xi_{i-1}}^2) \right] \leq (1 + n^{-\Omega(1)})\|x\|_\infty^2 (\lambda\|x\|_\infty)^{2|S|}$$

On the other hand we have

$$(1 + \lambda^2 \|x\|_\infty^2)^h = \sum_{S\subseteq[h]} \left( \lambda^2 \|x\|_\infty^2 \right)^{|S|}$$

Therefore we have

$$\mathbb{E}_{\xi\subseteq[n]\setminus V} \left[ x_{\xi_0}^2 x_{\xi_h}^2 \prod_{i\in[h]} (1 + \lambda^2 x_{\xi_i}^2 x_{\xi_{i-1}}^2) \right] \leq (1 + n^{-\Omega(1)})\|x\|_\infty^2 (1 + \lambda^2 \|x\|_\infty^2)^h$$

The other three bounds can be proved in very similar ways.  □

*Proof of lemma 2.7.* We first consider the case $r \geq k$, where $\alpha \neq \beta$. For each $\alpha, \beta$ intersecting on $k$ edges and $r$ vertices, we have the following bound:

$$\mathbb{E}[\chi_\alpha(Y)\chi_\beta(Y)] = \lambda^{2\ell-2k} \prod_{u \in \deg(\alpha\Delta\beta,2)} x_u^2 \prod_{u \in \deg(\alpha\Delta\beta,4)} x_u^4 \prod_{(u,v) \in \alpha \cap \beta} (1 + \lambda^2 x_u^2 x_v^2)$$

where $\deg(\alpha\Delta\beta, j)$ is the set of vertices with degree $j$ in the graph $\alpha\Delta\beta$.

For any subgraph $G$ of $K_n$, we denote by $V(G)$ the set of vertices incident to edges in $G$. We denote $|\deg(\alpha\Delta\beta, 4)|$ as $s$ and the number of shared vertices between $\alpha, \beta$ excluding $i, j$ as $r$. We denote the number of connected components in $\alpha \cap \beta$ not containing $i, j$ as $p$. Then we have the relation $p \leq r - s - k$ for $\alpha \neq \beta$ according to lemma 2.4. We note that $\alpha \cap \beta$ can be decomposed into a set of disjoint self-avoiding walks, which we denote as $\text{SAW}(\alpha \cap \beta)$.

Now we take the expectation over $(\alpha, \beta)$ on isomorphism class $\mathcal{S}$. This is equivalent to taking uniform expectation over the labeling of the $2(\ell - 1) - r$ vertices $\alpha, \beta$ which are not equal to $i$ or $j$. Then we have

$$\mathbb{E}_{(v_1,v_2,\dots v_{2(\ell-1)-r})} \left[ \prod_{u \in \deg(\alpha\Delta\beta,2)} x_u^2 \prod_{u \in \deg(\alpha\Delta\beta,4)} x_u^4 \prod_{\xi \in \text{SAW}(\alpha\cap\beta)} \left( \prod_{(u,v) \in \xi} (1 + \lambda x_u^2 x_v^2) \right) \right]$$

$$\leq (1 + n^{-\Omega(1)}) \|x\|_\infty^{2s} \mathbb{E}_{(v_1,v_2,\dots v_{2(\ell-1)-r})} \left[ \prod_{\substack{u \in V(\alpha\cap\beta) \\ u \in V(\alpha\Delta\beta)}} x_u^2 \prod_{\xi \in \text{SAW}(\alpha\cap\beta)} \left( \prod_{(u,v) \in \xi} (1 + \lambda x_u^2 x_v^2) \right) \right]$$

$$\leq (1 + n^{-\Omega(1)}) \|x\|_\infty^{2p+2s} \left(1 + \lambda^2 \|x\|_\infty^2\right)^k \leq (1 + n^{-\Omega(1)}) \|x\|_\infty^{2(r-k)} \left(1 + \lambda^2 \|x\|_\infty^2\right)^k$$

where we use lemma 2.2 in the first inequality, lemma 2.5 in the second inequality, and lemma 2.4 in the last inequality. This proves the first claim.

For any isomorphism class $\mathcal{S} \in \text{shape}(k, r, i, j)$ with $k = r + 1 = \ell$, and $(\alpha, \beta) \in \mathcal{S}$, we have $\alpha = \beta$. In this case we have

$$\mathbb{E}_W [\chi_\alpha(Y)\chi_\beta(Y)] = \prod_{(u,v) \in \alpha} (1 + \lambda^2 x_u^2 x_v^2) = \prod_{i \in [\ell]} \left(1 + \lambda^2 x_{v_{i-1}}^2 x_{v_i}^2\right)$$

By lemma 2.5, taking expectation over the labeling of the $\ell - 1$ vertices in $\alpha$ which are not equal to $i, j$, we have

$$\mathbb{E}_{v_1,v_2,\dots,v_{\ell-1}} \prod_{i \in [\ell]} \left(1 + \lambda^2 x_{v_{i-1}}^2 x_{v_i}^2\right) \leq \left(1 + n^{-\Omega(1)}\right) \left(1 + \lambda \|x\|_\infty^2\right)^\ell$$

This proves the second claim. $\square$

## A.2 Evaluation of self-avoiding walk estimator

In spiked matrix model $Y = \lambda xx^\top + W$, we denote $\delta = n^{1/2}\lambda - 1$. For evaluation of degree $\ell = O(\log_{1+\delta} n)$ self-avoiding walk polynomial:

$$P_{ij}(Y) = \sum_{\alpha \in \text{SAW}_\ell(i,j)} \prod_{(u,v) \in \alpha} Y_{uv}$$

we use color coding strategy pretty similar to the literature in [HS17]. The algorithm and construction of matrices has already been described in the main body. We restate the algorithm 2 for readers' convenience.

On complete graph $K_n$, for a specific coloring $c \in [n] \mapsto [\ell + 1]$, we say that a length-$\ell$ self-avoiding walk $\alpha = (v_0, v_1, \dots, v_\ell)$ is colorful if the colors of $v_0, v_1, \dots, v_\ell$ are different. Then a critical observation is that $p_{c,i,j}(Y) = \sum_{\alpha \in \text{SAW}_\ell(i,j)} F_{c,\alpha}\chi_\alpha(Y)$, where $F_{c,\alpha}$ is the 0-1 indicator of random event that $\alpha$ is colorful. Taking uniform expectation on $c$ over all random colorings, we have the following relation:

---

**Algorithm 2:** Algorithm for evaluating self-avoiding walk matrix

---

**Data:** Given $Y \in \mathbb{R}^{n \times n}$ s.t $Y = \lambda x x^\top + W$

**Result:** $P(Y) \in \mathbb{R}^{n \times n}$ where $P_{ij}(Y)$ is the sum of multilinear monomials corresponding to length $\ell$ self-avoiding walk between $i,j$(up to accuracy $1 + n^{-\Omega(1)}$)

$C = \exp(100\ell)$;

**for** $i \leftarrow 1$ **to** $C$ **do**

    Sample coloring $c_t : [n] \mapsto [\ell+1]$ uniformly at random;

    Construct a $\mathbb{R}^{2^{\ell+1}n \times 2^{\ell+1}n}$ matrix $M$, where rows and columns are indexed by $(v, S)$, where $v \in [n]$ and $S$ is a subset of $[\ell+1]$;

    a matrix $H \in \mathbb{R}^{n \times 2^{\ell+1}n}$ where each row is indexed by $[n]$ and each column is indexed by $(v, S)$ where $v \in [n]$ and $S$ is a subset of $[\ell+1]$;

    a matrix $N \in \mathbb{R}^{2^{\ell+1}n \times n}$, where each row is indexed by $(v, S)$ where $S$ is a subset of $[\ell+1]$ and each column is indexed by $[n]$;

    Record matrix $p_{c_i} = HM^{\ell-2}N$;

Return $\sum_{i=1}^{C} p_{c_i}/C$

---

**Lemma A.1.** *In the algorithm 1, for $C \geq \exp(100\ell)$, we have*

$$
\mathbb{E}_{Y,c_1,\ldots,c_C} \left[ \left( \frac{1}{C} \sum_t p_{c_t,i,j}(Y) - \mathbb{E}_c p_{c,i,j}(Y) \right)^2 \right] \leq \exp(-O(\ell)) \mathbb{E}_Y \left( \mathbb{E}_c P_{c,i,j}(Y) \right)^2
$$

*where random colorings $c_1, c_2, \ldots, c_C \in [n] \mapsto [\ell+1]$ are independently sampled uniformly at random and the expectation of random coloring $c \in [n] \mapsto [\ell+1]$ on right hand side is taken uniformly at random.*

*Proof.* For a fixed path $\alpha$, the probability that $F_{c,\alpha} = 1$ is bounded by $\frac{(\ell+1)!}{(\ell+1)^{\ell+1}} \geq \exp(-O(\ell))$. Therefore we have

$$
\mathbb{E}\left[ p_{c,i,j}^2(Y) \right] = \mathbb{E} \left[ \sum_{\alpha,\beta \in \mathrm{SAW}_\ell(i,j)} \chi_\alpha(Y) \chi_\beta(Y) F_{c,\alpha} F_{c,\beta} \right] \tag{3a}
$$

$$
\leq \mathbb{E}_Y \left[ \sum_{\alpha,\beta \in \mathrm{SAW}_\ell(i,j)} [\chi_\alpha(Y) \chi_\beta(Y)] \right] \tag{3b}
$$

$$
\leq \exp(O(\ell)) \mathbb{E}_Y \left[ \left( \sum_{\alpha \in \mathrm{SAW}_\ell(i,j)} \mathbb{E}_c [\chi_\alpha(Y) F_{c,\alpha}] \right)^2 \right] \tag{3c}
$$

$$
= \exp(O(\ell)) \mathbb{E}_Y \left[ \left( \mathbb{E}_c [p_{c,i,j}(Y)] \right)^2 \right] \tag{3d}
$$

For step (3b) and (3c), we use the fact that $0 \leq F_{c,\alpha} \leq 1$ and $\mathbb{E}[\chi_\alpha(Y)\chi_\beta(Y)] \geq 0$ for all $\alpha, \beta \in \mathrm{SAW}_\ell(i,j)$. For step (3c), we also use the fact that $\mathbb{E} F_{c,\alpha} \geq \exp(-O(\ell))$. Therefore

$$
\mathbb{E}_Y \mathbb{E}_c \left[ \left( p_{c,i,j}(Y) - \mathbb{E}_c p_{c,i,j}(Y) \right)^2 \right] \leq \exp(O(\ell)) \mathbb{E}_Y \left[ \left( \mathbb{E}_c p_{c,i,j}(Y) \right)^2 \right]
$$

By averaging $p_{c,i,j}(Y)$ for $C$ independent random colorings, the variance is reduced and we have

$$
\mathbb{E}_{Y,c_1,\ldots,c_C} \left[ \left( \frac{1}{C} \sum_{t,i,j} p_{c_t,i,j}(Y) - \mathbb{E}_c p_{c,i,j}(Y) \right)^2 \right] \leq \frac{1}{C} \exp(O(\ell)) \mathbb{E}_Y \left( \mathbb{E}_c P_{c,i,j}(Y) \right)^2
$$

Therefore let $C = \exp(100\ell)$, the lemma is proved. $\qquad\square$

This lemma implies that the average of $p_c(Y)$ for $n^{\delta^{-O(1)}}$ independent random colorings $p(Y)$ gives accurate approximation of $P(Y)$. The following simple corollary implies that the this matrix $p(Y)$ achieves the same correlation with $xx^\top$ as $P(Y)$.

**Lemma A.2** (Formal statement of Lemma 2.8)**.** *The algorithm 2 runs in $n^{\delta^{-O(1)}}$ time when $\ell = O(\log_{1+\delta} n)$. For matrix returned by algorithm 2, we have*

$$\mathop{\mathbb{E}}_{c_1,\ldots,c_C}\left[\frac{1}{C}\sum_t p_{c_t,i,j}(Y)\right] = \frac{(\ell+1)!}{(\ell+1)^{\ell+1}}P_{ij}(Y)$$

$$\mathop{\mathbb{E}}_{Y,c_1,\ldots,c_C}\left[\left(\frac{1}{C}\sum_t p_{c_t,i,j}(Y)\right)^2\right] \le \left(1+n^{-\Omega(1)}\right)\mathop{\mathbb{E}}_{Y}\left[\left(\frac{(\ell+1)!}{(\ell+1)^{(\ell+1)}}P_{ij}(Y)\right)^2\right]$$

*Proof of Lemma A.2.* First we note that for $\ell = O(\log_\delta n)$, algorithm 2 runs in time $n^{\delta^{-O(1)}}$.

For any random coloring $c$ and length-$\ell$ self-avoiding walk $\alpha$, the probability that $F_{c,\alpha} = 1$ is $(\ell+1)!/(\ell+1)^{\ell+1}$. Thus $\mathbb{E}\, F_{c,\alpha} = (\ell+1)!/\ell^{\ell+1}$. Since $p_{c,i,j} = \sum_{\alpha\in\mathrm{SAW}_\ell(i,j)} F_{c,\alpha}\chi_\alpha(Y)$, by linearity of expectation we get the first equality.

By lemma A.1, we have

$$\mathop{\mathbb{E}}_{Y,c_1,\ldots,c_C}\left[\left(\frac{1}{C}\sum_t p_{c_t,i,j}(Y) - \mathop{\mathbb{E}}_c p_{c,i,j}(Y)\right)^2\right] \le \exp(-O(\ell))\mathop{\mathbb{E}}_{Y}\left(\mathop{\mathbb{E}}_c P_{c,i,j}(Y)\right)^2$$

Therefore

$$\mathop{\mathbb{E}}_{Y,c_1,\ldots,c_C}\left[\left(\frac{1}{C}\sum_t p_{c_t,i,j}(Y)\right)^2\right] \le \left(1+n^{-\Omega(1)}\right)\mathop{\mathbb{E}}_{Y}\left[\left(\mathop{\mathbb{E}}_c p_{c,i,j}(Y)\right)^2\right]$$

Further as stated above we have $\mathbb{E}_c\, p_{c,i,j}(Y) = \frac{(\ell+1)!}{(\ell+1)^{\ell+1}}P_{ij}(Y)$. Thus we get the inequality.     □

Now the proof of theorem 1.2 is self-evident.

*Proof of Theorem 1.2.* We denote $\frac{1}{C}\sum_{t=1}^C p_{c_t}(Y)$ as $p(Y)$. Then by lemma A.2 and lemma 2.1, we have

$$\frac{\mathbb{E}_{c_1,\ldots,c_C,Y}\langle \hat{p}(Y), xx^\top\rangle}{n\left(\mathbb{E}_{c_1,\ldots,c_C,Y}\|\hat{p}(Y)\|_F^2\right)^{1/2}} = \delta^{O(1)} = \Omega(1)$$

By the same rounding procedure as in [HS17](or theorem 1.6), we obtain theorem 1.2 by extracting a random vector in the span of top $1/\delta^{O(1)}$ eigenvectors of $Y$.     □

## A.3   Guarantee and failure of Truncation algorithm

In this section we show that while truncating entries at threshold $\tau(n)$ can help on many occasions, it can fail for some noise distributions we consider.

The class of truncation algorithm we consider can be described as following:

**Algorithm A.3.** Given matrix $Y \in \mathbb{R}^{n\times n}$, set truncation threshold $\tau = \tau(n)$. We first obtain $Y'$ by truncating the entries $Y_{ij}$ with magnitude larger than $\tau$ to $\mathrm{sgn}(Y_{ij})\tau$. Then, we obtain $Y''$ by subtracting the average value of all entries in $Y'$. Finally we extract the top eigenvector of $Y''$. We return this top eigenvector with probability $1/2$ and all $1$ vector with probability $1/2$.

First we show that for many long tail distributions, PCA algorithm can be saved by such truncation. (We defer the proof to appendix A.4.)

**Theorem A.4.** *Consider problem 1.1 such that the signal-to-noise ratio satisfies $\varepsilon = n^{1/2}\lambda - 1 = \Omega(1)$, the upper triangular entries of $W$ are identically distributed, the entries of $X = \lambda xx^\top$ are bounded by $o(1)$. Then, for $\tau = \frac{100}{min(\varepsilon^2, 1)}$, the algorithm A.3 outputs unit norm estimator $\hat{x} \in \mathbb{R}^n$ s.t $\mathbb{E}\langle x, \hat{x}\rangle^2 = \Omega(n)$.*

However, as illustrated by the following examples, this truncation strategy can fail inherently when the noise entries are not identically distributed and their distributions are adversarially chosen (depending on the vector $x$). We show that there is no choice of truncation level $\tau$ for which Algorithm A.3 outputs a vector whose correlation with the planted vector $x$ is nonvanishing for all choices of noise matrix $W$ whose entries are independently sampled with zero mean and unit variance.

First truncating at $\tau = \Omega(\sqrt{n})$ fails the following example

**Example A.5.** For $d = \omega(1)$, $W_{ij}$ equals to $-\sqrt{\frac{n-d}{d}}$ with probability $d/n$ and $\sqrt{\frac{d}{n-d}}$ with probability $1 - \frac{d}{n}$. [3]

This is just normalized and centralized adjacency matrix of Erdos-Renyi random graph, the spectrum of which is well studied in the literature [BGBK17, MS16]. For $d = \omega(1)$, the spectral norm is of order $\omega(\sqrt{n})$, much larger than the spectral norm of $\lambda xx^\top$. Therefore, the leading eigenvector will not be correlated with hidden vector $x$ as we desire.

Then we only need to consider $\tau = o(\sqrt{n})$. We consider the example below

**Example A.6.** For $i + j$ even, we let $W_{ij}$ sampled as in example A.5. For $i + j$ odd, we let $-W_{ij}$ distributed the same as above.

Below, we analyze a strategy where entries $Y_{ij} > \tau$ are truncated to 0. Similar results for truncation to $\tau \mathrm{sgn}(Y_{ij})$ are in the appendix A.5.

For $d = o(n/\tau^2)$, only entries perturbed by noise $\pm\sqrt{\frac{d}{n-d}}$ are preserved. Then $Y'_{ij} = \lambda x_i x_j + \sqrt{\frac{d}{n-d}}(-1)^{i+j}$ with probability $1 - \frac{d}{n}$ and 0 with probability $d/n$. Therefore the leading eigenvector of $Y'$ will be well correlated with $h$ rather than $x$. Since $Y'$ has zero mean, $Y'' - Y'$ has small Frobenius norm, thus the leading eigenvector of $Y''$ is close to $Y'$.

## A.4  Proof of Theorem A.4

For proof of theorem A.4, we need a result available in previous literature stating about the universality of spiked matrix model

**Theorem A.7** (Theorem 1.1 in [PRS13])**.** *In spiked matrix model $Y = \lambda xx^T + W$, $x \in \mathbb{R}^n$ has norm $\sqrt{n}$, $W \in \mathbb{R}^{n \times n}$ is a symmetric random matrix of i.i.d entries with zero mean and variance bounded by 1. If the 5-th moment of entries in $W$ is bounded by $O(1)$, then the following guarantee will hold w.h.p:*

$$\lambda_{max}(Y) \geq (1 - o(1))\left(\lambda n + \frac{1}{\lambda}\right)$$

We also need a simple observation about the deterministic relation between leading eigenvalue and leading eigenvector in spiked matrix model.

**Lemma A.8.** *For matrix $M \in \mathbb{R}^{n \times n}$ and matrix $N = \gamma xx^T + M$(where $\gamma > 0$ and $x \in \mathbb{R}^n$ has $\sqrt{n}$), if the leading eigenvalue $\lambda_{max}(N)$ is larger than $\lambda_{max}(M)$ by $\Omega(n\gamma)$, then the unit norm leading eigenvector of $N$ denoted by $\xi$ will achieve constant correlation with $x$:*

$$\langle \xi, x\rangle^2 \geq \Omega(n)$$

*Proof.* We have $\lambda_{max}(N) = \xi^\top(\gamma xx^\top + M)\xi \leq \lambda_{max}(M) + \gamma\langle\xi, x\rangle^2$. Since $\lambda_{max}(N) - \lambda_{max}(M) = \Omega(n\gamma)$, we have $\langle\xi, x\rangle^2 = \Omega(n)$. $\qquad\square$

*Proof of Theorem A.4.* First if $\langle x, \mathbb{1} \rangle^2 = \Omega(n^2)$, then with probability $1/2$, the algorithm outputs $\frac{1}{\sqrt{n}}\mathbb{1}$ vector. Thus $\mathbb{E}\langle x, \hat{x} \rangle^2 = \Omega(n)$. Next we only consider the case $\langle x, \mathbb{1} \rangle^2 = o(n^2)$.

By definition we have

$$Y'_{ij} = Y \mathbb{1}_{|Y_{ij}| < \tau} + \tau \mathrm{sgn}(Y_{ij}) \mathbb{1}_{|Y_{ij}| \geq \tau}$$

Given the assumption on the $|\lambda x_i x_j| = o(\tau)$, one can observe that this can be decomposed into

$$Y' = \lambda xx^\top + T + M + \Delta$$

where we have

$$
\begin{aligned}
T_{ij} &= W_{ij} \mathbb{1}_{|W_{ij}| < \tau} + \tau \mathrm{sgn}(W_{ij}) \mathbb{1}_{|W_{ij}| \geq \tau} \\
M_{ij} &= -\lambda x_i x_j \mathbb{1}_{|W_{ij}| \geq \tau} \\
\Delta_{ij} &= (\tau \mathrm{sgn}(Y_{ij}) - Y_{ij})(\mathbb{1}_{|\lambda x_i x_j + W_{ij}| \geq \tau} - \mathbb{1}_{|W_{ij}| \geq \tau})
\end{aligned}
$$

Further we denote $Y' - Y''$ as $H$. Then we have $Y'' = \lambda(x - \bar{x})(x - \bar{x})^\top + (T - \mathbb{E}\,T) + M + \Delta - (H - \mathbb{E}\,T - \lambda \bar{x}\bar{x}^\top)$, where $\bar{x} = \frac{\sum_{i=1}^n x_i}{n}\mathbb{1}$

Next we analyze the terms in the decomposition of $Y''$. Specifically, we want to show that with constant probability the largest eigenvalue of $Y''$ is larger than the one of $Y'' - \lambda(x - \bar{x})(x - \bar{x})^T$ by $\Omega(\lambda n)$. If this is proved then for the leading unit eigenvector of $Y''$ denoted by $\hat{x}$, we must have $\langle \hat{x}, x - \bar{x} \rangle^2 = \Omega(n)$ with constant probability by lemma A.8. The theorem is then proved since $\langle \hat{x}, \bar{x} \rangle = o(n)$.

First for matrix $T - \mathbb{E}T$, the variance of each entry is bounded by 1. Further each entry is bounded by $2\tau$. According to theorem A.7, the largest eigenvalue of matrix $\lambda(x - \bar{x})(x - \bar{x})^\top + T - \mathbb{E}T$ is given by $\lambda n + \frac{1}{\lambda} - o(\lambda n)$ and the largest eigenvalue of matrix $T - \mathbb{E}T$ is given by $2\sqrt{n}$ with high probability.

For matrix $M$, we have $\mathbb{E}\mathbb{1}_{|W_{ij}| \geq \tau} \leq \frac{1}{\tau^2}$ because the variance of entries in $W$ is bounded by 1. Therefore the expectation $\mathbb{E}\|M\|_F$ is bounded by $\frac{\lambda n}{\tau}$. For non-zero entries $(i, j)$ in matrix $\Delta$, we must have $|Y_{ij} - \tau \mathrm{sgn}(Y_{ij})| \leq |\lambda x_i x_j|$. Therefore these non zero entries $\Delta_{ij}$ are bounded by $|\lambda x_i x_j|$. Further each entry in $\Delta$ is non-zero with probability bounded by $\frac{1}{\tau^2}$. Therefore we have $\mathbb{E}\|\Delta\|_F$ bounded by $\frac{\lambda n}{\tau}$.

Finally we have $H = h\mathbb{1}\mathbb{1}^\top$ where $h = \frac{\sum_{i,j} Y'_{i,j}}{n^2}$. We denote $\sum_{ij} T_{ij}/n^2$ as $t$. Since $T_{ij}$ are i.i.d for $i \leq j$, we have

$$\mathbb{E}\left[(t - \mathbb{E}[T_{ij}])^2\right] \leq \frac{4}{n^2}$$

Further by assumption $\sum_i x_i = 0$, we have

$$(h - t)^2 \leq \sum_{ij} \frac{(M_{ij} + \Delta_{ij})^2}{n^2}$$

By linearity of expectation $\mathbb{E}\|H - \mathbb{E}T\| \leq \mathbb{E}\|H - t\mathbb{1}\mathbb{1}^\top\|_F + \mathbb{E}\|t\mathbb{1}\mathbb{1}^\top - \mathbb{E}T\|_F \leq \frac{(2+o(1))\lambda n}{\tau}$.

In all we have $\mathbb{E}\|Y'' - \lambda xx^\top - T + \mathbb{E}T\| \leq \frac{(4+o(1))\lambda n}{\tau}$. By Markov inequality with probability $1/2$, we have $\|Y'' - \lambda xx^\top - T + \mathbb{E}T\| \leq \frac{9\lambda n}{\tau}$. As stated above, the largest eigenvalue of matrix $\lambda xx^\top + T - \mathbb{E}T$ is given by $\lambda n + \frac{1}{\lambda}$ and the largest eigenvalue of matrix $T - \mathbb{E}T$ is given by $2\sqrt{n}$ with high probability. If we take $\tau$ large enough constant(e.g $\frac{100}{\min(\varepsilon^2, 1)}$), then with probability $\Omega(1)$ the spectral norm of $Y''$ is larger than $\lambda n + \frac{1}{\lambda} - 0.1\min(\varepsilon^2, 1)\lambda n$ and spectral norm of $Y'' - \lambda xx^\top$ is smaller than $2\sqrt{n} + 0.1\min(\varepsilon^2, 1)\lambda n$.

Therefore for $\lambda = 1 + \varepsilon$ with $\varepsilon = \Omega(1)$, the spectral norm of $Y''$ is larger than the spectral norm of $Y'' - \lambda(x - \bar{x})(x - \bar{x})^\top$ by $\Omega(\lambda n)$ with constant probability. As a result, with constant probability the leading eigenvector of $Y''$ must achieve $\Omega(1)$ correlation with hidden vector $x$ by lemma A.8. $\square$

## A.5 Example of failure for truncation algorithm

For example A.6, we show that truncating to $\tau \text{sgn}(Y_{ij})$ will fail as well for any $d$ between $o(n/\tau^2)$ and $\omega(1)$(note that $n \gg \tau^2$).

We still denote $h = \{\pm 1\}^n$ as the Rademacher vector with alternating sign and $x \in \{\pm 1\}^n$ orthogonal to all-1 vector. We take $\lambda = \Theta(n^{-1/2})$. For $d = o(n/\tau^2)$ and $d = \omega(1)$, only entries perturbed by noise $\pm\sqrt{\frac{d}{n-d}}$ are not truncated. Then $Y'_{ij} = \lambda x_i x_j + \sqrt{\frac{d}{n-d}}(-1)^{i+j}$ with probability $1 - \frac{d}{n}$ and $-(-1)^{i+j}\tau$ with probability $d/n$. Therefore $Y'$ can be decomposed into

$$Y' = \lambda x x^\top + \sqrt{\frac{d}{n-d}} h h^\top + \Delta$$

where $\Delta_{ij} = \sqrt{\frac{d}{n-d}}(-1)^{i+j}$ with probability $1 - \frac{d}{n}$, and $-\tau - \lambda x_i x_j$ with probability $\frac{d}{n}$.

Since $\mathbb{E}\,\Delta_{ij} = -\frac{\tau d}{n}(-1)^{i+j} - \frac{\lambda d}{n}x_i x_j$, we have $\|\mathbb{E}\,\Delta\| = o(\sqrt{nd})$. Further for matrix $\Delta - \mathbb{E}\,\Delta$, the entries are independently distributed and the variances of entries are bounded by $O(1)$. Further the absolute value of entries are bounded by $2\tau$. Thus by the bound on the operator norm of Wigner matrix, we have $\|\Delta - \mathbb{E}\,\Delta\| = O(\sqrt{n}) = o(\sqrt{nd})$. Combining these, we have $\|\Delta\| = o(\sqrt{nd})$.

Above computational threshold $\lambda \geq n^{-1/2}$, the spectral norm of $\lambda x x^\top$ is smaller than $\sqrt{n}$. Further matrix $H = Y'' - Y'$ also has spectral norm $o(\sqrt{nd})$ by central limit theorem.

For unit norm leading eigenvector $\xi$ of matrix $Y''$, we suppose that $\mathbb{E}\langle \xi, x \rangle^2 \geq \Omega(n)$ and prove by contradiction. Because $\xi$ is leading eigenvector, we have $\mathbb{E}\langle \xi\xi^\top, Y'' \rangle \geq \mathbb{E}\langle hh^\top, Y'' \rangle \geq (1 - o(1))\sqrt{nd}$. However, we have $\langle \xi, h \rangle^2/n \leq 1 - \langle \xi, x \rangle^2/n$. Therefore, we have $\langle \xi\xi^\top, Y'' \rangle \leq (1 - \Omega(1))\sqrt{nd} + o(\sqrt{nd})$. This leads to contradiction.

## A.6 Algorithm for evaluating non-backtracking walk estimator

In experiments, we use estimator closely related to non-backtracking walk and color coding method.

On complete graph $K_n$, for vertice labels $i, j \in [n]$, we define the set of length-$\ell$ $k$-step non-backtracking walks $(i, v_1, v_2, \ldots, v_{\ell-1}, j)$ as $\text{NBW}_\ell(i, j)$. For non-backtracking walk $\alpha$ and random coloring $c : [n] \mapsto [\ell + 1]$, we denote $F_{c,\alpha}$ as $0, 1$ indicator of the random event that each length $k$ chunk of walk $\alpha$ is colorful(i.e, not containing repeated colors). For a fixed path $\alpha$, the probability that $F_{c,\alpha} = 1$ is bounded by $\left(1 - \frac{k}{\ell+1}\right)^\ell \geq \exp(-O(k))$.

Then we use a randomized non-backtracking walk estimator $P(Y) \in \mathbb{R}^{n \times n}$

$$P_{ij}(Y) = \sum_{t=1}^{C} \sum_{\alpha \in \text{NBW}_\ell(i,j)} \chi_\alpha(Y) F_{c_t,\alpha} \tag{4}$$

where colorings $c_1, c_2, \ldots, c_C : [n] \mapsto [\ell + 1]$ are taken uniformly at random, and $\chi_\alpha(Y) = \prod_{(u,v) \in \alpha} Y_{u,v}$. The expectation of this estimator is given by

$$P_{ij}(Y) = \sum_{\alpha \in \text{NBW}_\ell(i,j)} \chi_\alpha(Y) \,\mathbb{E}_c F_{c,\alpha}$$

where the expectation on $c : [n] \mapsto [\ell + 1]$ is taken uniformly at random.

We now describe how to construct matrix $H, M, N$. For matrix $M$, corresponding to index $((v_1, S), (v_2, T))$, the entry is given by $Y_{v_1, v_2}$ if

- the color of $v_1$ is not contained in $S$ and the color of $v_2$ is not contained in $T$
- ordered set $T$ is the concatenation of color of $v_1$ and first $\ell - 1$ elements of $S$.

Otherwise the entry is given by $0$.

For matrix $H$, corresponding to entry $(v_1, (v_2, S))$, the entry is given by $Y_{v_1, v_2}$ if $S$ contains single element: the color of $v_1$ and the color of $v_2$ is different with the color of $v_1$.

---

**Algorithm 3:** Algorithm for evaluating color-coding non-backtracking walk matrix

---

**Data:** Given $Y \in \mathbb{R}^{n \times n}$ s.t $Y = \lambda x x^\top + W$

**Result:** Approximation for $P(Y) \in \mathbb{R}^{n \times n}$ where $P_{ij}(Y) = \sum_{\alpha \in \mathrm{NBW}_\ell(i,j)} \hat{p}_\alpha \chi_\alpha(Y)$ where

$\quad$ $\mathrm{NBW}_\ell(i,j)$ is the set of length $\ell$ non-backtracking walk between $i, j$ and $\hat{p}_\alpha = \mathbb{E}_c F_{c,\alpha}$

**for** $t \leftarrow 1$ **to** $C$ **do**

$\quad$ Sample a random coloring $c_t : [n] \mapsto [\ell + 1]$ ;

$\quad$ Construct a $\mathbb{R}^{n \sum_{s=0}^{k} (\ell+1)^s \times n \sum_{s=0}^{k} (\ell+1)^s}$ matrix $M$, with rows and columns are indexed by

$\quad$ $(v, S)$, where $v \in [n]$ and $S$ is an ordered subset of $[\ell + 1]$ with size bounded by $k$ ;

$\quad$ A matrix $H \in \mathbb{R}^{n \times n \sum_{s=0}^{k} (\ell+1)^s}$ with rows indexed by $[n]$ and each column indexed by

$\quad$ $(v, S)$ where $v \in [n]$ and $S$ is ordered subset of $[\ell + 1]$ with size bounded by $k$;

$\quad$ A matrix $N \in \mathbb{R}^{n \sum_{s=0}^{k} \ell^s \times n}$, with columns indexed by $[n]$ and each row indexed by $(v, S)$

$\quad$ where $v \in [n]$ and $S$ is ordered subset of $[\ell + 1]$ with size bounded by $k$;

$\quad$ Record $p_{c_t} = H M^{\ell-2} N$;

Return $\sum_{t=1}^{C} p_{c_t} / C$

---

For matrix $N$, corresponding to entry $((v_1, S), v_2)$, the entry is given by $Y_{v_1, v_2}$ if

- $S$ contains $k$ colors, and the color of $v_1$ is different from the last color of $S$
- the color of $v_2$ is different from the color of $v_1$ and first $k - 1$ elements of $S$

and given by $0$ otherwise.

**Fact A.9.** *For a single random coloring $c(v) : [n] \mapsto [\ell + 1]$, the $(i, j)$ entry of $p_c(Y) \in \mathbb{R}^{n \times n}$ evaluated in the algorithm is exactly $\sum_{\alpha \in NBW_\ell(i,j)} \chi_\alpha(Y)$.*

If $P(Y)$ achieves constant correlation with $xx^\top$, then a random vector in the span of top-$\delta^{-O(1)}$ eigenvectors of matrix $P(Y)$ can be sampled in quasilinear time.

**Theorem A.10** (Evaluation of color-coding Non-backtracking walk estimator). *In spiked matrix model $Y = \lambda x x^T + W$, vector $x \in \mathbb{R}^n$ has norm $\sqrt{n}$ and entries in $W \in \mathbb{R}^{n \times n}$ are independently sampled with zero mean and unit variance. Considering $\ell = O(\log_{1+\delta} n)$ with $\delta = \Omega(1)$ and $k = O(1)$. Then for $P(Y)$ evaluated in algorithm, a unit norm random vector in the span of top $\delta^{-O(1)}$ eigenvectors of $P(Y)$ can be sampled in time $O(n^2 \log^{k+1}(n) \delta^{-O(1)})$.*

*If we assume that the matrix $P(Y)$ achieves correlation $\delta$ with $xx^\top$, then this random vector $\xi$ achieves constant correlation with $x$: $\langle \xi, x \rangle^2 = \delta^{O(1)} n$.*

*Proof.* For extracting the span of top $\delta^{-O(1)}$ eigenvectors, we apply power method. Since $p(Y)$ can be represented as a sum of chain product of matrices, we can iteratively apply matrix-vector product rather than obtaining $p(Y)$ explicitly. Since for matrix $H, M, N$, there are at most $n^2 \log^{k+1}(n)$ non-zero elements. The resulting complexity is thus given by $O(n^2 \log^{k+1}(n) \delta^{-1} \exp(O(k)))$.

If $\frac{\left( \mathbb{E} \langle P(Y), xx^\top \rangle \right)^2}{n^2 \, \mathbb{E} \, P^2(Y)} = \delta$, then applying theorem 1.6, a random vector in the span of top $\delta^{-O(1)}$ eigenvectors of $\hat{P}(Y)$ achieves $\delta^{O(1)}$ correlation with the spiking vector $x$.

$\square$

## A.7 Experiments

For comparing the performance of algorithms proposed, we conduct experiments with several typical distributions of noise: (1) the noise is distributed as example A.5. (2) the noise is distributed as example A.6 (3) entry $W_{ij}$ is distributed as $N(0,1)$ when $i + j$ is even and as example A.5 when $i + j$ is odd. In each case planted vector $x$ is randomly sampled from $N(0, \mathrm{Id}_n)$. In these examples, the smaller parameter $d$ corresponds to the more heavy tailed noise distribution.

In experiments with size $n = 10^2 - 10^3$, self-avoiding walk estimator shows better performance than naive PCA and truncation PCA algorithm. Furthermore, the non-backtracking algorithm achieves

(a) $n = 200, \lambda' = 1.5, d \in [0.1, 8]$    (b) $n = 200, \lambda' = 1.5, d \in [0.1, 8]$

(c) $n = 400, \lambda' = 1.1, d \in [0.05, 5]$    (d) $n = 2000, \lambda' \in [1.2, 1.6], d = 30$

Figure 2: (a)(b) The performance of non-backtracking walk estimator with $\ell = 10$ is no worse than self-avoiding walk estimator with $\ell = 7$ under distribution (2),(3). They drastically beat naive PCA algorithm. (c) The performance of non-backtracking walk estimator with length $\ell = 17$ can be much better than PCA under distribution. (1) (d) Truncating at $\tau = 5$ can fail drastically under distribution (2).

Each data point is the result of averaging 20 trials. For notation, $\lambda' = \lambda n^{1/2}$, the $y$ axis represents mean of squared correlation $\frac{\langle \hat{x}, x \rangle^2}{\|\hat{x}\|^2 \|x\|^2}$. The line "worst" represents the optimal guarantee in case of Gaussian noise with same $\lambda$, while the line "NBW" represents the experiment results from non-backtracking algorithm.

performance no worse than self-avoiding walk estimator under many settings. The results are shown in figure 2.

# B    Order-3 spiked tensor model

## B.1    Notations

For the hyperedges of $p$-uniform directed hypergraph, we denote the directed hyperedge between vertices labelled by $i_1, i_2, \ldots, i_p$ as ordered-triplet $(i_1, i_2, \ldots, i_p)$. We denote $\alpha \cap \beta$ as an hypergraph containing all of the shared edges and shared vertices between $\alpha, \beta$. We denote $\alpha \Delta \beta$ as an hypergraph containing all of the remaining hyperedges uniquely contained by $\alpha$ or $\beta$ and vertices not incident to shared hyperedges.

We denote the set of order-3 real tensor as $\mathbb{R}^{n \times n \times n}$ and the set of general order-$p$ tensor as $(\mathbb{R}^n)^{\otimes p}$. For vector $x \in \mathbb{R}^n$, we denote order $p$ tensor whose $(i_1, i_2, \ldots, i_p)$ entry is given by $x_{i_1} x_{i_2} \ldots x_{i_p}$ as $x^{\otimes p}$. We also use $N(0,1)^{n \times n \times n}$ to represent random tensor whose entries are i.i.d sampled from $N(0,1)$. We use $\|\cdot\|$ for the vector norm and spectral norm and $\|\cdot\|_F$ for Frobenius norm.

## B.2 Strong detection and weak recovery in order-3 spiked tensor model

For spiked tensor model, we define strong detection problem. Specifically given tensor $Y$ sampled from general spiked tensor model, we want to detect whether it's sampled with $\lambda = 0$ or large $\lambda$ with high probability.

**Definition B.1** (Strong detection). Given tensor $Y$ sampled from

- Planted distribution $\mathbb{P}$: the random tensor $Y \in \mathbb{R}^{n \times n \times n}$ is sampled as $Y = \lambda x^{\otimes 3} + W$, where $x \in \mathbb{R}^n$ is a random vector s.t $\mathbb{E}\, x_i = 0$, $\mathbb{E}\, x_i^2 = 1$, and $W \in \mathbb{R}^{n \times n \times n}$ has independent, zero mean and unit variance entries.

- Null distribution $\mathbb{Q}$: where random tensor $Y \in \mathbb{R}^{n \times n \times n}$ has independent, zero-mean and unit variance entries.

with equal probability, we need to find a function of entries in $Y$: $f(Y) \in \{0, 1\}$ such that

$$\frac{1}{2}\mathbb{P}[f(Y) = 1] + \frac{1}{2}\mathbb{Q}[f(Y) = 0] = 1 - o(1)$$

We also define the notion of weak recovery and strong recovery in spiked tensor model.

**Definition B.2.** In spiked tensor model $Y = \lambda x^{\otimes 3} + W$ where $x \in \mathbb{R}^n$ is a random vector s.t $\mathbb{E}\, x_i = 0$, $\mathbb{E}\, x_i^2 = 1$, and $W \in \mathbb{R}^{n \times n \times n}$ has independent, zero mean and unit variance entries, We define that estimator $\hat{x}(Y) \in \mathbb{R}^n$ achieves weak recovery if $\mathbb{E}\langle \hat{x}(Y), x \rangle \geq \Omega\left( \left( \mathbb{E}\|\hat{x}(Y)\|^2\, \mathbb{E}\|x\|^2 \right)^{1/2} \right)$. Further we define that $\hat{x}(Y) \in \mathbb{R}^n$ achieves strong recovery if $\langle \hat{x}(Y), x \rangle \geq (1 - o(1))\|\hat{x}(Y)\|\|x\|$ with high probability.

**Remark**: The weak recovery here can be equivalently defined as that with constant probability $\left\langle \frac{\hat{x}(Y)}{\|\hat{x}(Y)\|}, \frac{x}{\|x\|} \right\rangle^2 \geq \Omega(1)$. The equivalence follows by Markov inequality.

## B.3 Strong detection algorithm for spiked tensor model

It's not explicitly stated in previous literature how to obtain strong detection algorithm via low degree method. The following self-clear fact provides a systematic way for doing so.

**Theorem B.3** (Low degree polynomial thresholding algorithm). *We denote the likelihood ratio between planted distribution $\mathbb{P}$ and null distribution $\mathbb{Q}$ as $\mu(Y) = \frac{\mathbb{P}(Y)}{\mathbb{Q}(Y)}$. Given $Y$ sampled from distributions $\mathbb{P}$ and $\mathbb{Q}$ with equal probability, for polynomial $P(Y) = \sum_{\alpha \in S_\ell} \hat{\mu}_\alpha \chi_\alpha(Y)$ where $\{\chi_\alpha(Y) : \alpha \in S_\ell\}$ is a set of polynomial basis orthonormal under measure $\mathbb{Q}$ and $\hat{\mu}_\alpha = \mathbb{E}_{\mathbb{P}} \chi_\alpha(Y)$, if we have*

- *diverged low degree likelihood ratio $\sum_{\alpha \in S_\ell} \hat{\mu}_\alpha^2 = \omega(1)$*

- *concentration property $\mathbb{E}_{\mathbb{P}} P^2(Y) = (1 + o(1)) \left( \mathbb{E}_{\mathbb{P}} P(Y) \right)^2$*

*then this implies strong detection algorithm by thresholding polynomial $P(Y)$.*

**Remark**: The quantity $\hat{\mu}_\alpha$ is actually the projection of likelihood ratio function $\mu(Y)$ with respect to polynomial basis $\chi_\alpha(Y)$. For polynomial $P(Y)$ defined here, we say that it is the projection of likelihood ratio $\mu(Y)$ with respect to the set $S_{\ell,v}$.

*Proof.* Since $\sum_{\alpha \in S_\ell} \hat{\mu}_\alpha^2 = \omega(1)$, we have $\mathbb{E}_{\mathbb{P}} P(Y) = \mathbb{E}_{\mathbb{Q}} P^2(Y) = \omega(1)$. By Chebyshev's inequality, for $Y \sim \mathbb{P}$ w.h.p we have $P(Y) = (1 \pm o(1)) \mathbb{E}_{\mathbb{P}} P(Y) = \omega(\sqrt{\mathbb{E}_{\mathbb{Q}} P^2(Y)})$ while for $Y \sim \mathbb{Q}$ w.h.p we have $P(Y) = O(\sqrt{\mathbb{E}_{\mathbb{Q}} P^2(Y)})$ $\qquad\square$

Our guarantee for strong detection in spiked tensor model can be stated as following:

**Theorem B.4.** *For any small constant $\delta = \Omega(1)$, $c \geq n^{-1/8+\delta}$ and $\lambda \geq cn^{-3/4}$, there is $n^{O(1/c^4)}$ time algorithm achieving strong detection in spiked tensor model, i.e distinguishing distributions*

Figure 3: An example of possible directed hyperedge connection between adjacent layers $t-1, t, t+1$ for an hypergraph $\alpha \in S_{\ell,3}$. (1) When $t \in [2\ell - 2]$, the direction of hyperedges are given by $(v_1, v_4, v_5), (v_2, v_6, v_7), (v_3, v_8, v_9), (v_4, v_6, v_{10}), (v_5, v_7, v_{11}), (v_8, v_9, v_{12})$. (Note that each hyperedge directs from the layer $t-1$ to the layer $t$ or from the layer $t$ to the layer $t+1$.) (2) When $t = 0$, the layers are given by $2\ell - 2, 0, 1$ by periodic indexing. The directions of hyperedges are given by $(v_4, v_5, v_1), (v_6, v_7, v_2), (v_8, v_9, v_3), (v_4, v_6, v_{10}), (v_5, v_7, v_{11}), (v_8, v_9, v_{12})$.

- *Planted distribution $\mathbb{P}$: the random tensor $Y \in \mathbb{R}^{n \times n \times n}$ is sampled as $Y = \lambda x x^\top + W$, where $x \in \mathbb{R}^n$ is a random vector s.t $\mathbb{E}\, x_i = 0$, $\mathbb{E}\, x_i^2 = 1$ and $\mathbb{E}\, x_i^4 \leq n^{o(1)}$, and $W \in \mathbb{R}^{n \times n \times n}$ has independent, zero mean and unit variance entries.*

- *Null distribution $\mathbb{Q}$: where random tensor $Y \in \mathbb{R}^{n \times n \times n}$ has independent, zero-mean and unit variance entries.*

For strong detection algorithm, the thresholding polynomial we use is given by the following.

**Definition B.5** (Thresholding polynomial for strong detection). On the directed complete 3-uniform hypergraph with $n$ vertices, we define $S_{\ell,v}$ as the set of all copies of 2-regular hypergraphs generated in the following way:

- we construct $2\ell$ levels of distinct vertices labeled by $0, 1, \ldots, 2\ell-1$. For levels $t \in [0, 2\ell-1]$, it contains $v$ vertices if $t$ is even and $2v$ vertices if $t$ is odd.

- Then we construct a perfect matching between levels $t, t-1$ for $t \in [2\ell - 1]$ and between levels $0, 2\ell - 1$. For each hyperedge, 1 vertex comes from even level while 2 vertices come from odd level. The hyperedges are directed from level 0 to level $2\ell - 1$ and from level $t$ to level $t + 1$ for $t \in [0, 2\ell - 2]$. (An example of such construction is illustrated in figure 3).

Given tensor $Y \in \mathbb{R}^{n \times n \times n}$, the degree $2\ell v$ polynomial $P(Y)$ is given by $P(Y) = \sum_{\alpha \in S_{\ell,v}} \chi_\alpha(Y)$, where $\chi_\alpha(Y)$ is the corresponding multilinear polynomial basis $\chi_\alpha(Y) = \prod_{(i,j,k) \in \alpha} Y_{ijk}$.

For the null distribution $\mathbb{Q}$, since polynomials in the set $\{\chi_\alpha(Y) : \alpha \in S_{\ell,v}\}$ are multilinear, they are orthogonal with respect to each other.

For simplicity of formulation, we use periodic index below (i.e for level $t = -1$ we mean level $2\ell - 1$).

For proving strong detection guarantee, we first need two hypergraph properties.

**Lemma B.6.** *On directed complete hypergraph with $n$ vertices, the number of hypergraphs contained in the set $S_{\ell,v}$ defined in B.5 is given by*

$$|S_{\ell,v}| = (1 - o(1)) \left( \binom{n}{v} \binom{n}{2v} ((2v)!)^2 \right)^\ell$$

*Proof.* For fixed vertices, between level $t$ and level $t + 1$ there are $\frac{(2v)!v!}{v!}$ ways of connecting hyperedges. Therefore we have $|S_{\ell,v}| = (1 - o(1)) \left( \binom{n}{v} \binom{n}{2v} \left( \frac{v!(2v)!}{v!} \right)^2 \right)^\ell$. $\qquad \square$

**Lemma B.7.** *For a pair of hypergraphs $\alpha, \beta \in S_{\ell,v}$, we denote the number of shared vertices as $r$, the number of shared hyperedges as $k$, and the number of shared vertices with degree $0$ or $1$ in $\alpha \cap \beta$ as $s$. Then we have relation $2r - s \geq 3k$.*

*Further if $2r = 3k$, then*

- *either $\alpha, \beta$ are disjoint $k = r = 0$*

- *or for all levels $t \in [0, 2\ell - 1]$, the number of shared hyperedges between level $t$ and level $t + 1$ $k_t \geq 1$ and are equal to the same value.*

*Proof.* In 3-uniform sub-hypergraph $\alpha \cap \beta$, there are $k$ hyperedges, $r - s$ vertices with degree 2 and at most $s$ vertices with degree 1. By degree constraint, we have $3k \leq 2(r - s) + s = 2r - s$.

When $2r = 3k$, each shared vertice between $\alpha, \beta$ has degree 2 in $\alpha \cap \beta$. However if there exists $t \in [0, 2\ell - 1]$ such that $k_{t-1} \neq k_t$, then there are shared vertices at level $t - 1$ with degree 1 or 0 in subgraph $\alpha \cap \beta$. Thus either $\alpha, \beta$ are disjoint($k_t = 0$), or for all levels $t, k_t \geq 1$ and are all equal. $\square$

We consider the set of hypergraph pairs $\alpha, \beta \in S_{\ell,v}$ such that in $\alpha$

- at level $t$ there are $r_t$ vertices shared with $\beta$
- between level $t + 1$ and level $t$, there are $k_t$ hyperedges shared with $\beta$.

We denote such set of hypergraph pairs as $S_{\ell,v,k,r}$, where $k = \sum k_t, r = \sum r_t$. Then $r$ is just the number of shared vertices between $\alpha, \beta$ as $r$ and $k$ is just the number of shared hyperedges between $\alpha, \beta$. Although we abuse the notations(since the set $S_{\ell,v,k,r}$ is related to $k_t, r_t$), by the following lemma we can bound the size of such set only using $k, r, \ell, v$.

**Lemma B.8.** *On directed complete hypergraph with $n$ vertices, for any set $S_{\ell,v,k,r}$ with $\ell = O(\log n)$, $v = o(n^{1-\Omega(1)})$, $k, r \leq \ell v$, the number of hypergraph pairs contained in the set $S_{\ell,v,k,r}$ is bounded by*

$$|S_{\ell,v,k,r}| \leq |S_{\ell,v}|^2 n^{-r} v^{2r-4k} \ell^{2r-3k} v^{k/2} \exp(O(r))$$

*Proof.* We generate pairs of hypergraph $\alpha, \beta \in S_{\ell,v}$ in the following way: we first choose $\alpha \cap \beta$ and shared vertices as a subgraph of hypergraph in $S_{\ell,v}$, and then choose remaining graph respectively for $\alpha, \beta$. In hypergraph $\alpha$, suppose there are $r_t$ shared vertices in level $t$ and $k_t$ shared hyperedges between level $t + 1$ and level $t$.

We define parity function $\delta(t) = 2$ if $t$ is odd and $1$ if $t$ is even. Then we have relation $r_t \geq \delta(t) \max(k_{t-1}, k_t)$. For choice of $\alpha \cap \beta$, there are

$$N_{\alpha \cap \beta} = \prod_{t=0}^{\ell} \binom{n}{r_{2t}} \binom{n}{r_{2t+1}} \binom{r_{2t}}{k_{2t}} \binom{r_{2t}}{k_{2t-1}} \binom{r_{2t+1}}{2k_{2t}} \binom{r_{2t+1}}{2k_{2t+1}} (2k_{2t})!(2k_{2t+1})!$$

$$\leq n^r v^{k/2} \exp(O(r))$$

such subgraphs. On the other hand, the number of choices for the remaining hypergraph of $\alpha$ is bounded by

$$N_{\alpha-\beta} = \prod_{t=0}^{\ell-1} \binom{n}{v - r_{2t}} \binom{n}{2v - r_{2t+1}} (2(v - k_{2t}))!(2(v - k_{2t+1}))!$$

$$= |S_{\ell,v}| n^{-r} v^{r-2k} \exp(O(r))$$

Then we consider the number of $\beta$. Denote the number of degree-1 vertices in $\alpha \cap \beta$ as $s_1$ and the number of shared vertices with degree 0 in $\alpha \cap \beta$ as $s_0$. Let $s = s_0 + s_1$, then there are at most $\ell^s$ ways of embedding $\alpha \cap \beta$ in $\beta$. The number of choices for the remaining hypergraph of $\beta$ is also bounded by $|S_{\ell,v}| n^{-r} v^{r-2k} \exp(-O(r))$. Therefore, with respect to fixed number of vertices and hyperedges $r_t, k_t$ in each level of $\alpha \cap \beta$, the total number of such hypergraph pairs $S_{r,k,\ell,v}$ is bounded by

$$\sum_{s=0}^{2r-3k} \left[ |S_{\ell,v}|^2 n^{-r} v^{2r-4k} \ell^s \exp(O(r)) \prod_{t=0}^{2\ell-1} \frac{r_t!}{k_t!} \right] \leq |S_{\ell,v}|^2 n^{-r} v^{2r-4k} \ell^{2r-3k} v^{k/2} \exp(O(r))$$

$\square$

Next we prove the strong detection through lemma B.9 and B.12.

**Lemma B.9.** *When $\gamma = 0.001n^{3/2}v^{1/2}\lambda^2 = 1 + \Omega(1)$, $\ell = \Omega(\log_\gamma n)$, the projection of likelihood ratio $\mu(Y)$ with respect to $S_{\ell,v}$ diverges, i.e $\sum_{\alpha \in S_{\ell,v}} \hat{\mu}_\alpha^2 = \omega(1)$.*

*Proof.* Because for any hypergraph $\alpha \in S_{\ell,v}$, we have $|\alpha| = 2\ell v$ and $\hat{\mu}_\alpha = \mathbb{E}\, \chi_\alpha(Y) = \lambda^{2\ell v}$. By lemma B.6, we have

$$\sum_{\alpha \in S_{\ell,v}} \hat{\mu}_\alpha^2 \geq (1 - o(1)) \left( \binom{n}{v}\binom{n}{2v}((2v)!)^2 \right)^\ell \lambda^{4\ell v} \geq (1 - o(1))n^{3\ell v}v^{\ell v}\lambda^{4\ell v}$$

.

For $\gamma = 1 + \Omega(1)$ and $\ell = \Omega(\log_\gamma n)$, we have $\sum_{\alpha \in S_{\ell,v}} \hat{\mu}_\alpha^2 = n^{\Omega(1)}$. $\square$

For proving $(\mathbb{E}\, P(Y))^2 = (1 - o(1))\, \mathbb{E}\, P^2(Y)$, we first prove several preliminary lemmas. First we bound the expectation of $P(Y)$

**Lemma B.10.** *In spiked tensor model $Y = \lambda \cdot x^{\otimes 3} + W$ where entries in $x \in \mathbb{R}^n$ are sampled independently with zero mean and unit variance, entries in $W \in \mathbb{R}^{n \times n \times n}$ are sampled independently with zero mean and unit variance. Then for $P(Y)$ defined in B.5, we have $\mathbb{E}\, P(Y) = (1 - o(1))\lambda^{2\ell v}(\binom{n}{v}\binom{n}{2v}((2v)!)^2)^\ell$.*

*Proof.* First for each $\alpha \in S_{\ell,v}$, we have $\mathbb{E}[\chi_\alpha(Y)] = \lambda^{2\ell v}$. By lemma B.6, we have $|S_{\ell,v}| = (1 - o(1)) \left( \binom{n}{v}\binom{n}{2v}((2v)!)^2 \right)^\ell$. Since $\sum_{\alpha \in S_{\ell,v}} \mathbb{E}[\chi_\alpha(Y)] = \lambda^{2\ell v}|S_{\ell,v}|$, we get the lemma. $\square$

Finally we need a lemma for bounding the variance of the polynomial. For this we prove a bound on the summation over all possible $k_t, r_t$

**Lemma B.11.** *For $t \in \{0, 1, \ldots, 2\ell - 1\}$, we define $r_t, k_t \in \{0, 1, \ldots, 2v\}$ satisfying that $r_t \geq \delta(t)\max(k_{t-1}, k_t)$, where parity function $\delta(t) = 2$ if $t$ is odd and $1$ if $t$ is even. We denote $k = \sum_{t=0}^{2\ell-1} k_t$ and $r = \sum_{t=0}^{2\ell-1} r_t$. We take scalars $\eta = \omega(\ell^2)$ and constant $\psi < 1$. Then for $2r \geq 3k + 1$, we have:*

$$\sum_{\substack{k_t \geq 0}} \sum_{\substack{r_t \geq \delta(t)\max(k_{t-1}, k_t) \\ 2r \geq 3k+1}} \eta^{-r+3k/2}\psi^k \leq o(1)$$

*Proof.* We note that given $k_t$ for $t \in [\ell]$, we have $\ell^{r-3k/2}$ choices for $r_t$. Further we denote $k_\Delta = \sum_t |k_{t+1} - k_t|$. Then given $k_\Delta$, all $k_t$ can take at most $k_\Delta$ different values. As a result, fixing these $k_\Delta$ different values, there are $\ell^{k_\Delta}$ choices for $k_t$ for $t \in [\ell]$. Further we have $r - 3k/2 \geq k_\Delta/2$. Therefore the summation is bounded by

$$\sum_{k_\Delta \geq 1} \left[ \left(\frac{\eta}{\ell^2}\right)^{-k_\Delta/2} \prod_{t=1}^{k_\Delta} \left( \sum_{k_t \geq 0} \psi^{k_t} \right) \right] = o(1)$$

$\square$

**Lemma B.12.** *Denote $\gamma = 0.001n^{3/2}v^{1/2}\lambda^2$ and take $\ell = \Omega(\log_\gamma n)$ in the above estimator $P(Y)$, if we have $\gamma = 1 + \Omega(1)$ and $n = \omega(v^2\Gamma^2 poly(\log n))$, then $\mathbb{E}\, P^2(Y) = (1 + o(1))(\mathbb{E}\, P(Y))^2$.*

*Proof.* We need to show that

$$\left( \sum_{\alpha \in S_{\ell,v}} \mathbb{E}[\chi_\alpha(Y)] \right)^2 = (1 - o(1)) \left( \sum_{\alpha, \beta \in S_{\ell,v}} \mathbb{E}[\chi_\alpha(Y)\chi_\beta(Y)] \right)$$

For left hand side, we already have lemma B.10. Thus we only need to bound $\left(\sum_{\alpha,\beta \in S_{\ell,v}} \mathbb{E}[\chi_\alpha(Y)\chi_\beta(Y)]\right)$. First by direct computation we have

$$\mathbb{E}[\chi_\alpha(Y)\chi_\beta(Y)] = (1 + n^{-\Omega(1)})^k \lambda^{4\ell v - 2k} \mathbb{E}\left[\prod_{i \in \alpha\Delta\beta} x_i^{\deg(i,\alpha\Delta\beta)}\right] \leq \lambda^{4\ell v - 2k}\Gamma^{2r-3k}$$

where $\deg(i,\alpha)$ is the degree of vertex $i$ in hypergraph $\alpha$, $r$ is the number of shared vertices and $k$ is the number of shared hyperedges, $\Gamma = \mathbb{E}[x_i^4] = n^{o(1)}$ according to assumptions. Using lemma B.8, lemma B.10, for any set $S_{r,k,\ell,v}$, we have

$$\frac{\sum_{\alpha,\beta \in S_{r,k,\ell,v}} \mathbb{E}\,\chi_\alpha(Y)\chi_\beta(Y)}{(\mathbb{E}\,P(Y))^2} \leq n^{-r}v^{2r-4k}\ell^{2r-3k}\lambda^{-2k}\Gamma^{2r-3k}v^{\frac{k}{2}}\exp(cr)$$

where $c$ is large enough constant. Summing up for different $r_t, k_t$ and combining the fact that if $r = \frac{3}{2}k$ then $k \geq 2\ell, k_t \geq 1$, then we have

$$\frac{\sum_{\alpha,\beta \in S_{\ell,v}} \mathbb{E}\,\chi_\alpha(Y)\chi_\beta(Y)}{(\mathbb{E}\,P(Y))^2}$$

$$= \sum_{r_t,k_t} c^{3k/2} n^{-3k/2} v^{-k/2} \lambda^{-2k} \left(\frac{n}{cv^2\Gamma^2}\right)^{-r+3k/2} \ell^{2r-3k}$$

$$= \sum_{\substack{k_t \geq 0}} \sum_{\substack{r_t \geq \delta(t)\max(k_{t-1},k_t) \\ 2r \geq 3k+1}} \left(\frac{n}{cv^2\Gamma^2}\right)^{-r+3k/2} \left(c^{3/2} n^{-3/2} \lambda^{-2} v^{-1/2}\right)^k +$$

$$\sum_{k_t \geq 1} \left(c^{3/2} n^{-3/2} \lambda^{-2} v^{-1/2}\right)^{\sum_t k_t} + 1$$

where parity function $\delta(t) = 2$ if $t$ is odd and $1$ if $t$ is even. The term $1$ comes from the case $r = k = 0$. When $\gamma = 0.001 n^{3/2} \lambda^2 v^{1/2} > 1$ and $\ell = C\log_\gamma n$ with constant $C$ large enough, the second term is bounded by $n^{-\Omega(1)}$. Since $n = \omega(cv^2\Gamma^2\text{poly}(\ell))$, by lemma B.11 the first term is bounded by $o(1)$. Thus we get the theorem. $\square$

## B.4 Proof of weak recovery in spiked tensor model

The estimator $P(Y) \in \mathbb{R}^n$ we take is defined as following.

**Definition B.13** (Polynomial estimator for weak recovery). On directed complete 3-uniform hypergraph with $n$ vertices and for $i \in [n]$, we define $S_{\ell,v,i}$ as the set of all copies of hypergraphs generated in the following way:

- we construct $2\ell$ levels of distinct vertex. Level 0 contains vertice $i$ and $(v-1)/2$ vertex in addition. For $0 < t < \ell$, level $2t$ contains $v$ vertex and level $2t-1$ contains $2v$ vertex. Level $2\ell - 1$ contains $v$ vertex.

- We construct a perfect matching between level $t, t+1$ for $t \in [2\ell-2]$. For each hyperedge, 1 vertice comes from even level while 2 vertex come from odd level. Each hyperedge directs from level $t$ to level $t+1$.(They are connected in the same way as $S_{\ell,v}$ of the strong detection case, which is demonstrated in figure 3.)

- Level 0 and 1 are bipartitely connected s.t each vertice in level 0 excluding $i$ has degree 2 while vertice $i$ and vertex in level 1 has degree 1. Level $2\ell-2$ and $2\ell-1$ are bipartitely connected s.t vertex in level $2\ell-1$ have degree 2 while vertex in level $2\ell-2$ have degree 1.

Then given tensor $Y \in \mathbb{R}^{n \times n \times n}$, we have estimator $P(Y) \in \mathbb{R}^n$ where each entry is degree $(2\ell-1)v$ polynomial of entries in $Y$. For $i \in [n]$, the $i$-th entry is given by $P_i(Y) = \sum_{\alpha \in S_{\ell,v,i}} \chi_\alpha(Y)$, where $\chi_\alpha(Y)$ is multilinear polynomial basis $\chi_\alpha(Y) = \prod_{(i,j,k)\in\alpha} Y_{ijk}$ and $S_{\ell,v,i}$ is the set of hypergraphs defined above.

We prove that estimator $P(Y)$ defined in B.13 achieves constant correlation with the planted vector $x$. The proof is very similar to the proof of strong detection algorithm.

**Lemma B.14.** *In spiked tensor model, $Y = \lambda x^{\otimes 3} + W$, where the entries in $x \in \mathbb{R}^n$, $W \in \mathbb{R}^{n \times n}$ are independently sampled with zero mean and unit variance, we consider estimator $P(Y) \in \mathbb{R}^n$ defined above with $\gamma = 0.001 n^{3/2} v^{1/2} \lambda^2 = 1 + \Omega(1)$ and $\ell = O(\log_\gamma n)$. Then we have*

$$\mathbb{E}[P_i(Y)x_i] = (1 - o(1))\lambda^{2(\ell-1)v}\left(\binom{n}{v}\binom{n}{2v}\right)^{\ell-1}\binom{n}{(v-1)/2}\binom{n}{v}\frac{((2v)!)^{2\ell-1}}{2^{(3v-1)/2}}$$

*Proof.* Since $\sum_{\alpha \in S_{\ell,v,i}} \mathbb{E}[\chi_\alpha(Y)x_i] = \lambda^{(2\ell-1)v}|S_{\ell,v,i}|$, we only need to bound the size of $S_{\ell,v,i}$. Applying combinatorial arguments to the generating process of $S_{\ell,v,i}$, we have

$$|S_{\ell,v,i}| = (1 - o(1))\left(\binom{n}{v}\binom{n}{2v}\right)^{\ell-1}\binom{n}{(v-1)/2}\binom{n}{v}\frac{((2v)!)^{2\ell-1}}{2^{(3v-1)/2}}$$

. $\qquad\qquad\qquad\qquad\qquad\qquad\qquad\qquad\qquad\qquad\qquad\qquad\qquad\qquad\qquad\qquad\square$

**Lemma B.15.** *On the directed complete $3$-uniform hypergraph, for $i \in [n]$ and a set of simple hypergraph $S_{\ell,v,i}$, we consider any hypergraph $\alpha, \beta \in S_{\ell,v,i}$. Between $\alpha, \beta$, we denote the number of shared vertices(excluding vertice $i$) in level $t$ of $\alpha$ as $r_t$, the number of shared hyperedges between level $t$ and level $t+1$ of $\alpha$ as $k_t$. Further we denote $r = \sum_t r_t$ as the total number of shared vertices excluding $i$ and $k = \sum_t k_t$ as the total number of shared hyperedges.*

*Then one of the following relations must hold:*

- $2r > 3k$

- $2r = 3k$ and $\alpha \cap \beta$ is a directed hyper-path starting from vertice $i(k_t = 1$ for all $t \in [2\ell])$.

- $r = k = 0$, i.e $\alpha, \beta$ are disjoint.

- $3k \geq 2r \geq 3k - 1$ and $k_t \geq 1$ for all $t$

*Proof.* Suppose we have $2r \leq 3k - 1$. Then by degree constraint, in hypergraph $\alpha \cap \beta$, excluding vertice $i$, all other vertices have degree 2. This is only possible if for all levels $t$, vertices contained in $\alpha \cap \beta$ are connected to two hyperedges, implying that $k_t, k_{t-1} \geq 1$. Further we have $2r = 3k - 1$ in the case.

Suppose we have $2r = 3k \neq 0$ and there is $t \in [2\ell]$ such that $k_t = 0$. Then in the $\alpha \cap \beta$, exactly one vertice(excluding vertice $i$) has degree 1 and all the other vertices have degree 2. Thus there is only one level $t'$ such that $k_t = 0$. Thus all shared vertices in level $t$ have degree at most 1. This implies that there is exactly one shared vertice in level $t$. This is only possible if $\alpha \cap \beta$ is a hyperpath starting from vertice $i$. $\qquad\square$

**Lemma B.16.** *Given $Y \in \mathbb{R}^{n \times n \times n}$ sampled from spiked tensor model $Y = \lambda \cdot x^{\otimes 3} + W$, where entries in $x$ are sampled independently with zero mean and unit variance and entries in $W$ are sampled independently with zero mean and unit variance, for any $\alpha, \beta \in S_{\ell,v,i}$ with $k$ shared hyperedges and $r$ shared vertices(excluding vertice $i$), for any $\alpha, \beta \in S_{\ell,v,i}$ sharing $k$ hyperedges and $r$ vertices, we have*

$$\mathbb{E}[\chi_\alpha(Y)\chi_\beta(Y)] \leq (1 + n^{-\Omega(1)})\lambda^{-2k}\Gamma^{2r-3k}\mathbb{E}[\chi_\alpha(Y)x_i]\mathbb{E}[\chi_\beta(Y)x_i]$$

*when $\ell = O(\log n)$ and $v = O(n^{1/2 - \Omega(1)})$,*

*Proof.* This follows from direct computation. Particularly, we have

$$\mathbb{E}[\chi_\alpha(Y)\chi_\beta(Y)] = \mathbb{E}\left[\prod_{(i,j,k)\in\alpha\cap\beta}(\lambda x_i x_j x_k + W_{ijk})^2 \prod_{(i,j,k)\in\alpha\Delta\beta}(\lambda x_i x_j x_k + W_{ijk})\right]$$

$$\leq (1 + n^{-\Omega(1)})^k \lambda^{2(2\ell-1)v-2k}\mathbb{E}\left[\prod_{j\in\alpha\Delta\beta}x_j^{\deg(j,\alpha\Delta\beta)}\right]$$

$$\leq (1 + n^{-\Omega(1)})\lambda^{-2k}\Gamma^h\mathbb{E}[\chi_\alpha(Y)x_i]\mathbb{E}[\chi_\beta(Y)x_i]$$

where $\deg(j, \alpha \Delta \beta)$ represents the degree of vertex $j$ in the hypergraph $\alpha \Delta \beta$, and $h$ is the number of vertices with degree $4$ in the sub-hypergraph $\alpha \Delta \beta$. Such vertices are shared between $\alpha, \beta$ but do not incident to any shared hyperedge. By degree constraint of the shared vertices and shared hyperedges, we have $2r - 3k \geq s$. The claim thus follows. $\qquad \square$

We consider the set of hypergraph pairs $\alpha, \beta \in S_{\ell,v,i}$ such that in $\alpha$

- at level $t$ there are $r_t$ vertices(excluding vertice $i$) shared with $\beta$
- between level $t + 1$ and level $t$, there are $k_t$ hyperedges shared with $\beta$.

We denote such set of hypergraph pairs as $S_{\ell,v,i,k,r}$, where $k = \sum_t k_t, r = \sum_t r_t$. Then $r$ is just the number of shared vertices between $\alpha, \beta$ and $k$ is just the number of shared hyperedges between $\alpha, \beta$. Although we abuse the notations(since the set $S_{\ell,v,i,k,r}$ is related to $k_t, r_t$), by the following lemma we can bound the size of such set only using $k, r, \ell, v$.

**Lemma B.17.** *On directed complete hypergraph with $n$ vertices, for any set $S_{\ell,v,k,r}$ with $\ell = O(\log n)$, $v = o(n)$, $k, r \leq \ell v$, the number of hypergraph pairs contained in the set $S_{\ell,v,i,k,r}$ is bounded by*

$$|S_{\ell,v,i,k,r}| \leq |S_{\ell,v,i}|^2 n^{-r} v^{2r - 7k/2} \ell^{2r - 3k} \exp(O(r))$$

*Proof.* We first choose $\alpha \cap \beta$ and shared vertices as subgraph of hypergraph $\alpha \in S_{\ell,v,i}$, and then completing the remaining hypergraphs $\alpha \backslash \beta$ and $\beta \backslash \alpha$. If in the $\alpha$ there are $r_t$ shared vertices(excluding $i$) in level $t$, and $k_t$ shared hyperedges between level $t$ and level $t + 1$ for $t = 0, 1, \ldots, 2\ell - 1$, then the number of choices for shared vertices and $\alpha \cap \beta$ is bounded by

$$N_{\alpha \cap \beta} \leq \binom{n}{r_0} \binom{n}{r_{2\ell-1}} \binom{2r_0 + 1}{k_0} \binom{2r_{2\ell-1}}{2k_{2\ell-2}} \prod_{t=1}^{\ell-1} \left[ \binom{n}{r_{2t-1}} \binom{n}{r_{2t}} \binom{r_{2t-1}}{k_{2t-1}} \binom{r_{2t-1}}{k_{2t-2}} \right.$$
$$\left. \binom{r_{2t}}{2k_{2t}} \binom{r_{2t}}{2k_{2t-1}} \right] \prod_{t=0}^{2\ell-2} (2k_t)!$$

This is upper bounded by $\prod_{t=0}^{2\ell-2}(2k_t)! \prod_{t=0}^{2\ell-1} \binom{n}{r_t} \exp(O(r)) \leq n^r v^{k/2} \exp(O(r))$. Next we choose the remaining hypergraph $\alpha \setminus \beta$ and $\beta \setminus \alpha$ respectively. For $\alpha \setminus \beta$, we have

$$N_{\alpha \backslash \beta} = \binom{n}{\frac{v-1}{2} - r_0} \binom{n}{v - r_{2\ell-1}} \prod_{t=1}^{\ell-1} \binom{n}{2v - r_{2t-1}} \binom{n}{v - r_{2t}} \prod_{t=0}^{2\ell-2} (2(v - k_t))!$$
$$\leq |S_{\ell,v,i}| n^{-r} v^r v^{-2k} \exp(O(r))$$

Suppose there are $s_1$ degree 1 vertices in $\alpha \cap \beta$ and $s_0$ vertices shared between $\alpha, \beta$ but not contained in $\alpha \cap \beta$, denoting $s = s_0 + s_1$, then there are $\ell^s$ ways of placing $\alpha \cap \beta$ and shared vertices in hypergraph $\beta$ and the count of remaining hypergraph is also bounded by $|S_{\ell,v,i}| n^{-r} v^r v^{-2k} \exp(O(r))$. Multiplying together we will get the claim. $\qquad \square$

**Lemma B.18.** *For $t \in \{0, 1, \ldots, 2\ell - 2\}$, we define $r_t, k_t \in \{0, 1, \ldots, 2v\}$ and $r_{2\ell-1} \in \{0, 1, \ldots, v\}$ satisfying that*

- *For $t \in [2\ell - 1]$, $r_t \geq \delta(t) \max(k_{t-1}, k_t)$, where parity function $\delta(t) = 2$ if $t$ is odd and $1$ if $t$ is even.*

- *$r_0 + 1 \geq k_0$*

*We denote $k = \sum_{t=0}^{2\ell-1} k_t$ and $r = \sum_{t=0}^{2\ell-1} r_t$. We take scalars $\eta = O(\ell^{-10})$ and constant $\psi < 1$. Then for $2r \geq 3k + 1$, we have:*

$$\sum_{\substack{k_t \geq 0}} \sum_{\substack{r_t \geq \delta(t) \max(k_{t-1}, k_t) \\ 2r \geq 3k+1}} \eta^{r - 3k/2} \psi^k \leq o(1)$$

*Proof.* We note that given $k_t$ for $t \in [\ell]$, we have at most $\ell^{r-(3k-1)/2}$ choices for all $r_t$. Further we denote $k_\Delta = \sum_t |k_{t+1} - k_t|$. Then given $k_\Delta$, all $k_t$ can take at most $k_\Delta$ different values. As a result, fixing these $k_\Delta$ different values, there are $\ell^{k_\Delta}$ choices for $k_t$ for $t \in [\ell]$. Further we have $r - (3k-1)/2 \geq k_\Delta/2 \geq 1/2$. Therefore the summation is bounded by

$$\sum_{k_\Delta \geq 1} \left[ \left( \eta^{1/4}\ell^2 \right)^{k_\Delta/2} \prod_{t=1}^{k_\Delta} \left( \sum_{k_t \geq 0} \psi^{k_t} \right) \right] = o(1)$$

$\square$

**Lemma B.19** (Recovery for general spiked model). *In spiked tensor model $Y = \lambda x^{\otimes 3} + W$ with the same setting as theorem 1.3, taking $\gamma = 0.001n^{3/2}v^{1/2}\lambda^2 = 1 + \Omega(1)$ and $\ell = O(\log_\gamma n)$ in the estimator above, then if $n = \omega(v^2\Gamma^2 poly(\ell))$, we have $\frac{\mathbb{E}\langle P(Y), x \rangle}{(\mathbb{E}\|P(Y)\|^2 \, \mathbb{E}\|x\|^2)^{1/2}} = \gamma^{-O(1)}$.*

*Proof.* We need to show the estimator $P(Y) \in \mathbb{R}^n$ above achieves constant correlation with the hidden vector $x$. Equivalently, we want to show that for each $i \in [n]$

$$\left( \sum_{\alpha \in S_{\ell,v,i}} \mathbb{E}[\chi_\alpha(Y)x_i] \right)^2 = \Omega \left( \sum_{\alpha,\beta \in S_{\ell,v,i}} \mathbb{E}[\chi_\alpha(Y)\chi_\beta(Y)] \right)$$

For left hand side, we can simply apply B.14. For the right hand side, we have

$$\mathbb{E}[\chi_\alpha(Y)\chi_\beta(Y)] \leq \lambda^{-2k}\Gamma^{2r-3k} \, \mathbb{E}[\chi_\alpha(Y)x_i] \, \mathbb{E}[\chi_\beta(Y)x_i]$$

By lemma, B.17 the contribution to $\frac{\mathbb{E}[P_i^2(Y)]}{(\mathbb{E}[P_i(Y)x_i])^2}$ with respect to specific $r_t, k_t$ is bounded by

$$(n^{-r}v^r v^{-2k})^2 n^r v^{k/2}\ell^s \Gamma^{2r-3k}\lambda^{-2k} \exp(O(r)) \leq \left( \frac{n}{cv^2\ell^2\Gamma^2} \right)^{-r+3k/2} \left( cn^{-3/2}v^{-1/2}\lambda^{-2} \right)^k$$

where $c$ is constant. When $nv^{-2} = \omega(\Gamma^2 poly(\ell))$, using the lemma B.18, the dominating term is given by $r \leq \frac{3}{2}k$. For $2r = 3k-1$, by lemma B.15 we must have $k \geq \ell$, therefore for $cn^{-3/2}v^{-1/2}\lambda^{-2} < 1$ and $\ell = C \log n$ with constant $C$ large enough, the contribution is $n^{-\Omega(1)}$. For $2r = 3k$, by lemma B.15 either $k \geq 2\ell$ or $\alpha \cap \beta$ exists as a hyperpath starting from vertex $i$. The first case can be treated in the same way as $2r = 3k-1$. For the second case, the contribution is bounded by $\sum_{k=0}^{2\ell-1} \left( cn^{-3/2}v^{-1/2}\lambda^{-2} \right)^k \leq \frac{1}{1-cn^{-3/2}v^{-1/2}\lambda^{-2}}$

Therefore in all, we have $\frac{(\mathbb{E} P_i(Y)x_i)^2}{\mathbb{E} P_i^2(Y)} \geq 1 - cn^{-3/2}v^{-1/2}\lambda^{-2}$. This is $\gamma^{O(1)}$ when we have relation $cn^{-3/2}v^{-1/2}\lambda^{-2} = 1 - \Omega(1)$ and $n = \omega(v^2\Gamma^2 poly(\log(n)))$. $\square$

## B.5 Color coding method for polynomial evaluation in order-3 spiked tensor model

### B.5.1 Strong detection polynomial

For constant $v$, although the thresholding polynomial has degree $O(\log n)$, these polynomials can actually be evaluated in polynomial time via color coding method as a generalization of result in [HS17]. In the same way color coding method also improves the running time of sub-exponential time algorithms.

We first describe the evaluation algorithm for scalar polynomial, as shown in algorithm 4.

We next describe the construction of matrices $M, N, Q$ given random coloring $c : [n] \mapsto [3\ell v]$.

We have $M_{(V_1,S),(V_2,T)} = 0$ if $S \cup \{c(v) : v \in V_1\} \neq T$ or $\{c(v) : v \in V_1\}$ and $S$ are not disjoint. Otherwise $M_{(V_1,S),(V_2,T)}$ is given by $\sum_{\gamma \in \mathcal{L}_{V_1,V_2}} \chi_\gamma(Y)$ where $\mathcal{L}_{V_1,V_2}$ is the set of perfect matching induced by $V_1$ and $V_2$(each hyperedge in $\mathcal{L}_{V_1,V_2}$ direct from 1 vertice from $V_1$ to 2 vertice from $V_2$).

In the same way, We have $N_{(V_2,S),(V_1,T)} = 0$ if $S \cup \{c(v) : v \in V_2\} \neq T$ or $\{c(v) : v \in V_2\}$ and $S$ are not disjoint. Otherwise $N_{(V_2,S),(V_1,T)}$ is given by $\sum_{\gamma \in \mathcal{L}_{V_1,V_2}} \chi_\gamma(Y)$ where $\mathcal{L}_{V_1,V_2}$ is the set

---

**Algorithm 4:** Algorithm for evaluating the thresholding polynomial

---

**Data:** Given $Y \in \mathbb{R}^{n \times n \times n}$ s.t $Y = \lambda x^{\otimes 3} + W$

**Result:** $P(Y) = \sum_{\alpha \in S_{\ell,v}} \chi_\alpha(Y)$ (up to accuracy $1 + n^{-\Omega(1)}$)

$C = \exp(300\ell v)$;

**for** $i \leftarrow 1$ **to** $C$ **do**

    Sample coloring $c_i : [n] \mapsto [3\ell v]$ uniformly at random;

    Construct a matrix $M \in \mathbb{R}^{(2^{3\ell v}-1)n^v \times (2^{3\ell v}-1)n^{2v}}$, rows and columns of $M$ are indexed by $(V_1, S)$ and $(V_2, T)$ where $V_1 \in [n]^v$ and $V_2 \in [n]^{2v}$ are set of vertices while $S, T \subsetneq [3\ell v]$ are subset of colors.;

    Matrices $Q, N \in \mathbb{R}^{\mathbb{R}^{(2^{3\ell v}-1)n^{2v} \times (2^{3\ell v}-1)n^v}}$, the rows and columns of which are indexed by $(V_2, S)$ and $(V_1, T)$ where $V_1 \in [n]^v$ and $V_2 \in [n]^{2v}$ correspond to subset of vertices while $S, T \subseteq [3\ell v]$ are non-empty subset of colors.;

    Record $p_{c_i} = \frac{(3\ell v)^{3\ell v}}{(3\ell v)!} \mathsf{Tr}((MN)^{\ell-1}MQ)$;

Return $\frac{1}{C} \sum_{i=1}^{C} p_{c_i}$

---

of perfect matching induced by $V_1$ and $V_2$(each hyperedge in $\mathcal{L}_{V_1, V_2}$ direct from 2 vertice in $V_2$ to 1 vertice in $V_1$).

For matrix $Q$, the indexing and non-zero entry locations are the same as $N$. However the non-zero elements are given by $\sum_{\gamma \in \mathcal{L}_{V_2, V_1}} \chi_\gamma(Y)$ where $\mathcal{L}_{V_2, V_1}$ is the set of perfect matching induced by $V_1$ and $V_2$(each hyperedge in $\mathcal{L}_{V_2, V_1}$ direct from 1 vertice in $V_1$ to 2 vertice in $V_2$).

**Lemma B.20** (Evaluation of thresholding polynomial). *There exists a $n^{4v} \exp(O(\ell v))$-time algorithm that given a coloring $c: [n] \to [3\ell v]$(where $3\ell v$ is the number of vertices in hypergraph $\alpha \in S_{\ell,v}$) and a tensor $Y \in \mathbb{R}^{n \times n \times n}$ evaluates degree $2\ell v$ polynomial in polynomial time*

$$p_c(Y) = \sum_{\alpha \in S_{\ell,v}} \chi_\alpha(Y) F_{c,\alpha} \tag{5}$$

$$F_{c,\alpha} = \frac{(3\ell v)^{3\ell v}}{(3\ell v)!} \cdot \mathbf{1}_{c(\alpha)=[3\ell v]} \tag{6}$$

*when thresholding polynomial $P(Y)$ defined in B.5 satisfies $(\mathbb{E}\, P(Y))^2 = (1 - o(1))\,\mathbb{E}\, P^2(Y)$, we can take $\exp(O(\ell v))$ random colorings and give an accurate estimation of the thresholding polynomial by averaging $p_c(Y)$*

*Proof.* A critical observation is that for each given random coloring $c$, the algorithm above evaluates $p_c(Y)$.

We prove that averaging random coloring for $p_c(Y)$ will give accurate estimate for $P(Y)$. This follows from the same reasoning as in matrix case. First we note that given any $\alpha \in S_{\ell,v}$, the probability the all $3\ell v$ vertices in $\alpha$ are assigned different colors is given by $\frac{(3\ell v)!}{(3\ell v)^{3\ell v}}$. As a result $\mathbb{E}_c\, p_c(Y) = P(Y)$. Next, for single coloring we have

$$\mathbb{E}\, p_c^2(Y) = \sum_{\alpha, \beta \in S_{\ell,v}} \mathbb{E}[F_{c,\alpha} F_{c,\beta} \chi_\alpha(Y) \chi_\beta(Y)] \leq \exp(O(\ell v))\, \mathbb{E}\, P^2(Y) \leq \exp(O(\ell v))(\mathbb{E}\, p_c(Y))^2$$

where we use the result that $\mathbb{E}\, P^2(Y) = (1 + o(1))(\mathbb{E}\, P(Y))^2$. Therefore, by averaging $L = \exp(O(\ell v))$ independent random colorings, the variance can be reduced such that $\frac{\sum_{t=1}^{L} p_{c_t}(Y)}{L} = (1 \pm o(1))P(Y)$ w.h.p. $\qquad \square$

*Proof of theorem B.4.* Combining lemma B.3, B.12,B.20, and theorem B.3, thresholding $P(Y)$ will lead to strong detection algorithm. $\qquad \square$

### B.5.2 Evaluation of estimator for weak recovery

Next we discuss about the evaluation of polynomial estimator for weak recovery. We denote $\ell_v = \frac{3(2\ell-1)v+1}{2}$, which is the number of vertices in each hypergraph contained in the set $S_{\ell,v,i}$. The algorithm is described in 5.

---

**Algorithm 5:** Algorithm for evaluating estimation polynomial vector

**Data:** Given $Y \in \mathbb{R}^{n \times n \times n}$ s.t $Y = \lambda x^{\otimes 3} + W$
**Result:** $P(Y) \in \mathbb{R}^n$, the $i$-th entry $P_i(Y) = \sum_{\alpha \in S_{\ell,v,i}} \chi_\alpha(Y)$(up to accuracy $1 + n^{-\Omega(1)}$)

$C = \exp(100\ell_v)$;
**for** $i \leftarrow 1$ **to** $C$ **do**

> Sample coloring $c_i : [n] \mapsto [\ell_v]$ uniformly at random;
>
> Construct a matrix $M \in \mathbb{R}^{(2^{\ell_v}-1)n^v \times (2^{\ell_v}-1)n^{2v}}$, rows and columns of $M$ are indexed by $(V_1, S)$ and $(V_2, T)$ where $V_1 \in [n]^v$ and $V_2 \in [n]^{2v}$ are set of vertices while $S, T \subsetneq [\ell_v]$ are subset of colors.;
>
> Construct matrix $N \in \mathbb{R}^{(2^{\ell_v}-1)n^{2v} \times (2^{\ell_v}-1)n^v}$, the rows and columns of which are indexed by $(V_2, S)$ and $(V_1, T)$ where $V_1 \in [n]^v$ and $V_2 \in [n]^{2v}$ correspond to subset of vertices while $S, T \subseteq [\ell_v]$ are non-empty subset of colors.;
>
> Construct matrix $A \in \mathbb{R}^{n^{(v+1)/2} \times (2^{\ell_v}-1)n^{2v}}$. The rows are indexed by $(i, V_1)$ where $i \in [n]$ and $V_1 \in [n]^{(v-1)/2}$. The columns are indexed by $(V_2, S)$ where $V_2 \in [n]^{2v}$ and $S \subseteq [\ell_v]$;
>
> Construct matrix $B \in \mathbb{R}^{(2^{\ell_v}-1)n^v \times n^v}$. The rows are indexed by $(V_1, S)$ where $V_1 \in [n]^v, S \subseteq [\ell_v]$. The columns are indexed by $[n]^v$;
>
> Record $p_{c_i} = A(NM)^{\ell-1}NB\mathbf{1}$;

Return $\frac{1}{C}\sum_{i=1}^{C} p_{c_i}$

---

We construct matrices $M, N$ as defined in the strong detection algorithm(but replacing the size of color set $3\ell v$ with $\ell_v$). We need to construct two additional matrices $A \in \mathbb{R}^{n^{(v+1)/2} \times (2^{\ell_v}-1)n^{2v}}$ and $B \in \mathbb{R}^{(2^{\ell_v}-1)n^v \times n^v}$.

Then we describe how to construct matrices $A, B$. For $i \in [n]$,set of vertices $V_1$ in level 0,set of vertices $V_2$ in level 1 and set of colors $T$, denoting $S_{i,V_1,V_2}$ as the set of all possible connections between level 0 and 1, entry $A_{(i,V_1),(V_2,T)}$ is given by $\sum_{\alpha \in S_{i,V_1,V_2}} \chi_\alpha(Y)$ if $T = \{c(v) : v \in V_1 \cup v\}, v \notin V_1$ and 0 otherwise. In the same way, denoting $\mathcal{L}_{V_1,V_2}$ as all possible connections between level $2\ell-2$ and $2\ell-1$, we have entry $B_{(V_1,S),V_2}$ given by $\sum_{\alpha \in \mathcal{L}_{V_1,V_2}} \chi_\alpha(Y)$ if $S \cup \{c(v) : v \in V_1 \cup V_2\} = [\ell_v], S \cap \{c(v) : v \in V_1 \cup V_2\} = \emptyset$ and zero otherwise.

**Lemma B.21** (Evaluation of polynomial estimator). *There exists a $n^{3v}\exp(O(\ell_v))$-time algorithm that given a coloring $c:[n] \to [\ell_v]$ and a tensor $Y \in \mathbb{R}^{n \times n \times n}$ evaluates vector $p_c(Y) \in \mathbb{R}^n$ with each entry $p_{c,i}(Y)$ a polynomial of entries in $Y$*

$$p_{c,i}(Y) = \sum_{\alpha \in S_{\ell,v,i}} \chi_\alpha(Y)F_{c,\alpha}$$

$$F_{c,\alpha} = \frac{\ell_v^{\ell_v}}{\ell_v!} \cdot \mathbf{1}_{c(\alpha)=[\ell_v]}$$

*For polynomial estimator $P(Y)$ defined in 3.1, for $\ell = O(\log n), v = O(n^{1/2-\Omega(1)})$, if $0.001\lambda^2 n^{3/2}v^{1/2} > 1$, then we can take $\exp(O(\ell v))$ random colorings and give an accurate estimation of the estimation polynomial $P_i(Y)$ by averaging $p_{c,i}(Y)$. When $\lambda^2 n^{3/2} = \omega(1)$, we have $n^{3+o(1)}$ time algorithm for evaluation.*

*Proof.* The critical observation is that $p_{c,i}(Y)$ can be obtained from vector $\xi = A(NM)^{\ell-1}NB\mathbf{1}(\mathbf{1} \in \mathbb{R}^{n^v}$ is all-1 vector), by summing up all rows in $\xi$ indexed by $(i, \cdot)$.

By the same argument as in the strong detection algorithm, we can obtain accurate estimate of $P(Y)$ by averaging $\exp(O(\ell))$ random colorings when weak recovery is achieved. Therefore, the estimator can be evaluated in time $n^{O(v)}$ when $0.001\lambda^2 n^{3/2} v^{1/2} > 1$.

Moreover, when $\lambda^2 n^{3/2} = \omega(1)$, it's enough to take $v = 1$ and $\ell = o(\log n)$. Thus $\xi = A(NM)^{\ell-1}NB\mathbf{1}$ can be evaluated in $n^3 \exp(O(\ell)) = n^{3+o(1)}$ time by recursively executing matrix-vector multiplication. Therefore the polynomial estimator achieving strong recovery can be evaluated in time $n^{3+o(1)}$. $\qquad\qquad\qquad\qquad\qquad\qquad\qquad\qquad\qquad\qquad\qquad\qquad\qquad\qquad\Box$

*Proof of Theorem 1.3.* The estimator $\hat{x}$ can simply be given by $P(Y)$. By choosing $v$ such that $\gamma$ is a large enough constant, the correlation $\delta$ can be achieved by lemma B.19. Further according to lemma B.21 by the color-coding method, $P(Y)$ can be evaluated in time $n^{O(v)}$. These proves the claim in theorem 1.3. $\qquad\qquad\qquad\qquad\qquad\qquad\qquad\qquad\qquad\qquad\qquad\qquad\qquad\qquad\qquad\Box$

## B.6 Equivalence between strong and weak recovery

We focus on the polynomial time regime, i.e $v$ is constant. Our algorithm for weak recovery can achieve any recovery rate $1 - \Omega(1)$ by increasing $v$ when $n^{-3/2}v^{-1/2}\lambda^{-2} = \Theta(1)$. But achieving strong recovery with rate $1 - o(1)$ requires $n^{3/2}v^{1/2}\lambda^2 = \omega(1)$. In this section, we show that under some mild conditions, combining concentration argument and 'all or nothing' amplification, we can actually obtain strong recovery algorithm when $n^{-3/2}v^{-1/2}\lambda^{-2} = \Theta(1)$.

For this we need an assumption on the tensor injective norm. The injective norm of an order-$p$ tensor $W \in (\mathbb{R}^n)^{\otimes p}$ is defined as

$$\|W\|_{\text{inj}} = \max_{\|u^{(1)}\|=\cdots=\|u^{(p)}\|=1} \sum_{i_1,\ldots,i_p} W_{i_1,\ldots,i_p} u_{i_1}^{(1)} \cdots u_{i_p}^{(p)}$$

**Theorem B.22** (Strong recovery). *In general spiked tensor model $Y = \lambda \cdot x^{\otimes 3} + W$, we take estimation vector $y \in \mathbb{R}^n$, which is given by estimator 3.1 with $\gamma = 0.001 n^{3/2} v^{1/2} \lambda^2 = 1 + \Omega(1)$ and $\ell = O(\log_\gamma n)$. If injective norm of $W$ is $O(n^{3/4-\Omega(1)})$ with high probability, then for constant $v$ if $v = O(n^{1/2-\Omega(1)})$, we have $\hat{x} \in \mathbb{R}^n$ s.t $\hat{x}_i = \sum_{j_1,j_2} Y_{i,j_1,j_2} y_{j_1} y_{j_2}$ achieves strong recovery, i.e we have $\langle \frac{\hat{x}}{\|\hat{x}\|}, \frac{x}{\|x\|} \rangle = 1 - n^{-\Omega(1)}$ with high probability*

**Remark**: We use assumption that the injective norm of $W$ is $n^{3/4-\Omega(1)}$ w.h.p. Now we interpret this assumption. For Gaussian tensor the injective norm of $W$ is $O(n^{1/2})$ with high probability. For general tensor, this assumption is weaker than finite bounded moments.

**Lemma B.23.** *For a tensor $W \in (\mathbb{R}^n)^{\otimes 3}$, the injective norm is $O(\sqrt{n})$ with high probability if the absolute value of entries in $W$ are all bounded by $B = o(n^{1/4})$ with high probability.*

For arbitrarily small $\varepsilon = \Omega(1)$, if entries $W_{ijk}$ have bounded $12 + \varepsilon$-th moment, then by Markov inequality and union bound, the entries in $W$ are all bounded by $B = o(n^{1/4-\Omega(1)})$ with high probability.

*Proof.* For fixed unit vectors $x, y, z$ and $T = \sum_{ijk} W_{ijk} x_i y_j z_k$, by Hoeffding bound we have

$$\Pr[T \geq tB'] \leq \exp(-ct^2)$$

where $c$ is constant and $B'$ is the maximum absolute value of entries in $W$. We denote the event $B' \leq B$ as $\mathcal{A}$. By assumption $\mathcal{A}$ happens with high probability. The $\varepsilon$-net of unit sphere $\mathcal{S}_{\varepsilon,n}$ has size at most $(O(1)/\varepsilon)^n$ (see e.g Tao's random matrix book [Tao12] lemma 2.3.4). Thus the size of set $\mathcal{B} = \{(x,y,z)|x,y,z \in \mathcal{S}_{\varepsilon,n}\}$ is bounded by $(O(1)/\varepsilon)^{3n}$. Taking union bound on this set we have

$$\Pr[\max_{(x,y,z)\in\mathcal{B}} \langle W, x \otimes y \otimes z \rangle \geq Bt] \leq \exp(c_2 n)\exp(-ct^2) + o(1)$$

where $c_2$ is constant. Taking $t = C\sqrt{n}$ with constant $C$ large enough, it follows that $\max_{(x,y,z)\in\mathcal{B}} \langle W, x \otimes y \otimes z \rangle = O(n^{3/4-\Omega(1)})$ with high probability.

Finally we have

$$\|W\|_{\text{inj}} = \max_{\|x\|=1, \|y\|=1, \|z\|=1} \langle W, x \otimes y \otimes z \rangle = \langle W, x^* \otimes y^* \otimes z^* \rangle$$

, where $x^*, y^*, z^*$ is the maximizer. By definition of $\varepsilon$-net, we can find $(\tilde{x}, \tilde{y}, \tilde{z}) \in \mathcal{B}$ such that $\|\tilde{x} - x^*\|, \|\tilde{y} - y^*\|, \|\tilde{z} - z^*\| \leq \varepsilon$. Thus $\langle W, x^* \otimes y^* \otimes z^* - \tilde{x} \otimes \tilde{y} \otimes \tilde{z} \rangle \leq 4\varepsilon \langle W, x^* \otimes y^* \otimes z^* \rangle$. For small $\varepsilon$, we have $\|W\|_{\text{inj}} \leq 2 \max_{(x,y,z) \in \mathcal{B}} \langle W, x \otimes y \otimes z \rangle = o(n^{3/4 - \Omega(1)})$ with high probability. This proves the claim. $\qquad\square$

This proof of theorem naturally follows from the weak recovery result above and the following two lemmas.

**Lemma B.24** (All or nothing phenomenon). *In general spiked tensor model $Y = \lambda \cdot x^{\otimes 3} + W$, if the injective norm of tensor $W$ is $O(n^{3/4 - \Omega(1)})$, then if we have unit norm estimator $y \in \mathbb{R}^n$ satisfying $\langle y, x/\|x\| \rangle = \Omega(1)$ w.h.p, then let $\hat{x} \in \mathbb{R}^n$ with $\hat{x}_i = \sum_{j_1, j_2} Y_{i, j_1, j_2} y_{j_1} y_{j_2}$, we have w.h.p*

$$\left\langle \frac{\hat{x}}{\|\hat{x}\|}, \frac{x}{\|x\|} \right\rangle^2 = 1 - n^{-\Omega(1)}$$

This lemma follows the same proof as appendix D in [WAM19]

**Lemma B.25** (Concentration property). *For the above estimator $P(Y) \in \mathbb{R}^n$ in definition 3.1, when we take $\ell, v$ as described in theorem B.19, we have*

$$\mathbb{E}\langle P(Y), x \rangle^2 = (1 + o(1)) \left( \mathbb{E}\langle P(Y), x \rangle \right)^2$$

We now prove lemma B.25. The proof is very similar to the proof of strong detection in appendix B.3.

*Proof of Lemma B.25.* We denote $S_{\ell,v} = \cup_{i \in [n]} S_{\ell,v,i}$. For $\alpha \in S_{\ell,v}$, we denote $x_\alpha = x_i$ if $\alpha \in S_{\ell,v,i}$. Then equivalently we want to show.

$$\sum_{\alpha, \beta \in S_{\ell,v}} \mathbb{E}[\chi_\alpha(Y) \chi_\beta(Y) x_\alpha x_\beta] \leq (1 - o(1)) \sum_{\alpha, \beta \in S_{\ell,v}} \mathbb{E}[\chi_\alpha(Y) x_\alpha][\chi_\beta(Y) x_\beta]$$

Since $\left( \sum_{\alpha \in S_{\ell,v}} \mathbb{E}[\chi_\alpha(Y) x_\alpha] \right) = \lambda^{(2\ell-1)v} |S_{\ell,v}|$, we only need to bound the size of $S_{\ell,v}$. Applying combinatorial arguments to the generating process of $S_{\ell,v}$, we have

$$|S_{\ell,v}| = (1 - o(1)) \left( \binom{n}{v} \binom{n}{2v} \right)^{\ell-1} \binom{n}{(v+1)/2} \binom{n}{v} \frac{((2v)!)^{2\ell-1}}{2^{(3v-1)/2}}$$

.

Now we bound the right hand side. First for the case that $\alpha, \beta$ are disjoint($r = 0$), we have $\mathbb{E}[\chi_\alpha(Y) \chi_\beta(Y)] = \mathbb{E}[\chi_\alpha(Y)] \mathbb{E}[\chi_\beta(Y)]$. Thus we have

$$\sum_{\alpha, \beta \in S_{\ell,v}} \mathbb{E}[\chi_\alpha(Y) \chi_\beta(Y) x_\alpha x_\beta] \leq (1 - o(1)) \sum_{\alpha, \beta \in S_{\ell,v}} \mathbb{E}[\chi_\alpha(Y) x_\alpha][\chi_\beta(Y) x_\beta]$$

For each pair of $\alpha, \beta \in S_{\ell,v}$ sharing $k$ hyperedges and $r$ vertices, we have

$$\mathbb{E}[\chi_\alpha(Y) \chi_\beta(Y)] = (1 + n^{-\Omega(1)})^k \lambda^{2(2\ell-1)v - 2k} \mathbb{E}\left[ \prod_{j \in \alpha \Delta \beta} x_j^{\deg(j, \alpha \Delta \beta)} \right]$$

$$\leq \lambda^{-2k} \Gamma^{O(2r - 3k)} \mathbb{E}[\chi_\alpha(Y)] \mathbb{E}[\chi_\beta(Y)]$$

where $\deg(j, \alpha \Delta \beta)$ represents the degree of vertex $j$ in hypergraph $\alpha \Delta \beta$.

Next we bound the number of hypergraph pairs $(\alpha, \beta)$ sharing specified vertices and hyperedges. For this, we first choose $\alpha \cap \beta$ and shared vertices as a subgraph of a hypergraph $\alpha$ contained in $S_{\ell,v}$. We consider the following case:

- in level $t \in [2\ell - 1]$ of $\alpha$ there are $r_t$ shared vertices

- between the levels $t$ and $t+1$ of $\alpha$ there are $k_t$ shared hyperedges.

Then the number of choices for these shared hyperedges and vertices is bounded by $N_{\alpha \cap \beta}$

$$\binom{n}{r_0}\binom{n}{r_{2\ell-1}}\binom{2r_0}{k_0}\binom{2r_{2\ell-1}}{2k_{2\ell-2}}\prod_{t=1}^{\ell-1}\binom{n}{r_{2t-1}}\binom{n}{r_{2t}}\binom{r_{2t-1}}{k_{2t-1}}\binom{r_{2t-1}}{k_{2t-2}}\binom{r_{2t}}{2k_{2t}}\binom{r_{2t}}{2k_{2t-1}}\prod_{t=0}^{2\ell-2}(2k_t)!$$

By Strling's approximation, this is upper bounded by $\prod_{t=0}^{2\ell-2}(2k_t)! \prod_{t=0}^{2\ell-1}\binom{n}{r_t}\exp(O(r))$. Next we choose the remaining hypergraph $\alpha \setminus \beta$ and $\beta \setminus \alpha$ respectively. For $\alpha \setminus \beta$, we have

$$
\begin{aligned}
N_{\alpha \setminus \beta} &= \binom{n}{\frac{v+1}{2}-r_0}\binom{n}{v-r_{2\ell-1}}\prod_{t=1}^{\ell-1}\binom{n}{2v-2r_{2t-1}}\binom{n}{v-r_{2t}}\prod_{t=0}^{2\ell-2}(2(v-k_t))! \\
&\leq |S_{\ell,v,i}|n^{-r}v^r v^{-2k}\exp(O(r))
\end{aligned}
$$

For bounding the number of choices for $\beta$, suppose $\alpha, \beta$ share $s$ vertices with degree 0 or 1 in the subgraph $\alpha \cap \beta$. Then there are $\ell^s$ ways of embedding $\alpha \cap \beta$ and shared vertices in hypergraph $\beta$. Further the number of choices for the remaining hypergraph is also bounded by $|S_{\ell,v}|n^{-r}v^r v^{-2k}\exp(O(r))$. Therefore the contribution to $\frac{\mathbb{E}[\langle P(Y),x\rangle^2]}{(\mathbb{E}[\langle P(Y),x\rangle])^2}$ with respect to specific $r_t, k_t$ is bounded by

$$(n^{-r}v^r v^{-2k})^2 n^r v^{k/2}\ell^s \Gamma^{O(2r-3k)}\lambda^{-2k}\exp(O(r)) \leq \left(\frac{n}{cv^2\ell^2\Gamma^{O(1)}}\right)^{-r+3k/2}\left(cn^{-3/2}v^{-1/2}\lambda^{-2}\right)^k$$

where $c$ is constant. By lemma B.18, when $v = \omega(n^{1/2-\Omega(1)})$ the dominating term is given by $r = \frac{3}{2}k$. In this case, we must have $k \geq \ell$. For $cn^{-3/2}v^{-1/2}\lambda^{-2} < 1$ and $\ell = C\log n$ with constant $C$ large enough, the contribution is $n^{-\Omega(1)}$.

Therefore in all, we have $\frac{(\mathbb{E}\langle P(Y),x\rangle)^2}{\mathbb{E}\langle P(Y),x\rangle^2} = 1-o(1)$ when we have relation $cn^{-3/2}v^{-1/2}\lambda^{-2} = 1-\Omega(1)$ and $v = O(n^{1/2-\Omega(1)})$. $\qquad\square$

*Proof of Theorem B.22.* As a result of lemma B.25, by Chebyshev's inequality we have $\langle P(Y),x\rangle = (1 \pm o(1))\mathbb{E}\langle P(Y),x\rangle$ w.h.p. Combined with correlation in expectation and Markov inequality, we have $\frac{\langle P(Y),x\rangle}{\|x\|\|P(Y)\|} \geq \varepsilon \cdot \Omega(1)$ with probability $1-\varepsilon$. We set $\varepsilon = n^{-\Omega(1)}$ and take estimator $z \in \mathbb{R}^n$ with $z_k = \sum_{ijk}Y_{ijk}P_i(Y)P_j(Y)$. Then according to the theorem B.24, we get the strong recovery guarantee. $\qquad\square$

## C   Higher order general spiked tensor model

For clarity, we discuss the algorithms for order-3 spiked tensor model and spiked Wigner model above. Such claims can be generalized to any constant order tensor without difficulties.

### C.1   Strong detection and weak recovery in order-3 spiked tensor model

For spiked tensor model, we define strong detection problem. Specifically given tensor $Y$ sampled from general spiked tensor model, we want to detect whether it's sampled with $\lambda = 0$ or large $\lambda$ with high probability.

**Definition C.1** (Strong detection). Given tensor $Y \in (\mathbb{R}^n)^{\otimes p}$ sampled from

- Planted distribution $\mathbb{P}$: the random tensor $Y \in (\mathbb{R}^n)^{\otimes p}$ is sampled as $Y = \lambda \cdot x^{\otimes p} + W$, where $x \in \mathbb{R}^n$ is a random vector s.t $\mathbb{E}\,x_i = 0$, $\mathbb{E}\,x_i^2 = 1$, and $W \in (\mathbb{R}^n)^{\otimes p}$ has independent, zero mean and unit variance entries.

- Null distribution $\mathbb{Q}$: where random tensor $Y \in (\mathbb{R}^n)^{\otimes p}$ has independent, zero-mean and unit variance entries.

with equal probability, we need to find a function of entries in $Y$: $f(Y) \in \{0,1\}$ such that

$$\frac{1}{2}\mathbb{P}[f(Y)=1] + \frac{1}{2}\mathbb{Q}[f(Y)=0] = 1-o(1)$$

We also define the notion of weak recovery and strong recovery in spiked tensor model.

**Definition C.2.** In spiked tensor model $Y = \lambda \cdot x^{\otimes p} + W$ where $x \in \mathbb{R}^n$ is a random vector s.t $\mathbb{E}\, x_i = 0$, $\mathbb{E}\, x_i^2 = 1$, and $W \in (\mathbb{R}^n)^{\otimes p}$ has independent, zero mean and unit variance entries, We define that the estimator $\hat{x}(Y) \in \mathbb{R}^n$ achieves weak recovery if $\mathbb{E}\langle \hat{x}(Y), x\rangle^2 \geq \Omega\left(\left(\mathbb{E}\|\hat{x}(Y)\|^2 \,\mathbb{E}\|x\|^2\right)\right)$. Further we define that $\hat{x}(Y) \in \mathbb{R}^n$ achieves strong recovery if $|\langle \hat{x}(Y), x\rangle| \geq (1 - o(1))\|\hat{x}(Y)\|\|x\|$ with high probability.

**Remark**: The weak recovery here can be equivalently defined as that with constant probability $\left\langle \frac{\hat{x}(Y)}{\|\hat{x}(Y)\|}, \frac{x}{\|x\|} \right\rangle^2 \geq \Omega(1)$. The equivalence follows by Markov inequality.

**Theorem C.3.** *Let $x \in \mathbb{R}^n$ be a random vector with independent, mean-zero entries having $\mathbb{E}\, x_i^2 = 1$ and $\Gamma = \mathbb{E}\, x_i^4 \leq n^{o(1)}$. Let $\lambda > 0$ and $Y = \lambda \cdot x^{\otimes p} + W$, where $W \in (\mathbb{R}^n)^{\otimes p}$ has independent, mean-zero and unit variance entries. Then when $v = O(n^{1/2 - \Omega(1)})$ and $c_p n^{p/4} v^{(p-2)/4}\lambda = 1 + \Omega(1)$, there is $n^{O(pv)}$ time algorithm achieving strong detection.*

**Theorem C.4.** *Let $x \in \mathbb{R}^n$ be a random vector with independent, mean-zero and unit variance entries and $\Gamma = \mathbb{E}\, x_i^4 \leq n^{o(1)}$. Let $\lambda > 0$ and $Y = \lambda \cdot x^{\otimes p} + W$, where $W \in (\mathbb{R}^n)^{\otimes p}$ has independent, mean-zero entries with $\mathbb{E}\, W_{i_1, i_2, \ldots, i_p}^2 = 1$. Then when $v = O(n^{1/2 - \Omega(1)})$ and $\delta = c_p \lambda n^{p/4} v^{(p-2)/4} - 1 \geq \Omega(1)$, there is $n^{O(pv)}$ time algorithm giving unit norm estimator $\hat{x}$ s.t $\langle \hat{x}, \frac{x}{\|x\|}\rangle^2 \geq \delta^{O(1)}$.*

Specifically this leads to polynomial time algorithm when $\lambda = \Omega(n^{-p/4})$.

When the order-$p$ is odd, the analysis is very similar to the one for the case $p = 3$. Therefore we mainly talk about case where order $p$ is even and prove the guarantee of the theorem.

## C.2 Strong detection algorithm for even p

For strong detection we propose the following thresholding polynomial

**Definition C.5** (thresholding polynomial for even p)**.** On directed complete $p$-uniform hypergraph with $n$ vertices, we define $S_{\ell, v}$ as the set of all copies of 2-regular hypergraphs generated in the following way

- We construct $\ell$ levels of vertices, with each level containing $pv/2$ vertices.

- For $t \in [\ell]$, we connect a perfect matching with $v$ hyperedges between level $t$ and level $t+1$. Each hyperedge directs from $p/2$ vertices in level $t$ to $p/2$ vertices in level $t + 1$.

- Finally we similarly connect a perfect matching with $v$ hyperedges between level $\ell - 1$ and level $0$. Each hyperedge directs from $p/2$ vertices in level $0$ to $p/2$ vertices in level $\ell - 1$

The thresholding polynomial is given by $P(Y) = \sum_{\alpha \in S_{\ell,v}} \chi_\alpha(Y)$ where $\chi_\alpha(Y)$ is the Fourier basis associated with hypergraph $\alpha$: $\chi_\alpha(Y) = \sum_{(i_1, i_2, \ldots, i_p) \in \alpha} Y_{i_1, i_2, \ldots, i_p}$.

**Lemma C.6.** *Suppose we have $\gamma = c_p v^{(p-2)/2}\lambda^2 n^{p/2} = 1 + \Omega(1)$ with $c_p$ small enough constant related to p, then taking $\ell = O(\log_\gamma n)$, the projection of likelihood ratio $\hat{\mu}(Y)$ with respect to $S_{\ell,v}$ is $\omega(1)$ when $v = O(n^{1/2 - \Omega(1)})$.*

*Proof.* Given fixed vertices in level $t$ and level $t+1$, we have $((pv/2)!)^2/v!$ choices for the hyperedges between level $t$ and level $t + 1$. Therefore we have

$$S_{\ell,v} = (1 - o(1))\left(\frac{(pv/2)!}{v!}\right)^\ell n^{\ell pv/2}$$

. Therefore by Strling's approximation this implies $S_{\ell,v} \geq v^{(p-2)v/2}\lambda^{2v} n^{\ell pv/2} \exp(-O(v))$. Therefore we have

$$\sum_{\alpha \in S_\ell} \mu_\alpha^2 \geq \lambda^{2\ell v} n^{\ell pv/2} v^{(p-2)v/2} \exp(-O(v)) = \omega(1)$$

when we have $\ell, v$ as described. $\square$

**Lemma C.7.** *Suppose we have* $\gamma = c_p v^{(p-2)/2} \lambda^2 n^{p/2} = 1 + \Omega(1)$, *and* $\ell = O(\log_\gamma n)$ *with* $c_p$ *small enough constant related to p. If* $n/v^2 = \omega(poly(\Gamma \ell))$ *then we have the following concentration property:*

$$\mathbb{E}\, P^2(Y) = (1 + o(1))(\mathbb{E}\, P(Y))^2$$

*Proof.* We consider $\alpha, \beta \in S_{\ell,v}$. We first choose $\alpha \cap \beta$ and shared vertices as subgraph of $\alpha$ and then select the remaining hypergraph of $\alpha, \beta$. As before, we have $\mathbb{E}[\chi_\alpha(Y)\chi_\beta(Y)] \leq \lambda^{4\ell v - 2k}\Gamma^{2r-pk}$, where $r$ is the number of shared vertices, $k$ is the number of shared hyperedges, and $\Gamma = \mathbb{E}[x_i^4] = n^{o(1)}$. Considering shared vertices and hyperedges in $\alpha$, if there are $r_t$ shared vertices in level $t$ and $k_t$ shared hyperedges between level $t$ and level $t+1$, then there are

$$N_{\alpha \cap \beta} \leq \prod_{t=0}^{\ell} \binom{n}{r_t} \binom{r_t}{\frac{pk_t}{2}} \binom{r_t}{\frac{pk_{t-1}}{2}} \frac{(((pk_t/2)!)^2}{k_t!}$$

$$\leq n^r v^{(p-2)k/2} \exp(O(r))$$

such subgraphs. On the other hand, the number of choices for the remaining hypergraph of $\alpha$ is bounded by

$$N_{\alpha - \beta} = \prod_{t=0}^{\ell-1} \binom{n}{\frac{pv}{2} - r_t} \frac{((p(v-k_t)/2)!)^2}{(v-k_t)!}$$

$$= |S_{\ell,v}| n^{-r} v^{r - pk + k} \exp(O(r))$$

Then we consider choices for $\beta$. Denote the number of degree-1 vertices in $\alpha \cap \beta$ as $s_1$ and the number of shared vertices not contained in $\alpha \cap \beta$ as $s_0$. Let $s = s_0 + s_1$, then there are at most $\ell^s$ ways of putting $\alpha \cap \beta$ in $\beta$. For the same reasoning, number of ways for choosing the remaining hypergraph of $\beta$ is also bounded by $|S_{\ell,v}| n^{-r} v^{r - pk + k} \exp(O(r))$. Therefore, with respect to fixed number of vertices and hyperedges $r_t, k_t$ in each level of $\alpha$, the total number of such hypergraph pairs $S_{r,k,\ell,v}$ is bounded by

$$|S_{\ell,v}|^2 n^{-r} v^{2r - 2pk + 2k} \ell^s v^{(p-2)k/2} \exp(O(r))$$

Therefore the corresponding contribution is given by

$$\sum_{\alpha,\beta \in S_{r,k,\ell,v}} \frac{\mathbb{E}\, \chi_\alpha(Y)\chi_\beta(Y)}{(\mathbb{E}\, P(Y))^2} \leq n^{-r} v^{2r - 2pk + 2k} \ell^s v^{(p-2)k/2} \lambda^{-2k}\Gamma^{2r-pk} \exp(O(r))$$

Since we have $2r - s \geq pk$ by degree constraints, this is bounded by

$$c^{pk/2} n^{-pk/2} v^{-(p-2)k/2} \lambda^{-2k} \ell \left(\frac{n}{cv^2\Gamma^2}\right)^{-r+pk/2}$$

where $c$ is constant. Summing up for $\alpha, \beta$ with respect to different $r_t, k_t$ and combining the fact that if $r = pk/2$ then $k \geq 2\ell$, we have

$$\sum_{r_t, k_t} c^{pk/2} n^{-pk/2} v^{-(p-2)k/2} \lambda^{-2k} \ell \left(\frac{n}{cv^2\Gamma^2}\right)^{-r+pk/2}$$

$$= \sum_{\substack{k_t \geq 0 \\ 2r \geq pk+1}} \sum_{r_t \geq \delta(t)\max(k_t,k_{t-1})} c^{pk/2} n^{-pk/2} v^{-(p-2)k/2} \lambda^{-2k} \ell \left(\frac{n}{cv^2\Gamma^2}\right)^{-r+pk/2} +$$

$$\sum_{k_t \geq 1} \left(c^{p/2} n^{-p/2} \lambda^{-2} v^{-(p-2)/2}\right)^{\sum k_t}$$

When $\gamma = c_p n^{-p/2} \lambda^{-2} v^{-1/2} > 1$ and $\ell = C \log_\gamma n$ with constant $C$ large enough, the second term is bounded by $n^{-\Omega(1)}$. For the first term we note that given $k_t$ for $t \in [\ell]$, we have $\ell^{r-pk/2}$ choices for $r_t$. We denote $k_\Delta = |k_{t+1} - k_t|$. Then we have $k_\Delta = O(r - 3k/2)$. Then given $k_\Delta$ we have at

most $k_\Delta$ different values for $k_t$. As a result fixing these $k_\Delta$ different values, there are $\ell^{k_\Delta}$ choices for $k_t$ for $t \in [\ell]$. Therefore the first term is bounded by

$$\sum_{k_\Delta \geq 0} \left[ \left( \frac{n}{cv^2 \text{poly}(\Gamma \ell)} \right)^{-\max(1/2, k_\Delta)} \prod_{t=1}^{k_\Delta} \left( \sum_{k_t \geq 0} (c^{p/2} n^{-p/2} \lambda^{-2} v^{-(p-2)/2})^{k_t} \right) \right] = o(1)$$

In all, we have $\sum_{\alpha, \beta \in S_{\ell,v}} \mathbb{E}[\chi_\alpha(Y) \chi_\beta(Y)] \leq (1 + o(1))(\mathbb{E}\, P(Y))^2$ $\qquad\square$

Next we show that the running time can be improved using color-coding method. We describe the evaluation algorithm 6. We describe the matrices $M, N$ used in the algorithm. The rows and columns

---

**Algorithm 6:** Algorithm for evaluating the thresholding polynomial

---

**Data:** Given $Y \in (\mathbb{R}^n)^{\otimes p}$ s.t $Y = \lambda x^{\otimes p} + W$
**Result:** $P(Y) = \sum_{\alpha \in S_{\ell,v}} \chi_\alpha(Y)$(up to accuracy $1 \pm n^{-\Omega(1)}$)
**for** $i \leftarrow 1$ **to** $C$ **do**
    Sample coloring $c_i : [n] \mapsto [\ell v]$ uniformly at random;
    Construct a matrix $M, N \in \mathbb{R}^{(2^{p\ell v} - 1) n^{pv/2} \times (2^{p\ell v} - 1) n^{pv/2}}$;
    Record $p_{c_i} = \frac{(p\ell v)^{p\ell v}}{(p\ell v)!} \text{Tr}(M^{\ell-1} N)$;
Return $\frac{1}{C} \sum_{i=1}^{C} p_{c_i}$

---

of $M$ are indexed by $(V_1, S)$ and $(V_2, T)$ where $V_1 \in [n]^{pv/2}$ and $V_2 \in [n]^{pv/2}$ correspond to sets of labels of vertices while $S, T \subsetneq [\ell v]$ correspond to subsets of colors. We have $M_{(V_1,S),(V_2,T)} = 0$ if $S \cup \{c(v) : v \in V_1\} \neq T$ or $\{c(v) : v \in V_1\}$ and $S$ are not disjoint. Otherwise $M_{(V_1,S),(V_2,T)}$ is given by $\sum_{\gamma \in S_{V_1,V_2}} \chi_\gamma(Y)$ where $S_{V_1,V_2}$ is the set of perfect matching induced by $V_1$ and $V_2$(each hyperedge in $S_{V_1,V_2}$ direct from $p/2$ vertices from $V_1$ to $p/2$ vertices from $V_2$).

For matrix $N$, the indexing is the same as $M$. The entry $N_{(V_1,S),(V_2,T)} = 0$ if $S \cup \{c(v) : v \in V_1\} \neq T$ or $\{c(v) : v \in V_1\}$ and $S$ are not disjoint. Otherwise $N_{(V_1,S),(V_2,T)}$ is given by $\sum_{\gamma \in S_{V_2,V_1}} \chi_\gamma(Y)$ where $S_{V_2,V_1}$ is the set of perfect matching induced by $V_2$ and $V_1$(each hyperedge in hypergraph $\alpha \in S_{V_2,V_1}$ directs from $p/2$ vertices from $V_2$ to $p/2$ vertices from $V_1$).

**Lemma C.8** (Evaluation of thresholding polynomial). *There exists a $n^{O(v)}$-time algorithm that given a coloring $c:[n] \to [p\ell v]$(where $p\ell v$ is the number of vertices in hypergraph $\alpha \in S_{\ell,v}$) and a tensor $Y \in (\mathbb{R}^n)^{\otimes p}$ evaluates degree $2\ell v$ polynomial below in polynomial time*

$$p_c(Y) = \sum_{\alpha \in S_{\ell,v}} \chi_\alpha(Y) F_{c,\alpha} \tag{7}$$

$$F_{c,\alpha} = \frac{(p\ell v)^{p\ell v}}{(p\ell v)!} \cdot \mathbf{1}_{c(\alpha) = [p\ell v]} \tag{8}$$

*when thresholding polynomial $P(Y)$ defined in C.5 satisfies $(\mathbb{E}\, P(Y))^2 = (1 - o(1)) \mathbb{E}\, P^2(Y)$, we can take $\exp(O(\ell v))$ random colorings and give an accurate estimation of the thresholding polynomial by averaging $p_c(Y)$.*

*Proof.* The observation is that $p_c(Y)$ is just given by $(p\ell v)^{p\ell v}/(p\ell v)!$ times the trace of $(M)^{\ell-1} N$. For any coloring $c$, the evaluation of $p_c(Y)$ can be done in time $n^{3pv/2} \exp(O(\ell v))$.

Next we prove that averaging random coloring for $p_c(Y)$ will give accurate estimation for $P(Y)$ in the detection algorithm. First we note that $\mathbb{E}_c\, p_c(Y) = P(Y)$. Next, for single coloring we have

$$\mathbb{E}\, p_c^2(Y) = \sum_{\alpha, \beta \in S_{\ell,v}} \mathbb{E}[F_{c,\alpha} F_{c,\beta} \chi_\alpha(Y) \chi_\beta(Y)] \leq \exp(O(\ell v)) \mathbb{E}\, P^2(Y) \leq \exp(O(\ell v)) \mathbb{E}[(\mathbb{E}_c\, p_c(Y))^2]$$

where we use the result that $\mathbb{E}\, P^2(Y) = (1 + o(1))(\mathbb{E}\, P(Y))^2$. Therefore, by averaging $L = \exp(O(\ell v))$ random colorings, the variance can be reduced such that $\frac{\sum_{t=1}^{L} p_{c_t}(Y)}{L} = (1 \pm o(1)) P(Y)$ w.h.p. $\qquad\square$

**Remark**: When $\lambda = \Omega(n^{-p/4})$, we can take $c_p \lambda n^{p/4} v > 1$ with $c_p$ being small enough constant dependent on order $p$. This leads to $O(n^{2pv})$ time algorithm with constant $v$. When $\lambda = \omega(n^{-p/4})$, we can simply take $P(Y) = \sum_{i_1 < i_2 < ... < i_p} Y_{i_1, i_2, ..., i_p} Y_{i_p, i_{p-1}, ..., i_1}$. Obviously such polynomial can be evaluated in linear time.

*Proof of Theorem C.3.* Combining lemma C.6 and the concentration property proved in lemma C.7, we get a strong detection algorithm by thresholding $P(Y)$. Using the lemma C.8, such polynomial can be evaluated in $n^{O(pv)}$ time. This proves the theorem. □

**Remark**: When $p = 2$ and $v = 1$, this gives polynomial time strong detection algorithm for the spiked matrix model when $\lambda = (1 + \Omega(1))n^{-1/2}$.

### C.3 Weak recovery algorithm for even p

For weak recovery we want to propose estimator $P(Y) \in \mathbb{R}^{n \times n}$ such that

$$\frac{\left(\mathbb{E}\left\langle P(Y), xx^\top \right\rangle\right)^2}{\left(\mathbb{E}\, P(Y)\|_F^2 \|\mathbb{E}\|xx^\top\|_F^2\right)} = \Omega(1)$$

For even $p$, it can always be decomposed into two odd numbers $p_1$ and $p_2$ s.t $p = p_1 + p_2$. Let $p_1, p_2$ be such a pair of odd numbers that minimizes $|p_2 - p_1|$. Then we define the estimation vector for weak recovery as following:

**Definition C.9** (Estimator for even-$p$ weak recovery). Given tensor $Y \in (\mathbb{R}^n)^{\otimes p}$, we have estimator $P(Y) \in \mathbb{R}^{n \times n}$ where each entry $P_{ij}(Y)$ is a degree $(2\ell-1)v$ polynomial given by $\sum_{\alpha \in S_{\ell,v,i,j}} \chi_\alpha(Y)$ where $S_{\ell,v,i,j}$ is the set of hypergraph generated in the following way:

On $p$-uniform directed complete hypergraph on $n$ vertices, $S_{\ell,v,i,j}$ is the set of $p$-uniform hypergraph on $n$ vertices which can be generated in the following way

- we construct $2\ell$ levels of vertices. Level 0 contains vertex $i$ and $(p_1 v - 1)/2$ vertices in addition. For $0 < t < \ell$, level $2t$ contains $p_1 v$ vertices while for $0 < t < \ell - 1$ level $2t + 1$ contains $p_2 v$ vertices. Level $2\ell - 1$ contains vertex $j$ and $(p_2 v - 1)/2$ vertices in addition. All vertices are distinct.

- We construct a perfect matching between level $t, t + 1$ for $t \in [1, 2\ell - 2]$, each hyperedge directs from $p_1$ vertices in even level to $p_2$ vertices in odd level.

- Level 0 and 1 are connected as bipartite hypergraph s.t each vertex in level 0 excluding $i$ has degree 2 while vertex $i$ and vertices in level 1 has degree 1. Level $2\ell - 2$ and level $2\ell - 1$ are connected as bipartite hypergraph s.t vertices in level $2\ell - 1$ excluding $j$ has degree 2 while vertices in level $2\ell - 2$ and vertex $j$ has degree 1

We consider the set of hypergraph pairs $\alpha, \beta \in S_{\ell,v,i,j}$ such that in $\alpha$

- at level $t$ there are $r_t$ vertices(excluding vertice $i, j$) shared with $\beta$

- between level $t + 1$ and level $t$, there are $k_t$ hyperedges shared with $\beta$.

We denote such a set of hypergraph pairs as $S_{\ell,v,i,j,k,r}$, where $k = \sum_t k_t$, $r = \sum_t r_t$. Then $r$ is just the number of shared vertices between $\alpha, \beta$ and $k$ is just the number of shared hyperedges between $\alpha, \beta$. Although we abuse the notations(since the set $S_{\ell,v,i,j,k,r}$ is related to $k_t, r_t$), by the following lemma we can bound the size of such set only using $k, r, \ell, v$.

**Lemma C.10.** *On the directed complete $p$-uniform hypergraph with $n$ vertices, for any set $S_{\ell,v,i,j,k,r}$ with $\ell = O(\log n)$, $v = o(n)$, $k, r \leq \ell v$, the number of hypergraph pairs contained in the set $S_{\ell,v,i,j,k,r}$ is bounded by*

$$|S_{\ell,v,i,j,k,r}| \leq |S_{\ell,v,i,j}|^2 n^{-r} v^{2r - 2(1-p)k + (p-2)k/2} \ell^{O(2r - pk)} \exp(O(r))$$

*Proof.* we only need to bound the size of $S_{\ell,v,i,j}$. Applying combinatorial arguments to the generating process of $S_{\ell,v,i,j}$, we have

$$|S_{\ell,v,i,j}| = (1-o(1))\left(\binom{n}{p_1 v}\binom{n}{p_2 v}\right)^{\ell-1}\binom{n}{(p_1 v-1)/2}\binom{n}{(p_2 v-1)/2}\frac{((p_1 v)!(p_2 v)!)^{2\ell-1}}{v^{2\ell-1}2^{(pv-2)/2}}$$

On the other hand, first choose $\alpha \cap \beta$ and shared vertices(excluding $i$ and $j$) as subgraph of hypergraph $\alpha \in S_{\ell,v,i,j}$. For the shared vertices and hyperedges consisting in $\alpha$, if there are $r_t$ vertices in level $t \in [2\ell-1]$ and $k_t$ hyperedges between level $r_t$ and level $r_{t+1}$, then the number of such intersection is bounded by $N_{\alpha\cap\beta}$:

$$\binom{n}{r_0}\binom{n}{r_{2\ell-1}}\binom{2r_0+1}{p_1 k_0}\binom{2r_{2\ell-1}+1}{p_2 k_{2\ell-2}}\prod_{t=1}^{\ell-1}\binom{n}{r_{2t-1}}\binom{n}{r_{2t}}\binom{r_{2t-1}}{p_1 k_{2t-1}}\binom{r_{2t-1}}{p_1 k_{2t-1}}\binom{r_{2t}}{p_2 k_{2t}}$$

$$\binom{r_{2t}}{p_2 k_{2t-1}}\prod_{t=0}^{2\ell-2}\frac{(p_1 k_t)!(p_2 k_t)!}{k_t!}$$

This is upper bounded by

$$\prod_{t=0}^{2\ell-2}\frac{(p_1 k_t)!(p_2 k_t)!}{k_t!}\prod_{t=0}^{2\ell-1}\binom{n}{r_t}\exp(O(r))$$

. Next we choose the remaining hypergraph $\alpha \setminus \beta$ and $\beta \setminus \alpha$ respectively. For $\alpha \setminus \beta$, we have

$$N_{\alpha\setminus\beta} = \binom{n}{\frac{p_1 v-1}{2}-r_0}\binom{n}{\frac{p_1 v-1}{2}-r_{2\ell-1}}\prod_{t=1}^{\ell-1}\binom{n}{p_2 v-r_{2t-1}}\binom{n}{p_1 v-r_{2t}}$$

$$\prod_{t=0}^{2\ell-2}\frac{(p_1(v-k_t))!(p_2(v-k_t))!}{k_t!}$$

$$\leq |S_{\ell,v,i,j}|n^{-r}v^r v^{-(p-1)k}\exp(O(r))$$

Suppose there are $s_1$ degree 1 vertices in $\alpha\cap\beta$ and $s_0$ vertices shared between $\alpha,\beta$ but not contained in $\alpha\cap\beta$, denoting $s = s_0+s_1$, then there are $\ell^s$ ways of placing $\alpha\cap\beta$ and shared vertices in hypergraph $\beta$ and the count of remaining hypergraph is also bounded by $|S_{\ell,v,i,j}|n^{-r}v^r v^{-(p-1)k}\exp(O(r))$. Multiplying them together, we get the claim. $\qquad\square$

**Lemma C.11.** *For* $t \in \{0,1,\ldots,2\ell-2\}$, *we define* $r_t, k_t \in \{0,1,\ldots,p_2 v\}$ *and* $r_{2\ell-1} \in \{0,1,\ldots,p_1 v\}$ *satisfying that*

- *For* $t \in [2\ell-1]$, $r_t \geq \delta(t)\max(k_{t-1},k_t)$, *where parity function* $\delta(t) = p_2$ *if* $t$ *is odd and* $p_1$ *if* $t$ *is even.*
- $r_0+1 \geq p_2 k_0$

*We denote* $k = \sum_{t=0}^{2\ell-1} k_t$ *and* $r = \sum_{t=0}^{2\ell-1} r_t$. *We take scalars* $\eta = O(\ell^{-10})$ *and constant* $\psi < 1$. *Then for* $2r \geq pk+1$, *we have:*

$$\sum_{\substack{k_t\geq 0}}\sum_{\substack{r_t\geq\delta(t)\max(k_{t-1},k_t)\\2r\geq pk+1}}\eta^{r-pk/2}\psi^k \leq o(1)$$

*Proof.* We note that given $k_t$ for $t \in [\ell]$, we have at most $\ell^{r-(pk-1)/2}$ choices for all $r_t$. Further we denote $k_\Delta = \sum_t |k_{t+1} - k_t|$. Then given $k_\Delta$, all $k_t$ can take at most $k_\Delta$ different values. As a result, fixing these $k_\Delta$ different values, there are $\ell^{k_\Delta}$ choices for $k_t$ for $t \in [\ell]$. Further we have $r - (pk-1)/2 \geq k_\Delta/2 \geq 1/2$. Therefore the summation is bounded by

$$\sum_{k_\Delta\geq 1}\left[\left(\eta^{1/4}\ell^2\right)^{k_\Delta/2}\prod_{t=1}^{k_\Delta}\left(\sum_{k_t\geq 0}\psi^{k_t}\right)\right] = o(1)$$

$\qquad\square$

**Lemma C.12.** *Taking* $\gamma = c_p n^{p/2} v^{(p-2)/2} \lambda^2 = 1 + \Omega(1)$ *(where $c_p$ is a constant related to $p$) a large enough constant and $\ell = O(\log_\gamma n)$ in the estimator above, then if $v = O(n^{1/2 - \Omega(1)})$, we have*

$$\frac{\mathbb{E}\langle P(Y), xx^\top \rangle}{\left(\mathbb{E}\|P(Y)\|_F^2 \, \mathbb{E}\|x\|^4\right)^{1/2}} = 1 - \gamma^{-O(1)}$$

*Proof.* We need to show the estimator $P(Y) \in \mathbb{R}^n$ above achieves constant correlation with the hidden vector $x$. Equivalently we want to show that for each $i, j \in [n]$, we have

$$\left( \sum_{\alpha \in S_{\ell,v,i,j}} \mathbb{E}[\chi_\alpha(Y) x_i x_j] \right)^2 = \Omega \left( \sum_{\alpha, \beta \in S_{\ell,v,i,j}} \mathbb{E}[\chi_\alpha(Y) \chi_\beta(Y)] \right)$$

Since we have $\left( \sum_{\alpha, \beta \in S_{\ell,v,i,j}} \mathbb{E}[\chi_\alpha(Y) x_i x_j] \right) = \lambda^{(2\ell-1)v} |S_{\ell,v,i,j}|$, Moreover we have

$$\mathbb{E}[\chi_\alpha(Y)\chi_\beta(Y)] = (1 + n^{-\Omega(1)}) \lambda^{2(2\ell-1)v - 2k} \mathbb{E}\left[ \prod_{j \in \alpha \Delta \beta} x_j^{\deg(j, \alpha \Delta \beta)} \right]$$

$$\leq \lambda^{-2k} \Gamma^{O(2r - pk)} \mathbb{E}[\chi_\alpha(Y) x_i x_j] \, \mathbb{E}[\chi_\beta(Y) x_i x_j]$$

where $\deg(j, \alpha \Delta \beta)$ represents the degree of vertex $j$ in hypergraph $\alpha \Delta \beta$. By lemma C.10 the contribution to $\frac{\mathbb{E}[P_{ij}^2(Y)]}{(\mathbb{E}[P_{ij}(Y) x_i x_j])^2}$ with respect to specific $r_t, k_t$ is bounded by

$$(n^{-r} v^r v^{(1-p)k})^2 n^r v^{(p-2)k/2} \ell^s \Gamma^{O(2r-pk)} \lambda^{-2k} \exp(O(r))$$

Because we have $2r - s - 2 \geq pk$ by degree constraints, we study the terms in cases of $2r - s > pk$ and $2r - s <= pk$. For the case $2r - s > pk$, the sum of contribution is $o(1)$ by lemma C.11. For the case $2r - s = pk$, $\alpha \cap \beta$ consists of 2 hyperpaths respectively starting from $i$ and $j$. Further each shared vertex is contained in $\alpha \cap \beta$. For such case the contribution is bounded by

$$\sum_{\ell_1, \ell_2} c^{p(\ell_1 + \ell_2)} n^{-p(\ell_1 + \ell_2)/2} v^{-(p-2)(\ell_1 + \ell_2)/2} \lambda^{-2(\ell_1 + \ell_2)} \leq \left( \sum_{\ell_1} c^{\ell_1} n^{-p\ell_1/2} v^{-(p-2)\ell_1/2} \lambda^{-2\ell_1} \right)^2$$

$$\leq \left( \frac{1}{1 - v^{-(p-2)/2} n^{-p/2} \lambda^{-2}} \right)^2$$

For the case $2r - s < pk$, $\alpha \cap \beta$ contains more than $2\ell - 1$ hyperedges. For $\ell = O(\log_\gamma n)$ with hidden constant large enough, the contribution is also $o(1)$. Therefore in all we have

$$\frac{\left( \mathbb{E}\langle P(Y), xx^\top \rangle \right)^2}{\mathbb{E}\|P(Y)\|_F^2 \, \mathbb{E}\|xx^\top\|_F^2} = 1 - v^{-(p-2)/2} n^{-p/2} \lambda^{-2} + o(1)$$

Therefore when $n = \omega(cv^2 \Gamma^2 \mathrm{polylog}(n))$ and $\gamma = 1 + \Omega(1)$ a large enough constant, we have weak recovery algorithm by taking the leading eigenvector of matrix $P(Y)$. When $\gamma = \omega(1)$, taking leading eigenvector of $P(Y)$ gives strong recovery guarantee. $\qquad \square$

Next we evaluate the polynomial estimator for weak recovery using color-coding method, as shown in algorithm 7. We denote $\ell_v = \frac{p(2\ell-1)v+1}{2}$ as the number of vertices in each hypergraph contained in $S_{\ell,v,i,j}$.

Next we describe how to construct matrices $M, N, A, B$. The rows and columns of $M$ are indexed by $(V_1, S)$ and $(V_2, T)$ where $V_1 \in [n]^{p_1 v}$ and $V_2 \in [n]^{p_2 v}$ are set of vertices while $S, T \subsetneq [\ell v]$ are subset of colors. We have $M_{(V_1, S), (V_2, T)} = 0$ if $S \cup \{c(v) : v \in V_1\} \neq T$ or $\{c(v) : v \in V_1\}$ and $S$ are not disjoint. Otherwise $M_{(V_1, S), (V_2, T)}$ is given by $\sum_{\gamma \in S_{V_1, V_2}} \chi_\gamma(Y)$ where $S_{V_1, V_2}$ is the set of perfect matching induced by $V_1$ and $V_2$(each hyperedge in $S_{V_1, V_2}$ direct from $p_1$ vertices from $V_1$ to $p_2$ vertices from $V_2$).

For matrix $N$, the indexing are the same as $M$. We have $N_{(V_1, S), (V_2, T)} = 0$ if $S \cup \{c(v) : v \in V_1\} \neq T$ or $\{c(v) : v \in V_1\}$ and $S$ are not disjoint. Otherwise $N_{(V_1, S), (V_2, T)}$ is given by $\sum_{\gamma \in S_{V_2, V_1}} \chi_\gamma(Y)$

**Algorithm 7:** Algorithm for evaluating estimation matrix

---

**Data:** Given $Y \in (\mathbb{R}^n)^{\otimes p}$ s.t $Y = \lambda x^{\otimes p} + W$

**Result:** $P(Y) \in \mathbb{R}^{n \times n}$, with $P_{ij}(Y) = \sum_{\alpha \in S_{\ell,v,i,j}} \chi_\alpha(Y)$(up to accuracy $1 \pm n^{-\Omega(1)}$)

$C \leftarrow \exp(100\ell v)$;

**for** $i \leftarrow 1$ **to** $C$ **do**

    Sample coloring $c_i : [n] \mapsto [\ell_v]$ uniformly at random;

    Construct matrices $M, N \in \mathbb{R}^{(2^{\ell_v}-1)n^{p_1 v} \times (2^{\ell_v}-1)n^{p_2 v}}$;

    Construct matrices: $A \in \mathbb{R}^{n^{(p_1 v+1)/2} \times (2^{\ell_v}-1)n^{p_2 v}}$ ,$B \in \mathbb{R}^{(2^{\ell_v}-1)n^{p_1 v} \times n^{(p_2 v+1)/2}}$;

    Construct matrices $L^{(1)} \in \mathbb{R}^{n \times n^{(p_1 v+1)/2}}, L^{(2)} \in \mathbb{R}^{n^{(p_2 v+1)/2} \times n}$ ;

    Record matrix $p_{c_i} = L^{(1)} A (NM)^{\ell-2} N B L^{(2)}$;

Return $\frac{1}{C} \sum_{i=1}^{C} p_{c_i}$

---

where $S_{V_2,V_1}$ is the set of perfect matching induced by $V_2$ and $V_1$(each hyperedge in hypergraph $\alpha \in S_{V_2,V_1}$ directs from $p_2$ vertices from $V_2$ to $p_1$ vertices from $V_1$).

We consider a subset of hypergraphs contained in $S_{\ell,v,i,j}$ with $i,j \in [n]$, denoted by $\mathcal{H}_{i,j,V_1,V_2}$. The set of vertices in level 0 of these hypergraphs is fixed to be $\{i\} \cup V_1$, where $V_1 \subseteq [n]$. The set of vertices in the level 1 of these hypergraphs is fixed to be $V_2 \subseteq [n]$. We denote $S_{i,V_1,V_2}$ as the following set of spanning subgraphs: a hypergraph $\alpha \in S_{i,V_1,V_2}$ if and only if there exists a hypergraph $\beta \in \mathcal{H}_{i,j,V_1,V_2}$ such that the hyperedge set of $\alpha$ is the same as the set of hyperedges between level 0 and 1 of $\beta$.

In the same way, we consider a subset of hypergraphs contained in $S_{\ell,v,i,j}$ with $i,j \in [n]$, denoted by $\mathcal{L}_{i,j,V_1,V_2}$, with vertices in levels $2\ell-2$ and $2\ell-1$ fixed. We denote $S_{V_1,V_2,j}$ as the following set of spanning subgraphs: a hypergraph $\alpha \in S_{V_1,V_2,j}$ if and only if there exists a hypergraph $\beta \in \mathcal{L}_{i,j,V_1,V_2}$ such that the hyperedge set of $\alpha$ is the same as the set of hyperedges between level $2\ell-2$ and $2\ell-1$ of $\beta$.

By these definitions, the entry $A_{(i,V_1),(V_2,T)}$ is given by $\sum_{\alpha \in S_{i,V_1,V_2}} \chi_\alpha(Y)$ if $T = \{c(v) : v \in V_1 \cup v\}, v \notin V_1$ and 0 otherwise. The entry $B_{(V_1,S),V_2}$ is given by $\sum_{\alpha \in S_{V_1,V_2}} \chi_\alpha(Y)$ if $S \cup \{c(v) : v \in V_1 \cup V_2\} = [\ell_v], S \cap \{c(v) : v \in V_1 \cup V_2\} = \emptyset$ and zero otherwise.

Finally, we construct the deterministic matrices $L^{(1)}, L^{(2)}$. The columns of matrix $L^{(1)}$ are indexed by $(j,V_1)$, where $j \in [n]$ and $V_1$ is a size-$n^{(p_1 v+1)/2}$ subset of $[n]$. The rows are indexed by $i \in [n]$. The entry $L^{(1)}_{i,(j,V_1)} = 1$ if $j = i$ and 0 otherwise. The transpose of $L^{(2)}$ is indexed in the same way. The entry $L^{(2)}_{(j,V_1),i} = 1$ if $i = j$ and 0 otherwise.

**Lemma C.13** (Evaluation of polynomial estimator). *Given sampled colorings $c_1, c_2, \ldots, c_C : [n] \to [\ell_v]$ and a tensor $Y \in (\mathbb{R}^n)^{\otimes p}$, the algorithm 7 return a matrix $p(Y, c_1, \ldots, c_C) \in \mathbb{R}^{n \times n}$ in time $n^{O(pv)}$. When $\frac{\mathbb{E}\langle P(Y), xx^\top \rangle}{n (\mathbb{E}\|P(Y)\|_F^2)^{1/2}} = \delta = \Omega(1)$, with high probability over the sampling of random colorings $c_1, c_2, \ldots, c_C$ we have the following guarantee:*

$$\frac{\mathbb{E}_Y \langle p(Y, c_1, \ldots, c_C), xx^\top \rangle}{n (\mathbb{E}_Y \|p(Y, c_1, \ldots, c_C)\|_F^2)^{1/2}} \geq (1 - o(1))\delta \geq \Omega(1).$$

**Remark**: When $\lambda = \omega(n^{-p/4})$, using power method for extracting leading eigenvector, we have $n^{p+o(1)}$ time algorithm for evaluating the leading eigenvector of the matrix returned by the algorithm.

*Proof.* For any coloring $c : [n] \mapsto [\ell_v]$, and any $\alpha \in S_{\ell,v,i,j}$, we define indicator random variable $F_{c,\alpha}$ which equals to 1 if the vertices in $\alpha$ are assigned different colors and 0 otherwise. Then we have $\sum_{\alpha \in S_{\ell,v,i,j}} \chi_\alpha(Y) F_{c,\alpha}$ can be obtained from the matrix $H = A(NM)^{\ell-2}NB$ in the algorithm by summing up all entries in $H$ indexed by row $(i, \cdot)$ and column $(j, \cdot)$. Thus given random coloring

$c$ the algorithm evaluates matrix $p_c(Y) = L^{(1)}A(NM)^{\ell-2}NBL^{(2)}$ whose $(i,j)$ entry is given by

$$p_{c,i,j}(Y) = \sum_{\alpha \in S_{\ell,v,i,j}} \chi_\alpha(Y) F_{c,\alpha}$$

By the same argument in the strong detection algorithm, we can obtain accurate estimation of $P(Y)$ by averaging $n^{pv} \exp(O(\ell v))$ random colorings when $\gamma = c_p \lambda^2 n^{p/2} v^{(p-2)/2} = 1 + \Omega(1)$ where $c_p$ is small enough constant related to $p$. Therefore when $v$ is chosen such that $\gamma$ is a large enough constant, taking the leading eigenvector of $D(Y) = \frac{1}{L} \sum_{t=1}^{L} p_{c_t}(Y)$ generates an estimator achieving weak recovery. Since $\ell = O(\log_\gamma n)$, the polynomial can be evaluated in time $n^{O(pv)}$ when we have $c_p \lambda^2 n^{p/2} v^{(p-2)/2} > 1$ and $v = O(n^{1/2-\Omega(1)})$. This leads to polynomial time algorithm when $v$ is constant.

$\square$