[Reviews · NeurIPS 2020]

Review 1

Summary and Contributions: This paper studies recovery in a generalized version of the spiked Wigner model, where we are given Y = \lambda xx^T + W, we want to recover x, and W is a symmetric noise matrix with iid entries. In the case where the entries of W are reasonably nice (e.g. Gaussian) this is a classic and well-studied generative model for PCA, where taking the top eigenvector is an optimal algorithm. In this paper, the focus is on more heavy-tailed models where the entries of W may have infinite fourth moment (i.e. unbounded kurtosis). As noted in the paper, this is somewhat related to the study of community detection in sparse stochastic block models with 2 communities, which has been a pretty well-studied and well-understood problem. The reason sparsity is relevant is that if we make W have sparse entries (by making the entries 0 with a large probability), then the sparser we make it, the worse behaved the higher moments become. The authors also consider the extension to a 3-tensor version of the model which in the non-heavy-tailed case has been fairly well studied.

Strengths: Strengths: I found the explanation of this result and it's context in the previous literature easy to understand, and the result is clean and solves a problem in polynomial time for which there were previously no guarantees. I appreciated the inclusion and discussion of the truncation-based algorithm which helped clarify the new aspects of the self-avoiding walk based analysis.

Weaknesses: The techniques seem, generally speaking, similar to the previous work cited. The analysis is for an impractical version of the algorithm (which uses self-avoiding walks), whereas it would be preferable to have it for the more practical version using nonbacktracking walks from the experiments; as mentioned in the text, in the SBM literature results of the latter form are indeed known.

Correctness: I didn't have the chance to check the proofs in full detail, but the overall proof strategy makes sense and I didn't see any errors in the parts I read.

Clarity: Overall the paper is well written. One minor complaint is that it was hard to find the dependence on \delta which is not stated explicitly in or around the main theorem, though it is interesting to know. However it looks like from the statement of Lemma 2.6 we can find the dependence is n^{poly(1/\delta)}.

Relation to Prior Work: Yes, there is a significant discussion of the context in terms of PCA, known results on the SBM etc.

Reproducibility: Yes

Additional Feedback: Figure 1: the fact that the colors between different subplots are not consistent is unnecessarily confusing, this should be fixed. There are many missing periods for whatever reason. E.g. end of line 186, 195, 256. --- post-rebuttal: my overall opinion is unchanged. i agree with the other reviewers that some parts of the writing can be improved.


Review 2

Summary and Contributions: In the classic spiked Wigner model we observe an n \times n matrix Y = \lambda x x^T + W where \|x\|_2 = \sqrt{n} and W is a random matrix whose entries are standard Gaussians. Moreover \lambda is a parameter that controls the signal-to-noise ratio and the goal is to understand when it is possible to estimate x non-trivially. It is known that this is possible if and only if \lambda \sqrt{n} > 1 and this is called the Baik, Ben-Arous, Peche transition. In fact taking the top eigenvector of Y suffices. This paper studies a generalization whereby the entries of W are allowed to be heavy tailed. All that is assumed is that their mean is zero and their variance is one, but they might not even have a 2.1^{th} moment. In this setting taking the top eigenvector no longer succeeds in reaching the tight information-theoretic threshold. (In fact, as the paper shows if the distributions of the entries in W are allowed to be different and are chosen in an adversarial manner then the top eigenvector badly fails.) The paper's main contribution is to design algorithms based on applying spectral methods to a matrix that counts self-avoiding walks. These methods work down to the same threshold as in the Gaussian case and are hence optimal (at least for the detection problem). They also give generalizations to spiked tensor models where they recover the same tradeoff between the signal-to-noise ratio and the subexponential running time as other methods that work with stronger assumptions on the moments (and this matches the low degree threshold that has been used to predict where the problem becomes computationally hard).

Strengths: The strength of the paper is that it presents a clean extension to the well-studied spiked Wigner model that allows for heavy-tailed noise and resolves the threshold for detection exactly. It also gets algorithms for spiked tensor models with heavy tailed noise that match the state-of-the-art that work under stronger assumptions. The paper has a new technique (although the other techniques already appeared in the work of Hopkins and Steurer) that is interesting. I will discuss both the overlap and new ideas below.

Weaknesses: The main weakness of the paper is that it is mostly an adaptation of the techniques in the work of Hopkins and Steurer for community detection in sparse random graphs. Personally I don't think this is such a big deal. I think it's still nice to get compelling extensions by using existing techniques. Since the beginning it has been known that a major difficulty of community detection in the sparse stochastic block model is that there are nodes with logarithmic degree, and the top eigenvectors will localize around them. Both spectral methods on counts of self avoiding (and non-backtracking) walks and belief propagation do not have this pitfall. Hopkins and Steurer showed that the walks that do repeat nodes contribute high variance terms to an unbiased estimator of xx^T. They showed that self avoiding walks have lower variance, and they give rise to a polynomial that correlates with xx^T in a way that it can be rounded to solve community detection. Moreover they showed a way to use color coding to speed up the computation of the number of self-avoiding walks that gets the algorithm down from quasipolynomial to polynomial. In the special case when the spike x is \pm 1 valued essentially the same strategy works. This time repeated nodes correspond to higher moments of the entries of W and thus may make the estimator have unbounded variance. The paper does give a clever new idea in how to adapt the methods to arbitrary x's, provided that there is a bound on their \infty norm and that its entries are non-trivially spread out. Their analysis is based on controlling the bias of an average self-avoiding walk. I think this is an important idea that may have other applications.

Correctness: The paper looks correct to me.

Clarity: The paper is very well written and easy to follow. Though, this may be because I was already familiar with the Hopkins and Steurer approach and color coding.

Relation to Prior Work: The relationship to prior work is clearly explained.

Reproducibility: Yes

Additional Feedback:


Review 3

Summary and Contributions: The authors study recovery of a symmetric rank-one signal from a noisy symmetric data matrix or higher-order tensor, when the noise has independent entries with unit variance but potentially heavy tails. In heavy-tailed settings, the leading eigenvector can be dominated by extreme values of the noise, and be uncorrelated with the true signal. The main results of the paper show that weak recovery (i.e. estimating a vector with positive correlation to the true signal) is nonetheless achievable under a mild delocalization property of the true signal, in essentially the same settings as when the noise is not heavy-tailed: In polynomial time for matrices everywhere above the BBP phase transition, and in polynomial time or sub-exponential time for tensors up to the same computational barriers as conjectured for Gaussian noise. The authors show in the tensor setting that furthermore weak recovery may be strengthened to strong recovery (i.e. achieving correlation 1-o(1)) using a power iteration. For matrices, the algorithm analyzed is based on a self-avoiding walk matrix, where the (i,j) entry is a polynomial that sums over all self-avoiding walks of length O(log n) from i to j of the product of matrix entries along the walk. In the tensor setting, the algorithm is based on low-degree polynomials constructed from a generalization of this self-avoiding walk construction to hyper-edges. In both cases, the main result follows from a first-moment and second-moment calculation for these polynomials. Direct implementations of these algorithms are not polynomial time, but the authors show how to adapt a previous randomized coloring idea to provide sufficiently accurate approximations in polynomial time.

Strengths: The theoretical results are reasonably sharp. Recovery guarantees are established for matrices everywhere above the BBP threshold and for any true signal vector where ||x||_infty / ||x||_2 < n^{-1/2-delta}, and it is clear that these conditions are essentially necessary. The results for tensors make a stronger assumption that the signal vector x has i.i.d. entries, but the requirements for the signal-to-noise also match (what are believed to) the correct thresholds for both polynomial-time and sub-exponential-time recovery.

Weaknesses: I'm not sure if the level of novelty in this work is sufficiently high for publication in NeurIPS. Most of the ideas were developed in the analysis of community-detection models for sparse networks, where similar heavy-tailed phenomena arise. In particular, these types of first-moment and second-moment computations for self-avoiding walks, generalizations to higher-order tensors, and the coloring idea for producing a polynomial-time algorithm, were all already developed in some form in the cited [HS17] reference. While this paper studies more general distributions and conditions for both the noise and the entries of the signal vector, these extensions seem somewhat minor.

Correctness: I've only made a high-level check of the arguments, but I believe that they are correct.

Clarity: The introduction is well-written, but I find the exposition of the rest of the paper to be somewhat sub-par. Overall, there are too many scattered threads---matrix estimation, tensor estimation, truncation methods, non-backtracking walks, strong versus weak recovery---and different results often require slightly different assumptions and provide slightly different guarantees, detracting from the coherence of the paper. Sections 2.1 and 2.3 feel distracting. Notation in Lemma 2.5 is not defined. It is difficult to understand the construction of lines 275-284 in Definition 3.1, which constitutes the core idea of the tensor analysis. I also don't find the proof of Lemma 2.4 to be clearly explained. (E.g. line 223, where does the relation p <= r-s-k come from? Line 224, why are there n^{2(l-1)-r}l^{O(r-k)} pairs of alpha and beta? Display equation after line 227, what does the notation of the second expectation mean? Line 237, what does "so l^O(1) factor diminishes" mean?)

Relation to Prior Work: Yes, the relation to prior work is clearly and adequately discussed.

Reproducibility: Yes

Additional Feedback: A few possible typos: - Line 42, missing sqrt(n) - Line 117, missing factor of lambda^l - Display after line 213: one of the l's is off by 1, maybe in the product on the left side - Display after line 217 is missing an expectation


Review 4

Summary and Contributions: The theoretical guarantees for PCA are characterized by the renowned BBP transition. However, the proof of this transition along with the analyses of all eigenvector-based methods seem to either require assumptions on the tails of the entries of the matrix or symmetry assumptions such as the entries being i.i.d. (i.e. the truncation algorithm in Section 2.1). The paper introduces a self-avoiding walk/color-coding algorithm (an extension of ideas in HS17) to perform rank-1 matrix estimation when eigenvectors methods provably fail. Furthermore, the guarantees for this algorithm hold up to the BBP transition which is optimal. The paper also presents a natural extension to tensor PCA. UPDATE: Thank you to the authors for the response and changes. The proposed changes sound good and like they would answer my questions. My review remains that this paper should be accepted.

Strengths: The main strength of this paper is that it makes an important contribution to a well-studied problem. The setting of heavy-tailed noise is important across several communities: (1) in random matrix theory, heavy-tailed noise (and similarly sparse matrices) are notoriously difficult to analyze, and (2) heavy-tailed noise naturally arises when studying robustness and adversarial corruptions. It is surprising that no prior work has tried to perform weak recovery in spiked matrix models up to the BBP transition in the presence of heavy tailed noise. Furthermore, the analysis and color-coding technique in the paper is flexible and may find applications elsewhere. Tensor PCA has been studied extensively in the last few years from an algorithmic perspective. The guarantees for the algorithm here on tensor PCA seem to be stronger than most of what is in the literature, making much weaker assumptions. Often algorithms in the literature even assume Gaussian noise.

Weaknesses: As discussed in the paper, the algorithm for the matrix case presented here seems to be an extension of the ideas in HS17. Furthermore, the idea of reducing variance with nonbacktracking/self-avoiding walks has appeared previously in the literature on the stochastic block model. While it does seem as though a different analysis is necessary in this paper and there are technical complications that are nontrivial to overcome, the core ideas seem similar. It seems as though more technical novelty lies in the extension to the tensor case, which is not as heavily discussed. It might be better to defer some of the proofs in Section 2 to the supplementary material in favour of only outlines of these proofs and a longer discussion of the ideas in the tensor case. For example, it is never made clear why x must be random in the tensor case while it can be arbitrary in the matrix case. The presentation and organization in the paper could use improvement. Section 1 is well-written and does a good job contextualizing the results of the paper. However, Sections 2 and 3 and parts of the supplementary material have a number of typos and feel disorganized. These should be edited to be consistent with the style and organization in Section 1. While Section 2.1 is important to contextualize the main color-coding/SAW algorithm, this feels like something that can be more of a qualitative remark in the main body of the paper and deferred to the appendix. The proof of Lemma 2.4 seems to essentially be an involved/difficult mean and variance computation. Rather than go through the details of this lemma in the body, it may be better to summarize the main ideas, defer this proof to the appendix, and move some description of the color-coding part of the algorithm into the body. While this color-coding part is likely deferred to the appendix because it is not the novel part of the paper (uses ideas similar to HS17), to a reader curious how the SAW statistic can be computed in polynomial time, it would be preferable to not have to find this in the supplement. While the presentation of the later sections in the paper could use improvement, this has no bearing on the content of the paper which I think is strong. Also fixing the presentation in this paper doesn’t seem like it would be very difficult to do. Furthermore, while some techniques are borrowed from HS17, the applications here (especially to tensor PCA) seem different enough and important enough on their own that this isn’t a serious downside. Overall, I think this paper should be accepted based on the fact that it makes interesting contributions to an important problem.

Correctness: As far as I can tell, the proofs seem to be correct. The empirical comparison of various techniques is on synthetic data and seems to be sound. It would be helpful if this were better explained in Section 2.4. For example, what are “naive” and “worst” in the plots? Why do some plots have SAW and others have truncate? I am guessing this is because SAW is slow to compute and when n = 2000 it’s no longer possible. I think there should be at least one plot with all five methods (naive, backtracking, SAW, truncate and worst).

Clarity: As mentioned above, the introduction s very well-written while the other sections come across as a little disorganized. However, this seems easy to fix.

Relation to Prior Work: The results of the paper are well-motivated and contextualized in Section 1. However, in light of the fact that no prior work seems to have directly addressed the problem at hand — the BBP transition with heavy tails — it would be helpful for context to discuss literature that even is tangentially relevant. For example, some prior work has directly tried to address ways to fix the spectral concentration of sparse random graphs (which are heavy-tailed when shifted and normalized) by deleting vertices e.g. “Sparse random graphs: regularization and concentration of the Laplacian” by Le et al.

Reproducibility: Yes

Additional Feedback: - Extra “of” in line 55 - Space missing in line 84 - Forgot period in line 124 - Section 2 could use some grammatical editing - Should there be an expectation on line 216? - Vertice everywhere should be replaced with vertex - Is xx^T in line 287 a typo (should be x^{\otimes 3}) - The comment on line 292 in the conclusion could use a little more discussion. How general is the result in PWBM18 i.e. does it require light tails?

[Author Response · NeurIPS 2020]

We are very grateful to the reviewers for their constructive and detailed feedback. We address specific reviewer
comments and questions below; we will incorporate all the feedback into any final version of our paper.

**Novelty of the paper** (Reviewer 3): *What's the main novelty of this paper, in light of techniques in prior work?*

We agree that the core self-avoiding walk technique (as well as the color-coding technique) appeared in HS17 in the
context of stochastic block models, and that nonbacktracking walks appeared in other prior work on stochastic block
models. However, we feel that the main result of our paper – a polynomial-time algorithm for the spiked matrix model
with heavy-tailed noise – is compelling on its own, because it provides a sharp algorithmic guarantee for a very simple
and widely-studied problem. Our results also illustrate the versatility of the self-avoiding walk technique, in directions
which were not obvious from prior literature, which focused on stochastic block models. For example,

- the technique can provide sharp guarantees for **general noise distributions**, not just **discrete distributions**,
- the technique does not require the assumption that the entries in **spiked vector** are sampled from some **i.i.d**
**distribution with zero mean and unit variance**,
- the technique can extend beyond matrix settings, to handle **spiked tensor models**.

Furthermore, our work overcomes nontrivial technical difficulties in extending the self-avoiding walk method to these
more general settings – for instance, allowing general spike vectors $x$ (rather than $x \in \{\pm 1\}^n$ as in the block model
setting) requires a substantially more challenging analysis of both the mean and the variance of the self-avoiding walk
estimator.

**Exposition**: *Writing in sections 2 and 3 is uneven*

Several reviewers remarked that while the introduction to our paper is well written, the exposition in sections 2 and 3 is
more uneven. We are grateful for this feedback. We have already worked to improve the exposition in these sections,
and we will additionally incorporate the feedback from reviewers in the final version of our paper.

**Technical clarifications and detailed responses**

*Dependence on $\delta$* (Reviewer 1): The dependence of $\delta$ in the main theorem is indeed $n^{\text{poly}(1/\delta)}$.

*Clarifications in proof of Theorem 2.4* (Reviewer 3): We will include proof of the graph-theoretic relation $p \le r - s - k$
and the bound $n^{2(\ell-1)-r} \ell^{O(r-k)}$ in a revised full version. Following the line 227, the second expectation is uniformly
taken over the labeling of $2(\ell - 1) - r$ vertices in $\alpha \cup \beta$. Finally, '$\ell^{O(1)}$ diminish' means that for $r = k$, the bound is
given by $(1 + n^{-\Omega(1)})\lambda^{2\ell} n^{2(\ell-1)} n^{-k} \lambda^{-2k}$, without additional $\ell^{O(1)}$ factor. Again, we will clarify these points upon
revision.

*Clarification in the statement of lemma 2.5* (Reviewer 3): Let $V \subseteq [n]$, $\|x\| = \sqrt{n}$ and $t_1, t_2 \in \mathbb{N}$. We define the
quantity $S_{t_1,t_2,V} = \mathbb{E}_{(v_1,\ldots,v_{t_1+t_2}) \subseteq [n] \setminus V} \left[ \prod_{i=1}^{t_1} x_{v_i}^2 \prod_{i=t_1+1}^{t_1+t_2} x_{v_i}^4 \right]$ where $(v_1, v_2, \ldots, v_{t_1+t_2})$ is uniformly sampled
from all size-$(t_1 + t_2)$ ordered subsets of $[n] \setminus V$ (without repeating elements). Then assuming $|V|, t_1, t_2 = O(\log n)$
and $\|x\|_\infty^2 = n^{1-\Omega(1)}$, we have $S_{t_1,t_2,V} \le (1 + n^{-\Omega(1)})\|x\|_\infty^{2t_2}$. Further if $t_2 = 0$, we have $S_{t_1,t_2,V} \ge 1 - n^{-\Omega(1)}$.

*Notation in figures describing experiments* (Reviewer 4): The label "naive" corresponds to the naive PCA algo-
rithm(extracting the leading eigenvector). The label "worst" corresponds to the information-theoretically optimal
recovery rate in case of Gaussian noise.

*Why do some figures only have curves of truncation method and some figures only have curves of self-avoiding walk*
*estimator?* The scale $2000 \times 2000$ will be **computationally expensive** for **self-avoiding walk estimator**; at this scale
we can still run the truncation-based algorithm. We thank reviewer 4 for the suggestion of putting all five methods in at
least one figure (this can be done for matrices with smaller dimension) – we will add such a figure.

*In the spiked matrix model, the authors only make assumption about the infinity norm of the spiked vector. However in*
*the spiked tensor model, the authors assume that the entries are i.i.d sampled with zero mean and unit variance. Is this*
*difference essential?* (Reviewer 3, Reviewer 4): This difference is not essential. We can prove similar guarantee for
spiked tensor model using nearly the same techniques, only assuming bound on the infinity norm of the spiked vector.
However, the proof becomes lengthier. We will discuss this in a revised version of our paper.

*Does the result in [PWBM18] require light tail?* (Reviewer 4): [PWBM18] requires the 10-th moment of each entry in
the noise matrix to remain constant as $n \to \infty$.

*Typos:* We are grateful to several reviewers for supplying a list of typos and small errors; we will address all of these in
a revised manuscript.

*Relevant literature:* We are grateful to reviewer 4 for pointing out a relevant literature.

[Meta-Review · NeurIPS 2020]

Though the paper conceptually follows prior work (most relevantly Hopkins-Steurer), all reviewers agreed there is a significant amount of technical analysis required, and the result would be of interest to the relevant NeurIPS subcommunity. The reviewers also agreed that the paper could use from significant amount of polishing (please see their reviews) -- we encourage the authors to take these comments to heart.